# Euclidean-Norm-Induced Schatten-p Quasi-Norm Regularization for Low-Rank Tensor Completion and Tensor Robust Principal Component Analysis

**Jicong Fan**  *fanjicong@cuhk.edu.cn*
*The Chinese University of Hong Kong, Shenzhen, and Shenzhen Research Institute of Big Data*

**Lijun Ding**  *ljding@uw.edu*
*University of Washington*

**Chengrun Yang**  *cy438@cornell.edu*
*Cornell University*

**Zhao Zhang**  *cszzhang@gmail.com*
*Hefei University of Technology*

**Madeleine Udell**  *udell@stanford.edu*
*Stanford University*

**Reviewed on OpenReview:** *https://openreview.net/forum?id=Grhi8OOjVz*

## Abstract

The nuclear norm and Schatten-$p$ quasi-norm are popular rank proxies in low-rank matrix recovery. However, computing the nuclear norm or Schatten-$p$ quasi-norm of a tensor is hard in both theory and practice, hindering their application to low-rank tensor completion (LRTC) and tensor robust principal component analysis (TRPCA). In this paper, we propose a new class of tensor rank regularizers based on the Euclidean norms of the CP component vectors of a tensor and show that these regularizers are monotonic transformations of tensor Schatten-$p$ quasi-norm. This connection enables us to minimize the Schatten-$p$ quasi-norm in LRTC and TRPCA implicitly via the component vectors. The method scales to big tensors and provides an arbitrarily sharper rank proxy for low-rank tensor recovery compared to the nuclear norm. On the other hand, we study the generalization abilities of LRTC with the Schatten-$p$ quasi-norm regularizer and LRTC with the proposed regularizers. The theorems show that a relatively sharper regularizer leads to a tighter error bound, which is consistent with our numerical results. Particularly, we prove that for LRTC with Schatten-$p$ quasi-norm regularizer on $d$-order tensors, $p = 1/d$ is always better than any $p > 1/d$ in terms of the generalization ability. We also provide a recovery error bound to verify the usefulness of small $p$ in the Schatten-$p$ quasi-norm for TRPCA. Numerical results on synthetic data and real data demonstrate the effectiveness of the regularization methods and theorems.

## 1 Introduction

Low-rank tensor completion (LRTC) (Gandy et al., 2011; Acar et al., 2011; Liu et al., 2012; Romera-Paredes & Pontil, 2013; Kressner et al., 2014; Yuan & Zhang, 2016; Cheng et al., 2016; Zhou et al., 2017; Lacroix et al., 2018; Ghadermarzy et al., 2019; Liu & Moitra, 2020; Wimalawarne & Mamitsuka, 2021; Yang et al., 2021; Fan, 2022), as a high-order generalization of low-rank matrix completion (LRMC) (Candès & Recht, 2009; Hardt, 2014; Fan & Chow, 2017), aims to recover the missing entries of a low-rank tensor. One may organize LRTC methods into different categories according to the types of decomposition model, e.g., CP

(CANDECOMP/PARAFAC) decomposition based LRTC (Acar et al., 2011; Jain & Oh, 2014; Zhao et al., 2015; Liu & Moitra, 2020), Tucker decomposition bassed LRTC (Xu et al., 2013; Xu & Yin, 2013; Kasai & Mishra, 2016; Xie et al., 2018; Kong et al., 2018), and tensor ring based LRTC (Yuan et al., 2019). This paper will focus on CP decomposition based LRTC.

It is known that the tensor nuclear norm is hard to compute in practice. One has to consider other tractable solutions such as learning a CP model directly from incomplete data (Acar et al., 2011; Jain & Oh, 2014). Jain & Oh (2014) proved that an $n \times n \times n$ tensor of rank $r$ can be recovered from $O(n^{3/2} r^5 \log^4(n))$ randomly sampled entries, via alternating minimization. Barak & Moitra (2016), Potechin & Steurer (2017), and Foster & Risteski (2019) used the sum-of-squares hierarchy as a tractable relaxation to the tensor nuclear norm. For 3rd-order tensor completion, Bazerque et al. (2013) used the sum of squared Frobenius norms as a regularizer and showed that it is related to the $\ell_{2/3}$ norm of the weights in CP decomposition. Yang et al. (2016) applied group-sparse regularization to LRTC and showed that the regularizer is related to the $\ell_{1/3}$ norm. Shi et al. (2017) proposed to directly minimize the $\ell_1$ norm of the weights in CP decomposition for LRTC. The regularizers presented in the above three works are closely related to the tensor Schatten-$p$ (quasi) norm with $p = 1$, $2/3$, or $1/3$. Notice that these works (Bazerque et al., 2013; Yang et al., 2016; Shi et al., 2017) are purely empirical-motivated and have no theoretical guarantee on the tensor completion performance. Moreover, the regularizations are limited in the discrete values $\{1, 2/3, 1/3\}$ and only for 3rd-order tensors. One may expect to exploit Schatten-$p$ quasi-norm with arbitrary $p$ on arbitrary-order tensor and have theoretical guarantees for the recovery.

Besides LRTC, tensor robust principal component analysis (TRPCA) is another important problem of tensor recovery. TRPCA is a generalization of robust PCA (De la Torre & Black, 2001; Ding et al., 2006; Candès et al., 2011; Haeffele & Vidal, 2019; Fan & Chow, 2020) and aims to decompose a noisy tensor to the sum of a low-rank tensor and a sparse tensor. Many recent works on TRPCA can be found in (Anandkumar et al., 2016; Lu et al., 2016; Xie et al., 2018; Goldfarb & Qin, 2014; Zheng et al., 2019; Lu et al., 2019; Wang et al., 2020). For example, Lu et al. (2019) defined a new tensor nuclear norm based on the $t$-product (Kilmer & Martin, 2011) of tensors and provided sufficient conditions for exact recovery. Notice that these TRPCA algorithms have high computational costs on large-scale data. One approach to reducing the computational costs is using low-rank factorization, which, however, requires the estimation of rank.

In this paper, we focus on fast and accurate LRTC and TRPCA. Our contributions are two-fold:

- We propose a new class of regularizers as a tensor rank proxy for tensors of arbitrary order. These are based on the Euclidean norms of the component vectors in the form of CP decomposition. We show that the regularizers are monotonic transformations of Schatten-$p$ quasi-norms, where $p$ could be any positive value. We also provide asymmetric variational forms for the tensor Schatten-$p$ quasi-norm with discrete $p$. When applying the regularizers to LRTC and TRPCA, empirically, the recovery performance is robust to the choice of initial rank and the recovery accuracy is high when $p$ is much less than 1.

- We provide generalization error bounds for LRTC with Schatten-$p$ quasi-norm regularization and LRTC with our regularizers. We show that a smaller $p$ (but not too small) in the tensor Schatten-$p$ quasi-norm or a smaller $q$ in the proposed regularizers leads to a tighter generalization error bound. More importantly, our theory indicates that for LRTC on $d$-order tensors, $p = 1/d$ is always better than any $p > 1/d$ in terms of the generalization error bound. Note that our bounds are also applicable to 2-order tensors. In other words, our bound works for LRMC. We also provide a recovery error bound for TRPCA with Schatten-$p$ quasi-norm regularization, which verifies the usefulness of small $p$ in TRPCA.

The experiments of LRTC and TRPCA on synthetic data, image inpainting, and image denoising demonstrate the effectiveness of the proposed regularization methods in comparison to a few baselines. Moreover, the numerical results related to the Schatten-$p$ quasi-norm or the proposed regularizers are consistent with the error bounds in our theorems.

## 2 Euclidean Norm Regularization (ENR) for Tensor Rank

First of all, recall the following definitions in terms of the CP decomposition.

**Definition 1.** *Let $\boldsymbol{x}_i^{(j)} \in \mathbb{R}^{n_j \times 1}$, $i \in [r]$, $j \in [d]$. The rank of a tensor $\boldsymbol{\mathcal{X}} \in \mathcal{R}^{n_1 \times n_2 \ldots \times n_d}$ is defined as the minimum number of rank-one tensors that sum to $\boldsymbol{\mathcal{X}}$:*

$$\mathrm{rank}(\boldsymbol{\mathcal{X}}) = \min \left\{ r \in \mathbb{N} : \ \boldsymbol{\mathcal{X}} = \sum_{i=1}^{r} \boldsymbol{x}_i^{(1)} \circ \boldsymbol{x}_i^{(2)} \cdots \circ \boldsymbol{x}_i^{(d)} \right\}.$$

**Definition 2.** *The nuclear norm (Friedland & Lim, 2018) of a tensor $\boldsymbol{\mathcal{X}}$ is defined as*

$$\|\boldsymbol{\mathcal{X}}\|_* = \inf \left\{ \sum_{i=1}^{r} |s_i| : \ \boldsymbol{\mathcal{X}} = \sum_{i=1}^{r} s_i \boldsymbol{u}_i^{(1)} \circ \boldsymbol{u}_i^{(2)} \cdots \circ \boldsymbol{u}_i^{(d)}, \|\boldsymbol{u}_i^{(j)}\| = 1, \ r \in \mathbb{N} \right\}.$$

Definition 2 shows that the tensor nuclear norm is a convex relaxation of tensor rank. However, both the the tensor rank and tensor nuclear norm are NP-hard to compute (Friedland & Lim, 2018). Similarly, we may define a tensor Schatten-$p$ quasi-norm that extends matrix Schatten-$p$ quasi-norm. The $p$-th power of the tensor Schatten-$p$ quasi-norm is a nonconvex relaxation of tensor rank.

**Definition 3.** *The Schatten-$p$ quasi-norm $(0 < p < 1)$ of a tensor $\boldsymbol{\mathcal{X}}$ is defined as*

$$\|\boldsymbol{\mathcal{X}}\|_{S_p} = \inf \left\{ \left( \sum_{i=1}^{r} |s_i|^p \right)^{1/p} : \ \boldsymbol{\mathcal{X}} = \sum_{i=1}^{r} s_i \boldsymbol{u}_i^{(1)} \circ \boldsymbol{u}_i^{(2)} \ldots \circ \boldsymbol{u}_i^{(d)}, \|\boldsymbol{u}_i^{(j)}\| = 1, \ r \in \mathbb{N}, \ 0 < p < 1 \right\}.$$

For convenience, we make the following definition.

**Definition 4.** *Let $d, k \in \mathbb{N}$ and given a set of vectors $\{\boldsymbol{x}_i^{(j)}\}_{i \in [k]}^{j \in [d]}$. Define $\mathcal{CP}_k^d(\boldsymbol{x}_i^{(j)}) := \sum_{i=1}^{k} \boldsymbol{x}_i^{(1)} \circ \boldsymbol{x}_i^{(2)} \cdots \circ \boldsymbol{x}_i^{(d)}$.*

Note that $\mathcal{CP}_k^d(\boldsymbol{x}_i^{(j)})$ is just a shorthand for the sum of the $k$ outer products of $d$ vectors. We provide the following variational form of the tensor Schatten-$p$ (quasi) norm ($p > 0$):

**Theorem 1** (Symmetric Regularizer[1])**.** *The tensor Schatten-$q/d$ (quasi) norm for any real positive $q$ and positive integer $d$ (tensor order) can be represented as a function of the Euclidean norms of the CP components:*

$$\|\boldsymbol{\mathcal{X}}\|_{S_{q/d}}^{q/d} = \inf_{\boldsymbol{\mathcal{X}} = \mathcal{CP}_k^d(\boldsymbol{x}_i^{(j)})} \frac{1}{d} \sum_{i=1}^{k} \sum_{j=1}^{d} \|\boldsymbol{x}_i^{(j)}\|^q.$$

Note that in the theorem $k \geq \mathrm{rank}(\boldsymbol{\mathcal{X}})$ is assumed implicitly via the constraint $\boldsymbol{\mathcal{X}} = \mathcal{CP}_k^d(\boldsymbol{x}_i^{(j)})$. Compared to the definition (Definition 3) of the tensor Schatten-$p$ quasi-norm, $\|\boldsymbol{\mathcal{X}}\|_{S_p}$ given by Theorem 1 internalizes the weight factors $\boldsymbol{s}$ into the vectors $\boldsymbol{u}$ and removes the unit-norm constraints on $\boldsymbol{u}$. This reduces decision variables and the computational cost of calculating the gradient in optimization. In addition, for some choice of $p$, there are no nonsmooth functions on the factors in Theorem 1, while there are always nonsmooth functions in Definition 3. Hence, the formulation of Schatten-$p$ quasi-norm in Theorem 1 is more tractable than that in Definition 3. For instance, in Theorem 1, letting $q = d$, 2, or 1, we have $\|\boldsymbol{\mathcal{X}}\|_* = \inf_{\boldsymbol{\mathcal{X}} = \mathcal{CP}_k^d(\boldsymbol{x}_i^{(j)})} \frac{1}{d} \sum_{i=1}^{k} \sum_{j=1}^{d} \|\boldsymbol{x}_i^{(j)}\|^d$, $\|\boldsymbol{\mathcal{X}}\|_{S_{2/d}}^{2/d} = \inf_{\boldsymbol{\mathcal{X}} = \mathcal{CP}_k^d(\boldsymbol{x}_i^{(j)})} \frac{1}{d} \sum_{i=1}^{k} \sum_{j=1}^{d} \|\boldsymbol{x}_i^{(j)}\|^2$, and $\|\boldsymbol{\mathcal{X}}\|_{S_{1/d}}^{1/d} = \inf_{\boldsymbol{\mathcal{X}} = \mathcal{CP}_k^d(\boldsymbol{x}_i^{(j)})} \frac{1}{d} \sum_{i=1}^{k} \sum_{j=1}^{d} \|\boldsymbol{x}_i^{(j)}\|$, respectively. These three special cases are based on convex functions of Euclidean norms of component vectors, and hence are possibly easier to handle in optimization.

---

[1]It is worth mentioning that the Theorem 2 of (Cheng et al., 2016) and the Proposition 1 of (Lacroix et al., 2018) are two special cases of our Theorem 1 (when $p = 1$ and $d = 3$ or $p = 2/3$ and $d = 3$). The earliest works exploring the $p = 2/3$ and $p = 1/3$ regularizers are (Bazerque et al., 2013) and (Yang et al., 2016) respectively.

According to Theorem 1, for a relatively low-order tensor, when we want to obtain a sharp enough regularizer, we have to use a sufficiently small $p$ for all $i$ and $j$. Thus, every term in $\{\|\boldsymbol{x}_i^{(j)}\|^{pd}\}_{i\in[k]}^{j\in[d]}$ can be nonconvex and nonsmooth, making it difficult to solve the optimization problem of low-rank tensor recovery. The following theorem provides a class of asymmetric regularizers that have fewer nonconvex and nonsmooth terms than those in Theorem 1.

**Theorem 2** (Asymmetric Regularizer). *Suppose $q \in \{1, 1/2, 1/3, 1/4, \ldots\}$. Let $p_1 = q/(1 + qd - q)$ and $p_2 = 2q/(2 + qd - q)$. We have*

$$(a) \ \|\boldsymbol{\mathcal{X}}\|_{S_{p_1}}^{p_1} = \inf_{\boldsymbol{\mathcal{X}}=\mathcal{CP}_k^d(\boldsymbol{x}_i^{(j)})} \ p_1 \sum_{i=1}^{k} \Big( \frac{1}{q}\|\boldsymbol{x}_i^{(1)}\|^q + \sum_{j=2}^{d} \|\boldsymbol{x}_i^{(j)}\| \Big);$$

$$(b) \ \|\boldsymbol{\mathcal{X}}\|_{S_{p_2}}^{p_2} = \inf_{\boldsymbol{\mathcal{X}}=\mathcal{CP}_k^d(\boldsymbol{x}_i^{(j)})} \ p_2 \sum_{i=1}^{k} \Big( \frac{2}{q}\|\boldsymbol{x}_i^{(1)}\|^q + \sum_{j=2}^{d} \|\boldsymbol{x}_i^{(j)}\|^2 \Big).$$

In Theorem 2 (b), the terms of $j \geq 2$ are convex and smooth while those in Theorem 2 (a) are convex but nonsmooth. Therefore, the optimization related to Theorem 2 (b) is easier. Since 3rd-order tensors (e.g. color images and videos) are more prevalent than tensors with other orders, we list their symmetric and asymmetric regularizers with only convex terms in Table 1 for convenience. Note that the last two regularizers in the table are not from Theorem 2. The derivations are in Appendix D.3.

Table 1: Regularizers ($\mathcal{R}(\boldsymbol{\mathcal{X}})$) given by the sum of only convex terms for 3rd-order tensor ($\boldsymbol{\mathcal{X}} = \mathcal{CP}_k^3(\boldsymbol{x})$).

| | $\mathcal{R}(\boldsymbol{\mathcal{X}})$ | Characterization based on Euclidean norm | values of $q, p_1, p_2$ in Theorems 1 and 2 |
|---|---|---|---|
| **Symmetric** | $\|\boldsymbol{\mathcal{X}}\|_*$ | $\frac{1}{3}\sum_{i=1}^{k}\big(\|\boldsymbol{x}_i^{(1)}\|^3 + \|\boldsymbol{x}_i^{(2)}\|^3 + \|\boldsymbol{x}_i^{(3)}\|^3\big)$ | $q = 3$ (Theorem 1) |
| | $\|\boldsymbol{\mathcal{X}}\|_{S_{2/3}}^{2/3}$ | $\frac{1}{3}\sum_{i=1}^{k}\big(\|\boldsymbol{x}_i^{(1)}\|^2 + \|\boldsymbol{x}_i^{(2)}\|^2 + \|\boldsymbol{x}_i^{(3)}\|^2\big)$ | $q = 2$ (Theorem 1) |
| | $\|\boldsymbol{\mathcal{X}}\|_{S_{1/3}}^{1/3}$ | $\frac{1}{3}\sum_{i=1}^{k}\big(\|\boldsymbol{x}_i^{(1)}\| + \|\boldsymbol{x}_i^{(2)}\| + \|\boldsymbol{x}_i^{(3)}\|\big)$ | $q = 1$ (Theorem 1) |
| **Asymmetric** | $\|\boldsymbol{\mathcal{X}}\|_{S_{1/2}}^{1/2}$ | $\frac{\sqrt{2}}{4}\sum_{i=1}^{k}\big(\|\boldsymbol{x}_i^{(1)}\|^2 + \|\boldsymbol{x}_i^{(2)}\|^2 + \|\boldsymbol{x}_i^{(3)}\|\big)$ | $q = 1, p_2 = 1/2$ (Theorem 2) |
| | $\|\boldsymbol{\mathcal{X}}\|_{S_{2/5}}^{2/5}$ | $\frac{16^{1/5}}{5}\sum_{i=1}^{k}\big(\|\boldsymbol{x}_i^{(1)}\|^2 + \|\boldsymbol{x}_i^{(2)}\| + \|\boldsymbol{x}_i^{(3)}\|\big)$ | derived from Appendix D.3 |
| | $\|\boldsymbol{\mathcal{X}}\|_{S_{3/7}}^{3/7}$ | $\frac{81^{1/7}}{7}\sum_{i=1}^{k}\big(\|\boldsymbol{x}_i^{(1)}\|^3 + \|\boldsymbol{x}_i^{(2)}\| + \|\boldsymbol{x}_i^{(3)}\|\big)$ | derived from Appendix D.3 |

In sum, the regularizers we proposed in this section cover all Schatten-$p$ (quasi) norms with any $0 < p \leq 1$ for any-order tensors. Some of the regularizers, especially those asymmetric ones, are based on convex or/and smooth functions of the tensor factors, which are convenient for application and optimization. These regularizers can be applied to LRTC and TRPCA that enjoy a variety of real applications in machine learning and computer vision.

The tensor regularizers we presented in this paper are closely related to the variational forms of matrix nuclear norm and Schatten-$p$ quasi-norms (Shang et al., 2016; 2017; Fan et al., 2019; Giampouras et al., 2020). For instance, the squared sum of Frobenius norms of two factors of a matrix is lower bounded by the nuclear norm of the matrix, which is a special case of our Theorem 1 with $d = 2$ and $p = 1$. In (Fan et al., 2019), the authors provided a class of SVD-free variational forms of the matrix Schatten-$p$ quasi-norm with discrete $p$ that can be arbitrarily small. Our tensor regularizers can be regarded as a generalization of the variational form of Schatten-$p$ quasi-norm from matrix to tensor. For example, in Theorem 1, when $d = 2$ and $p = 1/2$, the regularizer is equivalent to the FGSR-1/2 regularizer of (Fan et al., 2019), which is related to the Schatten-1/2 quasi-norm of matrix. In Theorem 2(b), when $d = 2$ and $q = 1$, the regularizer is equivalent to the FGSR-2/3 regularizer of (Fan et al., 2019), which is related to the Schatten-2/3 quasi-norm of matrix. Nevertheless, theoretical guarantees about these tensor regularizers in low-rank tensor recovery are more difficult to derive than their matrix counterparts.

# 3 Low-Rank Tensor Completion with ENR

## 3.1 LRTC-ENR algorithm

Let $\boldsymbol{\mathcal{X}}^* \in \mathbb{R}^{n_1 \times n_2 \dots \times n_d}$ be a rank-$r$ tensor. Suppose we observed a few noisy entries of $\boldsymbol{\mathcal{X}}^*$ randomly (without replacement):

$$[\boldsymbol{\mathcal{D}}]_{j_1 j_2 \dots j_d} = [\boldsymbol{\mathcal{X}}^*]_{j_1 j_2 \dots j_d} + [\boldsymbol{\mathcal{N}}^*]_{j_1 j_2 \dots j_d}, \quad (j_1, \dots, j_d) \in \Omega$$

where $\Omega$ consists of the locations of the observed entries and each entry of the noise tensor $\boldsymbol{\mathcal{N}}^*$ is drawn from $\mathcal{N}(0, \sigma^2)$. To recover $\boldsymbol{\mathcal{X}}^*$ from $\boldsymbol{\mathcal{D}}$, one may solve

$$\underset{\boldsymbol{\mathcal{X}}}{\text{minimize}} \frac{1}{2} \| \mathcal{P}_\Omega (\boldsymbol{\mathcal{D}} - \boldsymbol{\mathcal{X}}) \|_F^2 + \lambda \| \boldsymbol{\mathcal{X}} \|_{S_p}^p, \tag{1}$$

where $[\mathcal{P}_\Omega(\boldsymbol{\mathcal{Y}})]_{j_1 j_2 \dots j_d} = [\boldsymbol{\mathcal{Y}}]_{j_1 j_2 \dots j_d}$ if $(j_1, \dots, j_d) \in \Omega$ and $[\mathcal{P}_\Omega(\boldsymbol{\mathcal{Y}})]_{j_1 j_2 \dots j_d} = 0$ otherwise. The solution of (1) is an estimate of $\boldsymbol{\mathcal{X}}^*$. However, Problem (1) is intractable because the computation of the Schatten-$p$ quasi-norm is NP-hard. Instead, based on the analysis in Section 2, assuming $k \geq r$, we propose to solve

$$\underset{\{\boldsymbol{x}_i^{(j)}\}}{\text{minimize}} \frac{1}{2} \left\| \mathcal{P}_\Omega \left( \boldsymbol{\mathcal{D}} - \mathcal{CP}_k^d(\boldsymbol{x}_i^{(j)}) \right) \right\|_F^2 + \lambda \mathcal{R} \left( \{\boldsymbol{x}_i^{(j)}\} \right), \tag{2}$$

where $\mathcal{R} \left( \{\boldsymbol{x}_i^{(j)}\} \right)$ denotes a certain regularizer in Theorem 1, Theorem 2, or Table 1, e.g.,

$$\mathcal{R} \left( \{\boldsymbol{x}_i^{(j)}\} \right) = \frac{1}{d} \sum_{i=1}^{k} \sum_{j=1}^{d} \| \boldsymbol{x}_i^{(j)} \|^q = \frac{1}{d} \sum_{j=1}^{d} \| \boldsymbol{X}^{(j)} \|_{2,q}^q, \tag{3}$$

where $\boldsymbol{X}^{(j)} = [\boldsymbol{x}_1^{(j)}, \dots, \boldsymbol{x}_k^{(j)}]$. Similarly, the regularizers in Theorem 2 (a) and (b) can be formulated as $p_1(\frac{1}{q}\|\boldsymbol{X}^{(1)}\|_{2,q}^q + \sum_{j=2}^{d} \|\boldsymbol{X}^{(j)}\|_{2,1})$ and $p_2(\frac{2}{q}\|\boldsymbol{X}^{(1)}\|_{2,q}^q + \sum_{j=2}^{d} \|\boldsymbol{X}^{(j)}\|_F^2)$ respectively. Note that even finding a stationary point of (1) is hard due to the computation of the Schatten-$p$ quasi-norm while finding a stationary point of (2) is computationally feasible. For convenience, we call problem (2) Low-Rank Tensor Completion with Euclidean Norm Regularization (**LRTC-ENR**). LRTC-ENR is an LRTC framework that covers all the regularizers presented in Theorem 1, Theorem 2, and Table 1, where the $p = 1, 2/3, 1/3$ regularizers for 3rd-order tensors studied by (Bazerque et al., 2013; Cheng et al., 2016; Yang et al., 2016; Shi et al., 2017; Lacroix et al., 2018) are special cases.

We propose to solve (2) by Block Coordinate Descent (BCD) with Extrapolation (BCDE for short) (Xu & Yin, 2013), which is usually more efficient than BCD alone in practice. The decision variables can be organized into $d$ blocks ($d$ matrices, i.e., $\boldsymbol{X}^{(j)}$, $j = 1, \dots, d$). Then we need to iteratively solve $d$ subproblems related to the $\ell_q$ norm minimization. When $q < 1$, we integrate BCDE with iteratively reweighted method (Lu, 2014). If we use the symmetric regularizers given by Theorem 1 with $q \leq 1$, we need to solve $d$ nonsmooth subproblems (related to $\|\boldsymbol{X}^{(j)}\|_{2,q}$, $j = 1, \dots, d$) in every iteration and when $q < 1$ we need to perform the iteratively reweighted method for $\|\boldsymbol{X}^{(j)}\|_{2,q}$, $j = 1, \dots, d$. In contrast, if we use the asymmetric regularizers given by Theorem 2 (b), the number of nonsmooth subproblems (related to $\|\boldsymbol{X}^{(1)}\|_{2,q}$) is reduced to 1 and when $q < 1$ we only need to perform iteratively reweighted method for $\|\boldsymbol{X}^{(1)}\|_{2,q}$. Therefore, the optimization related to the asymmetric regularizers is more efficient than that related to the symmetric regularizers for the same problem.

We may also use quasi-Newton methods such as L-BFGS (Liu & Nocedal, 1989) to solve problem (2) even when the objective function is nonsmooth. The details of the optimization for (2) are in Appendix A.

It is worth mentioning that according to Definition 3, one may consider the following LRTC problem

$$\begin{aligned} \underset{\{\boldsymbol{x}_i^{(j)}\}, \boldsymbol{s}}{\text{minimize}} \quad & \frac{1}{2} \left\| \mathcal{P}_\Omega \left( \boldsymbol{\mathcal{D}} - \widetilde{\mathcal{CP}}_k^d(\{\boldsymbol{x}_i^{(j)}\}, \boldsymbol{s}) \right) \right\|_F^2 + \lambda \sum_{i=1}^{k} |s_i|^p, \\ \text{subject to} \quad & \|\boldsymbol{x}_i^{(j)}\| = 1, \forall i \in [k], j \in [d]. \end{aligned} \tag{LRTC-Schatten-$p$}$$

where $\widetilde{\mathcal{CP}}_k^d(\{\boldsymbol{x}_i^{(j)}\}, \boldsymbol{s}) = \sum_{i=1}^k s_i \boldsymbol{x}_i^{(1)} \circ \boldsymbol{x}_i^{(2)} \ldots \circ \boldsymbol{x}_i^{(d)}$. Nevertheless, it is more difficult to solve LRTC-Schatten-$p$ than LRTC-ENR due to the following reasons. First, LRTC-Schatten-$p$ has one more block of variables $\boldsymbol{s}$, of which the gradient computation is costly (it requires evaluating every rank-one component of $\boldsymbol{\mathcal{X}}$). Second, constrained optimization LRTC-Schatten-$p$ is generally harder than unconstrained optimization. Heuristically, we suggest using BCDE with iteratively reweighted minimization embedded to solve LRTC-Schatten-$p$. It is compared with LRTC-ENR in Figure 3 of Section 5.1.1.

## 3.2 Generalization Error Bound of LRTC

In this section, we study the generalization error bounds of LRTC-Schatten-$p$ and LRTC-ENR. The generalization error of tensor completion is defined as the difference between the prediction error for all entries and the training error for the observed entries, i.e.,

$$\mathcal{GE}_{\text{TC}} := \mathcal{E}(\boldsymbol{\mathcal{D}}, \boldsymbol{\mathcal{X}}, [n] \times \cdots \times [n]) - \mathcal{E}(\boldsymbol{\mathcal{D}}, \boldsymbol{\mathcal{X}}, \Omega).$$

In this study, we let $\mathcal{E}(\boldsymbol{\mathcal{D}}, \boldsymbol{\mathcal{X}}, [n] \times \cdots \times [n]) = \frac{1}{\sqrt{n^d}}\|\boldsymbol{\mathcal{D}} - \boldsymbol{\mathcal{X}}\|_F$ and $\mathcal{E}(\boldsymbol{\mathcal{D}}, \boldsymbol{\mathcal{X}}, \Omega) = \frac{1}{\sqrt{|\Omega|}}\|\mathcal{P}_\Omega(\boldsymbol{\mathcal{D}} - \boldsymbol{\mathcal{X}})\|_F$, though their squares can also be used. It should be pointed out that, compared to LRMC, it is much more difficult to analyze the generalization ability of LRTC. The main reason is that we may not have orthogonal factors in a CP decomposition. Thus, we first consider the case of orthogonal CP decomposition. We will also provide generalization error bounds for the general cases (without the restriction of orthogonality).

Without loss of generality, we consider the case of hyper-cubic tensors, denoted by $\boldsymbol{\mathcal{X}} \in \mathbb{R}^{n^{\otimes d}}$. For convenience, we define the following two tensor sets.

**Definition 5** (Orthogonal CP tensor sets). *Let $\mathcal{S}_{d,n}^\perp$ be the set of hyper-cubic tensors with orthogonal CP factors. Denote the Schatten-$p$ quasi-norm of $\boldsymbol{\mathcal{X}} \in \mathcal{S}_{d,n}^\perp$ by $\|\boldsymbol{\mathcal{X}}\|_{S_p^\perp}$. Let $\mathcal{S}_{d,n,p}^\perp$ be the set of tensors in $\mathcal{S}_{d,n}^\perp$ and with bounded (by $\psi_p$) Schatten-$p$ quasi norm. To be more precise,*

*(a)* $\mathcal{S}_{d,n}^\perp := \Big\{\boldsymbol{\mathcal{X}} \in \mathbb{R}^{n^{\otimes d}} : \ \boldsymbol{\mathcal{X}} = \sum_{i=1}^r s_i \boldsymbol{u}_i^{(1)} \circ \boldsymbol{u}_i^{(2)} \ldots \circ \boldsymbol{u}_i^{(d)}; \forall j \in [d], \boldsymbol{u}_i^{(j)^\top} \boldsymbol{u}_l^{(j)} = 1 \text{ if } i = l, \boldsymbol{u}_i^{(j)^\top} \boldsymbol{u}_l^{(j)} = 0 \text{ if } i \neq l\Big\}$,

*(b)* $\mathcal{S}_{d,n,p}^\perp := \{\boldsymbol{\mathcal{X}} \in \mathcal{S}_{d,n}^\perp : \ \|\boldsymbol{\mathcal{X}}\|_{S_p^\perp} \leq \psi_p\}$.

We have the following covering number result.

**Theorem 3.** *The covering numbers (denoted by $\mathcal{N}$) of $\mathcal{S}_{d,n,p}^\perp$ with respect to the Frobenius norm satisfy*

$$\log \mathcal{N}(\mathcal{S}_{d,n,p}^\perp, \|\cdot\|_F, \epsilon) \leq \left(\frac{1}{2} + \frac{1}{p}\right) nd \left(\log(d+1)\right) \left(\frac{c\psi_p}{\epsilon}\right)^{\frac{2p}{2-p}},$$

*where $c > 0$ is a universal constant and $0 < p < 2$.*

Note that although Theorem 3 does not include the case $p = 2$, it is much easier to obtain the covering number of $\mathcal{S}_{d,n,2}^\perp$, which will not be detailed in this paper because $\|\boldsymbol{\mathcal{X}}\|_{S_2^\perp}$ is useless in regularizing the tensor rank.

Based on Theorem 3, we derive the following generalization error bound for the Schatten-$p$ quasi-norm regularized orthogonal tensor completion (minimize$_{\boldsymbol{\mathcal{X}} \in \mathcal{S}_{d,n}^\perp} \frac{1}{2}\|\mathcal{P}_\Omega(\boldsymbol{\mathcal{D}} - \boldsymbol{\mathcal{X}})\|_F^2 + \lambda\|\boldsymbol{\mathcal{X}}\|_{S_p^\perp}^p$).

**Theorem 4.** *Suppose $\boldsymbol{\mathcal{D}} \in \mathbb{R}^{n^{\otimes d}}$, $\boldsymbol{\mathcal{X}} \in \mathcal{S}_{d,n}^\perp$, $\max\{\|\boldsymbol{\mathcal{D}}\|_\infty, \|\boldsymbol{\mathcal{X}}\|_\infty\} \leq \varepsilon$, and $0 < p < 2$. Then there exists a numerical constant $c$ such that with probability at least $1 - 2n^{-d}$,*

$$\frac{1}{\sqrt{n^d}}\|\boldsymbol{\mathcal{D}} - \boldsymbol{\mathcal{X}}\|_F - \frac{1}{\sqrt{|\Omega|}}\|\mathcal{P}_\Omega(\boldsymbol{\mathcal{D}} - \boldsymbol{\mathcal{X}})\|_F \leq c\varepsilon \left(\frac{(\frac{1}{2} + \frac{1}{p})nd\log(d+1)}{|\Omega|} \left(\frac{\|\boldsymbol{\mathcal{X}}\|_{S_p^\perp}}{\varepsilon\sqrt{dn}}\right)^{\frac{2p}{2-p}}\right)^{1/4}. \tag{4}$$

In the theorem, since $\|\mathcal{X}\|_{S_p^{\perp}}/(\varepsilon\sqrt{dn}) > 1$, the bound is a U-shape function of $p$ when $0 < p < 2$. Therefore, there exists a threshold $\bar{p} = \operatorname{argmin}_{0<p<2}(1/2 + 1/p)\big(\|\mathcal{X}\|_{S_p^{\perp}}/(\varepsilon\sqrt{dn})\big)^{2p/(2-p)}$ such that when $\bar{p} \leq p < 2$, a smaller $p$ will lead to a tighter error bound. However, it is difficult to obtain $\bar{p}$ analytically. Here we use a toy example to illustrate the result of Theorem 4. According to Definition 3, we generate $\boldsymbol{s}$ as $\boldsymbol{s} = [2^{\nu}, 1.9^{\nu}, 1.8^{\nu}, \ldots, 0.2^{\nu}, 0.1^{\nu}]$ and normalize it by $\boldsymbol{s} \leftarrow \boldsymbol{s}/\sum_i s_i$. We choose $\nu$ from $\{0.1, 1, 5, 10, 20, 50\}$ and generate six different $\boldsymbol{s}$. We see that a larger $\nu$ leads to a higher decay rate of the element of $\boldsymbol{s}$. We use these six $\boldsymbol{s}$ together with three random orthogonal matrices $\boldsymbol{U}$ of size $100 \times 20$ to form six low-rank tensors of size $100 \times 100 \times 100$ and rank 20. These six tensors have different "low-rankness" because of the different $\nu$. For instance, the tensor corresponding to $\nu = 50$ can be well approximated by a rank-4 tensor, where the relative error is less than 0.01%. We let $|\Omega| = 0.1n^3$ and compute $\big((\frac{1}{2} + \frac{1}{p})nd\log(d + 1)|\Omega|^{-1}(\|\mathcal{X}\|_{S_p^{\perp}}/(\varepsilon\sqrt{dn}))^{\frac{2p}{2-p}}\big)^{1/4} \triangleq \Delta$ for each tensor. The six $\boldsymbol{s}$ and average $\Delta$ (corresponding to each $\boldsymbol{s}$) of 100 repeated trials are shown in Figure 1. We see that in each case, a smaller $p$ but not too small will lead to a smaller $\Delta$. When the decay rate is higher, the approximate rank is lower, which leads to a smaller $\Delta$. On the other hand, when the decay rate is lower, reducing $p$ is more useful for reducing the error bound $\Delta$.

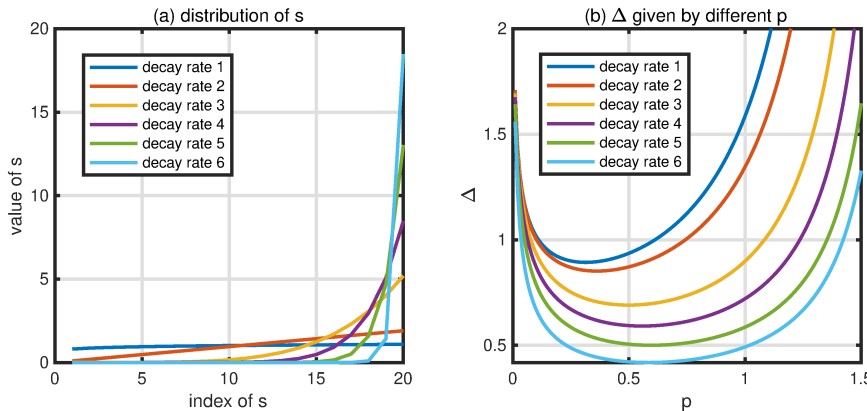

Figure 1: An intuitive example of the error bound of Theorem 4. The decay rates $1, \ldots, 6$ correspond to $\nu = 0.1, \ldots, 50$ respectively.

Now we study the generalization error of LRTC-ENR (2). For convenience, let $\|\boldsymbol{X}^{(j)}\|_{2,q} := \big(\sum_i \|\boldsymbol{x}_i^{(j)}\|^q\big)^{1/q}$ and define

$$\mathcal{S}_{d,n,q}^k = \Big\{\mathcal{X} \in \mathbb{R}^{n^{\otimes d}} : \ \mathcal{X} = \mathcal{CP}_k^d(\boldsymbol{x}_i^{(j)}); \|\boldsymbol{X}^{(j)}\|_{2,q} \leq \alpha_q^{(j)}, \ \|\boldsymbol{X}^{(j)}\|_{op} \leq \gamma_j, \ j \in [d]\Big\}, \tag{5}$$

where $\boldsymbol{X}^{(j)} = [\boldsymbol{x}_1^{(j)}, \boldsymbol{x}_2^{(j)}, \ldots, \boldsymbol{x}_k^{(j)}]$. We have

**Theorem 5.** *The covering numbers of $\mathcal{S}_{d,n,q}^k$ with respect to the Frobenius norm satisfy*

$$\log\mathcal{N}(\mathcal{S}_{d,n,q}^k, \|\cdot\|_F, \epsilon) \leq \frac{c(n + \log(ek))}{q}\sum_{j=1}^d \left(\frac{d\phi\alpha_q^{(j)}}{\gamma_j\epsilon}\right)^{\frac{2q}{2-q}},$$

*for any $0 < q < 2$, and*

$$\log\mathcal{N}(\mathcal{S}_{d,n,q}^k, \|\cdot\|_F, \epsilon) \leq c'nk^{2(q-1)/q}\log(2nk)\sum_{j=1}^d \left(\frac{d\phi\alpha_q^{(j)}}{\gamma_j\epsilon}\right)^2$$

*for any $q \geq 2$, where $\phi = \prod_{j=1}^d \gamma_j$ and $c$, $c'$ are universal constants.*

Based on Theorem 5, we have the following generalization error bound for LRTC-ENR (2) with $\mathcal{R}(\{\boldsymbol{x}_i^{(j)}\}) = \sum_{i=1}^k \sum_{j=1}^d \|\boldsymbol{x}_i^{(j)}\|^q$.

**Theorem 6.** *Suppose $\boldsymbol{\mathcal{D}} \in \mathbb{R}^{n^{\otimes d}}$, $\boldsymbol{\mathcal{X}} \in \mathcal{S}^{k}_{d,n,q}$, $\phi = \prod_{j=1}^{d} \gamma_j$, and $\max\{\|\boldsymbol{\mathcal{D}}\|_\infty, \|\boldsymbol{\mathcal{X}}\|_\infty\} \leq \varepsilon$. Denote $\bar{\Omega}$ the index set of the missing entries of $\boldsymbol{\mathcal{D}}$, suppose $|\bar{\Omega}| > |\Omega|$, and let $\kappa = \frac{(|\Omega| + |\bar{\Omega}|)^2}{|\Omega||\bar{\Omega}|}$. Then with probability at least $1 - \delta$ over the random partition $\Omega$ and $\bar{\Omega}$, we have*

$$\frac{1}{|\bar{\Omega}|}\|\mathcal{P}_{\bar{\Omega}}(\boldsymbol{\mathcal{D}} - \boldsymbol{\mathcal{X}})\|_F^2 \leq \frac{1}{|\Omega|}\|\mathcal{P}_{\Omega}(\boldsymbol{\mathcal{D}} - \boldsymbol{\mathcal{X}})\|_F^2 + \varepsilon^2(B_{\mathcal{R}} + B_\delta)$$

*where $B_\delta = \dfrac{44n^d}{\sqrt{|\Omega||\bar{\Omega}|}} + 12\sqrt{\dfrac{n^d}{|\Omega||\bar{\Omega}|}\log\dfrac{1}{\delta}}$ and*

$$B_{\mathcal{R}} = \begin{cases} \dfrac{c_1\kappa}{|\Omega|}\sqrt{nk^{2(q-1)/q}\log(2nk)\sum_{j=1}^{d}\left(\dfrac{d\phi\alpha_q^{(j)}}{\varepsilon\gamma_j}\right)^2 \log|\Omega|} & \text{if } q \geq 1 \\[3ex] \dfrac{c_2\kappa}{|\Omega|}\sqrt{\dfrac{(n + \log(ek))(2-q)^2}{q(2-2q)^2}\sum_{j=1}^{d}\left(\dfrac{d\phi\alpha_q^{(j)}}{\varepsilon\gamma_j}\right)^{\frac{2q}{2-q}}|\Omega|^{\frac{1-q}{4-2q}}} & \text{if } 0 < q < 1 \end{cases}$$

*with constants $c_1$ and $c_2$.*

We see that in Theorem 6, the bound is not continuous around $q = 1$. In the theorem when $0 < q < 1$, the bound is a U-shape function of $q$. It indicates that a smaller $q$ but not too small leads to a tighter error bound and for a smaller $|\Omega|$ reducing the value of $q$ becomes more useful. We show an intuitive example in Figure 2 to illustrate the role of $q$ and $|\Omega|$ in the error bound of Theorem 6. In the example, we use $d = 3$, $n = 100$, and $k = 20$ to generate synthetic random tensors $\boldsymbol{\mathcal{X}} = \sum_{i=1}^{20} \boldsymbol{x}_i^{(1)} \circ \boldsymbol{x}_i^{(2)} \cdots \circ \boldsymbol{x}_i^{(3)}$, where $\boldsymbol{x}_i^{(1)}, \cdots, \boldsymbol{x}_i^{(3)}$ are drawn from Gaussian distribution and $\|\boldsymbol{x}_i^{(1)}\| = \cdots = \|\boldsymbol{x}_i^{(3)}\| = \delta_i$. We consider six different decay rates for $\delta_1, \ldots, \delta_{20}$ corresponding to different levels of low-rankness. In general, a larger decay rate means an easier tensor recovery problem. We let $\rho = \frac{|\Omega|}{n^d}$ and

$$\Delta = \frac{1}{|\Omega|}\sqrt{\frac{(n + \log(ek))(2-q)^2}{q(2-2q)^2}\sum_{j=1}^{d}\left(\frac{d\phi\alpha_q^{(j)}}{\varepsilon\gamma_j}\right)^{\frac{2q}{2-q}}|\Omega|^{\frac{1-q}{4-2q}}}, \tag{6}$$

where $0 < q < 1$. In the sub-figures (b,c,d), we see that a smaller $q$ but not too small provides a tighter error bound. When the decay rate is smaller, reducing the value of $q$ becomes more useful, which is consistent with the result of Figure 1. In addition, when $\rho$ is smaller, namely the problem is harder, reducing the value of $q$ becomes much more effective.

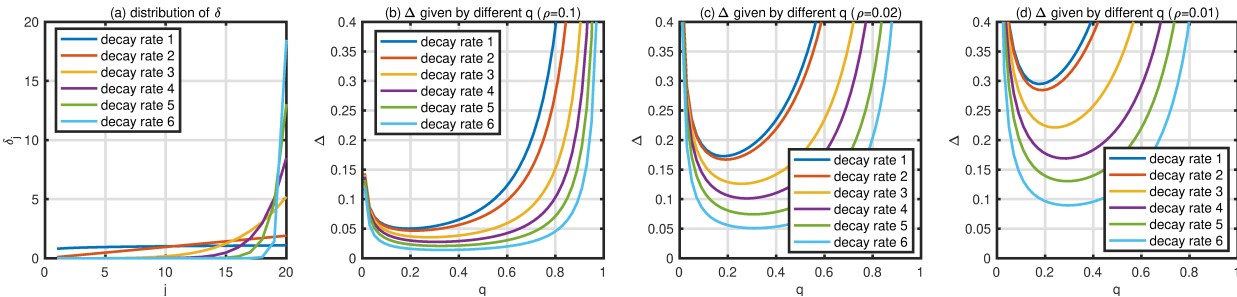

Figure 2: An intuitive exxample for the error bound of Theorem 6.

Based on Theorem 6, we can obtain the generalization error bound for (1), which is shown in the following corollary.

**Corollary 1.** *Suppose* $\boldsymbol{\mathcal{D}} \in \mathbb{R}^{n^{\otimes d}}$, $\boldsymbol{\mathcal{X}} \in \mathbb{R}^{n^{\otimes d}}$, $\max\{\|\boldsymbol{\mathcal{D}}\|_\infty, \|\boldsymbol{\mathcal{X}}\|_\infty\} \leq \varepsilon$, $\|\boldsymbol{\mathcal{X}}\|_{S_p}$ *is attained at* $\boldsymbol{\mathcal{X}} = \mathcal{CP}_k^d(\boldsymbol{x}_i^{(j)})$, *and* $\max_{j \in [d]} \|\boldsymbol{X}^{(j)}\|_{op} \leq \bar{\gamma}$. *Denote* $\bar{\Omega}$ *the index set of the missing entries of* $\boldsymbol{\mathcal{D}}$, *suppose* $|\bar{\Omega}| > |\Omega|$, *and let* $\kappa = \frac{(|\Omega| + |\bar{\Omega}|)^2}{|\Omega||\bar{\Omega}|}$. *Then with probability at least* $1 - \delta$ *over the random sampling of* $\Omega$, *we have*

$$\frac{1}{|\bar{\Omega}|}\|\mathcal{P}_{\bar{\Omega}}(\boldsymbol{\mathcal{D}} - \boldsymbol{\mathcal{X}})\|_F^2 \leq \frac{1}{|\Omega|}\|\mathcal{P}_\Omega(\boldsymbol{\mathcal{D}} - \boldsymbol{\mathcal{X}})\|_F^2 + \varepsilon^2(B_{\mathcal{R}} + B_\delta)$$

*where* $B_\delta = \dfrac{44n^d}{\sqrt{|\Omega||\bar{\Omega}|}} + 12\sqrt{\dfrac{n^d}{|\Omega||\bar{\Omega}|}\log\dfrac{1}{\delta}}$ *and*

$$B_{\mathcal{R}} = \begin{cases} \dfrac{c_1\kappa}{|\Omega|}\sqrt{nk^{2(pd-1)/pd}\log(2nk)d\left(\dfrac{d\bar{\gamma}^{d-1}}{\varepsilon}\right)^2\|\boldsymbol{\mathcal{X}}\|_{S_p}^{2/d}\log|\Omega|} & \text{if } p \geq 1/d \\[3ex] \dfrac{c_2\kappa}{|\Omega|}\sqrt{\dfrac{(n+\log(ek))(2-pd)^2}{p(2-2pd)^2}\left(\dfrac{d\bar{\gamma}^{d-1}}{\varepsilon}\right)^{2pd/(2-pd)}\|\boldsymbol{\mathcal{X}}\|_{S_p}^{2p/(2-pd)}|\Omega|^{\frac{1-pd}{4-2pd}}} & \text{if } 0 < p < 1/d \end{cases}$$

*with constants* $c_1$ *and* $c_2$.

In Corollary 1 as well as Theorem 6, the error bounds are for $\frac{1}{|\bar{\Omega}|}\|\mathcal{P}_{\bar{\Omega}}(\boldsymbol{\mathcal{D}} - \boldsymbol{\mathcal{X}})\|_F^2$ and are almost linear with $\frac{1}{|\Omega|}$. In contrast, in Theorem 4, the error bound is for $\frac{1}{\sqrt{n^d}}\|\boldsymbol{\mathcal{D}} - \boldsymbol{\mathcal{X}}\|_F$ and is linear with $\left(\frac{1}{|\Omega|}\right)^{1/4}$. Therefore, the bounds of Corollary 1 and Theorem 6 are tighter than that of Theorem 4, however, at a price of breaking the continuity at $q = 1$ or $p = 1/d$.

Consider the terms related to $p$ in $B_{\mathcal{R}}$ of Corollary 1 with $p \geq 1/d$, i.e., $k^{2(pd-1)/pd}\|\boldsymbol{\mathcal{X}}\|_{S_p}^{2/d} \triangleq \pi_p$. Let $1/d \leq p_2 < p_1$. Using the power-mean inequality[2], we have $k^{-1/p_2}\|\boldsymbol{\mathcal{X}}\|_{S_{p_2}} \leq k^{-1/p_1}\|\boldsymbol{\mathcal{X}}\|_{S_{p_1}}$. It follows that $\pi_{p_2} = k^{2(p_2 d-1)/p_2 d}\|\boldsymbol{\mathcal{X}}\|_{S_{p_2}}^{2/d} \leq k^{2(p_2 d-1)/p_2 d}(k^{1/p_2 - 1/p_1})^{2/d}\|\boldsymbol{\mathcal{X}}\|_{S_{p_1}}^{2/d} = \pi_{p_1}$, where the equality holds if and only if $s_1 = s_2 = \ldots = s_k$ (it will not happen in practice almost surely). Therefore, when $1/d \leq p_2 < p_1$, the error bound given by $p_2$ is always tighter than that given by $p_1$. This provides a rule of thumb for real applications that $p$ should be at most $1/d$. Together with the result of Corollary 1 for $0 < p < 1/d$, we suggest using $p = 1/2d$.

**Comparison with previous work** Barak & Moitra (2016), Foster & Risteski (2019) and Wimalawarne & Mamitsuka (2021) have also studied the generalization error of CP based tensor completion. Their bounds are based on the nuclear norm (or Schatten-1 norm equivalently). We compare their bounds with our bound in Corollary 1, in terms of third-order tensor with unit nuclear norm and $r$ rank. The generalization error bound given by (Barak & Moitra, 2016) scales as $O\left(rn^{3/2}\log^4 n/|\Omega|\right)$. Foster & Risteski (2019) provided an empirical risk bound for orthogonal tensor completion that scales as $O\left(r^2 n^{3/2}\log^6 n/|\Omega|\right)$. The generalization error bound (Theorem 1(b)) given by (Wimalawarne & Mamitsuka, 2021) is $O(rn^{3/2}s_{\max}/|\Omega|)$, where $1/r < s_{\max} < 1$. In our Corollary 1, let $d = 3$, $p = 1$, $k = r$, $\bar{\gamma} = 1$, and $\|\boldsymbol{\mathcal{X}}\|_{S_p} = 1$, our bound becomes $O(r^2\sqrt{n\log(2nr)}\log|\Omega|/|\Omega|)$. We see our bound is tighter than all others if $r\sqrt{\log(2nr)}\log|\Omega| \leq ns_{\max}$.

## 4 Tensor Robust PCA with ENR

### 4.1 TRPCA-ENR algorithm

Let $\boldsymbol{\mathcal{X}}^* \in \mathbb{R}^{n_1 \times n_2 \times \cdots \times n_d}$ be a rank-$r$ tensor. Suppose $\boldsymbol{\mathcal{X}}^*$ is corrupted as

$$\boldsymbol{\mathcal{D}} = \boldsymbol{\mathcal{X}}^* + \boldsymbol{\mathcal{N}}^* + \boldsymbol{\mathcal{E}}^*, \tag{7}$$

---

[2]For any $z_1, z_2, \ldots, z_n \geq 0$, if $\infty > \alpha > \beta > 0$, we have $\left(\frac{1}{n}\sum_{i=1}^n z_i^\alpha\right)^{\frac{1}{\alpha}} \geq \left(\frac{1}{n}\sum_{i=1}^n z_i^\beta\right)^{\frac{1}{\beta}}$ with equality if and only if $z_1 = z_2 = \ldots = z_n$.

where $\boldsymbol{\mathcal{N}}^*$ is a dense noise tensor drawn from $\mathcal{N}(0, \sigma^2)$ and $\boldsymbol{\mathcal{E}}^*$ is a sparse noise tensor with randomly distributed nonzero entries. To recover $\boldsymbol{\mathcal{X}}^*$ from $\boldsymbol{\mathcal{D}}$, we wish to solve

$$\underset{\boldsymbol{\mathcal{X}}, \boldsymbol{\mathcal{E}}}{\text{minimize}} \, \frac{1}{2} \|\boldsymbol{\mathcal{D}} - \boldsymbol{\mathcal{X}} - \boldsymbol{\mathcal{E}}\|_F^2 + \lambda_x \|\boldsymbol{\mathcal{X}}\|_{S_p}^p + \lambda_e \|\boldsymbol{\mathcal{E}}\|_1, \tag{8}$$

but the problem is intractable. Then, based on the analysis in Section 2, we propose to solve

$$\underset{\{\boldsymbol{x}_i^{(j)}\}, \, \boldsymbol{\mathcal{E}}}{\text{minimize}} \, \frac{1}{2} \left\| \boldsymbol{\mathcal{D}} - \sum_{i=1}^k \boldsymbol{x}_i^{(1)} \circ \boldsymbol{x}_i^{(2)} \dots \circ \boldsymbol{x}_i^{(d)} - \boldsymbol{\mathcal{E}} \right\|_F^2 + \lambda_x \mathcal{R}\left(\{\boldsymbol{x}_i^{(j)}\}\right) + \lambda_e \|\boldsymbol{\mathcal{E}}\|_1, \tag{9}$$

where $\lambda_e$ is a penalty parameter for the sparse tensor $\boldsymbol{\mathcal{E}}$. For convenience, we call (9) **TRPCA-ENR**. Similar to LRTC-ENR, TRPCA-ENR is a TRPCA framework that covers all the regularizers presented in Theorem 1, Theorem 2, and Table 1, where the $p = 1, 2/3, 1/3$ regularizers for 3rd-order tensors studied by (Bazerque et al., 2013; Cheng et al., 2016; Yang et al., 2016; Shi et al., 2017; Lacroix et al., 2018) are special cases. Regarding problem (9), we proposed to update the $d + 1$ blocks of decision variables (i.e. $\boldsymbol{\mathcal{E}}$ and $\boldsymbol{X}^{(j)} = [\boldsymbol{x}_1^{(j)}, \dots, \boldsymbol{x}_k^{(j)}]$, $j = 1, \dots, d$) alternately. Namely, we need to iteratively solve $d + 1$ subproblems related to the $\ell_q$ $(0 < q \le 2)$ norm minimization. Similar to LRTC-ENR, we use the iteratively reweighted method (Lu, 2014) to solve the subproblems associated with $q < 1$. Particularly, if we use the symmetric regularizers given by Theorem 1, when $q = 2$, we use alternating minimization to solve (9) because every subproblem has a closed-form solution; when $q \le 1$, there are $d$ subproblems having no closed-form solution. If we use the asymmetric regularizers given by Theorem 2 (b), there is only one subproblem having no closed-form solution and hence we propose to solve (9) by the (nonconvex) alternating direction method of multipliers (ADMM) (Wang et al., 2015). We see that the asymmetric regularizers lead to much easier optimization problems than the symmetric counterparts do. The optimization for (9) is detailed in Appendix B.

## 4.2 Recovery Error Bound of TRPCA

Currently, it is difficult to provide recovery guarantees for problem (8) and problem (9) because of the non-convexity and non-smoothness. Instead, we consider the following constrained orthogonal tensor recovery problem

$$\underset{\boldsymbol{\mathcal{X}} \in \mathcal{S}_{d,n}^{\perp}, \boldsymbol{\mathcal{E}}}{\text{minimize}} \, \|\boldsymbol{\mathcal{D}} - \boldsymbol{\mathcal{X}} - \boldsymbol{\mathcal{E}}\|_F^2$$
$$\text{subject to } \|\boldsymbol{\mathcal{X}}\|_{S_p^{\perp}}^p \le R_x^p, \, \|\boldsymbol{\mathcal{E}}\|_{p'}^{p'} \le R_e^{p'}, \tag{10}$$

where $p, p' \in (0, 1]$, the orthogonal CP tensor set $\mathcal{S}_{d,n}^{\perp}$ was defined by Definition 5, and the observed tensor $\boldsymbol{\mathcal{D}}$ was generated by (7) with $n_1 = n_2 \cdots = n_d = n$. In terms of the regularization on $\boldsymbol{\mathcal{E}}$, problem (10) is more general than (8) and (9) because here we consider the $\ell_{p'}$ regularizer. However, compared to (8) and (9), we have to consider the orthogonal tensor set for simplicity. We provide the following bound of recovery error.

**Theorem 7.** *Let* $p, p' \in (0, 1]$, $\|\boldsymbol{\mathcal{X}}^*\|_{S_p^{\perp}}^p \le R_x^p$, *and* $\|\boldsymbol{\mathcal{E}}^*\|_{p'}^{p'} \le R_e^{p'}$. *Suppose* $\hat{\boldsymbol{\mathcal{X}}}, \hat{\boldsymbol{\mathcal{E}}}$ *is the optimal solution of* (10) *and* $\max(\|\boldsymbol{\mathcal{X}}^*\|_\infty, \|\hat{\boldsymbol{\mathcal{X}}}\|_\infty) \le \alpha$. *Then with probability at least* $1 - 4n^{-d}$,

$$\|\boldsymbol{\mathcal{X}}^* - \hat{\boldsymbol{\mathcal{X}}}\|_F^2 + \|\boldsymbol{\mathcal{E}}^* - \hat{\boldsymbol{\mathcal{E}}}\|_F^2$$
$$\le (24 + 16\sqrt{2}) \left( \left(2\sqrt{2}\sigma\sqrt{dn\log(5d) + d\log(n)}\right)^{2-p} R_x^p + \left(2\alpha + 2\sigma\sqrt{d\log(n)}\right)^{2-p'} R_e^{p'} \right).$$

In this theorem, for convenience, let $c_1 = 2\sqrt{2}\sigma\sqrt{dn\log(5d) + d\log(n)}$ and $c_2 = 2\alpha + 2\sigma\sqrt{d\log(n)}$. When $p$ and $p'$ decrease, $c_1^{2-p}$ and $c_2^{2-p'}$ increase but $R_x^p$ and $R_w^{p'}$ decrease, provided that there are a few elements in $\boldsymbol{s}$ (refer to Definition 3) and elements in $\boldsymbol{\mathcal{E}}^*$ larger than 1. Therefore, when $c_1$ and $c_2$ are not too large,

reducing the value of $p$ or $p'$ will lead a tighter recovery error bound for $\boldsymbol{\mathcal{X}}^*$ and $\boldsymbol{\mathcal{E}}^*$. Although Theorem 7 is for (10) rather than (8) and (9), it verified the superiority of smaller $p$ of the tensor Schatten-$p$ quasi-norm in TRPCA.

## 5 Numerical Results

In this section, we conduct experiments of LRTC and TRPCA on both synthetic data and real image data. All the tensor computations are implemented in the MATLAB Tensor Tool Box of (Bader et al., 2019). We use a MacBook Pro with 2.6 GHz Intel Core (i7) and 16 GB RAM. It is worth noting that the main goal of the experiments is to show the effectiveness of the symmetric and asymmetric regularizers in LRTC and TRPCA and the influence of $p$ and $q$ on the recovery performance, not to compare our optimization strategies with previous works (Bazerque et al., 2013; Cheng et al., 2016; Yang et al., 2016; Shi et al., 2017; Lacroix et al., 2018). In most cases, $q = 1$ (or $p = 1/3$) (Yang et al., 2016) on the 3rd-order tensors has the best performance. The asymmetric regularizers slightly outperform the symmetric regularizers in the experiments of TRPCA. In addition, the experiments show that ENRs, though simple, can outperform a few strong baselines of LRTC and TRPCA.

### 5.1 Experiments of LRTC

We compare LRTC-ENR with a few baselines on synthetic low-rank tensors and multi-spectral images. We use Relative recovery error $\|\mathcal{P}_{\bar{\Omega}}(\boldsymbol{\mathcal{T}} - \hat{\boldsymbol{\mathcal{T}}})\|_F / \|\mathcal{P}_{\bar{\Omega}}(\boldsymbol{\mathcal{T}})\|_F$ to evaluate the tensor completion performance, where $\boldsymbol{\mathcal{T}}$ is the original complete tensor and $\hat{\boldsymbol{\mathcal{T}}}$ denotes the tensor recovered by a tensor completion method. The minimum value of the metric is zero. A meaningful value of the metric is in the range of $[0, 1]$.

#### 5.1.1 Synthetic data

With the size for each dimension $n = 50$, we generate noisy low-rank synthetic tensors $\boldsymbol{\mathcal{D}} = \boldsymbol{\mathcal{T}} + \boldsymbol{\mathcal{N}} \in \mathbb{R}^{n \times n \times n}$, where $\boldsymbol{\mathcal{T}} = \sum_{i=1}^r \boldsymbol{x}_i^{(1)} \circ \boldsymbol{x}_i^{(2)} \circ \boldsymbol{x}_i^{(3)}$. The entries of $\boldsymbol{x}_i^{(j)} \in \mathbb{R}^n$ ($i \in [r]$, $j \in [3]$) are drawn from $\mathcal{N}(0, 1)$. The entries of the noise tensor $\boldsymbol{\mathcal{N}}$ are drawn from $\mathcal{N}(0, (v\sigma_{\boldsymbol{\mathcal{T}}})^2)$, where $\sigma_{\boldsymbol{\mathcal{T}}}$ denotes the standard deviation of the entries of $\boldsymbol{\mathcal{T}}$ and $v$ means the noise level. We randomly mask a fraction (which we call missing rate) of the entries to test the performance of tensor completion.

We compare the performance of LRTC-Schatten-$p$ and LRTC-ENR (solved by BCDE and LBFGS) in Figure 3, in which we set $r = 10$ and considered different noise levels, missing rates, and $p$ values. In all methods, we set $k = 2r = 20$, $t_{\max} = 500$, and select $\lambda$ from $\{0.01, 0.1, 0.5, 1, 5, 10, 50, 100, 500\}$. We can see that:

a. *Regularization is helpful.* In Figure 3 (a-d), the recovery error when there is no regularization (i.e., $\lambda = 0$) is higher than the others.

b. *Smaller $p$ leads to lower recovery error?* In almost all cases, $p < 1$ outperforms $p = 1$, verifying the superiority of Schatten-$p$ quasi-norm over nuclear norm in LRTC (both LRTC-ENR and LRTC-Schatten-$p$). In LRTC-ENR solved by BCDE and LBFGS, smaller $p$ provides lower recovery error provided that $p$ is not too small (e.g. $p \geq 1/6$), which matches Theorem 6 and Corollary 1.

c. *LRTC-ENR v.s. LRTC-Schatten-$p$* Compared to Euclidean norm minimization, direct Schatten-$p$ norm minimization has higher recovery error and time cost because the optimization problem is more difficult to solve.

d. *BCDE v.s. LBFGS* When $p$ is too small (i.e., smaller than $1/6$), the recovery error of LRTC-ENR solved by BCDE does not change, but the recovery error of LRTC-ENR solved by LBFGS increases. One possible reason is that LBFGS is not effective in handling nonsmooth objective functions and is often stuck in bad local minima or even saddle points, especially when $p$ is very small. When $p < 1/3$, LRTC-ENR has at least one nonsmooth nonconvex subproblem, which brings difficulty to LBFGS. However, the time cost of LBFGS is less than that of BCDE.

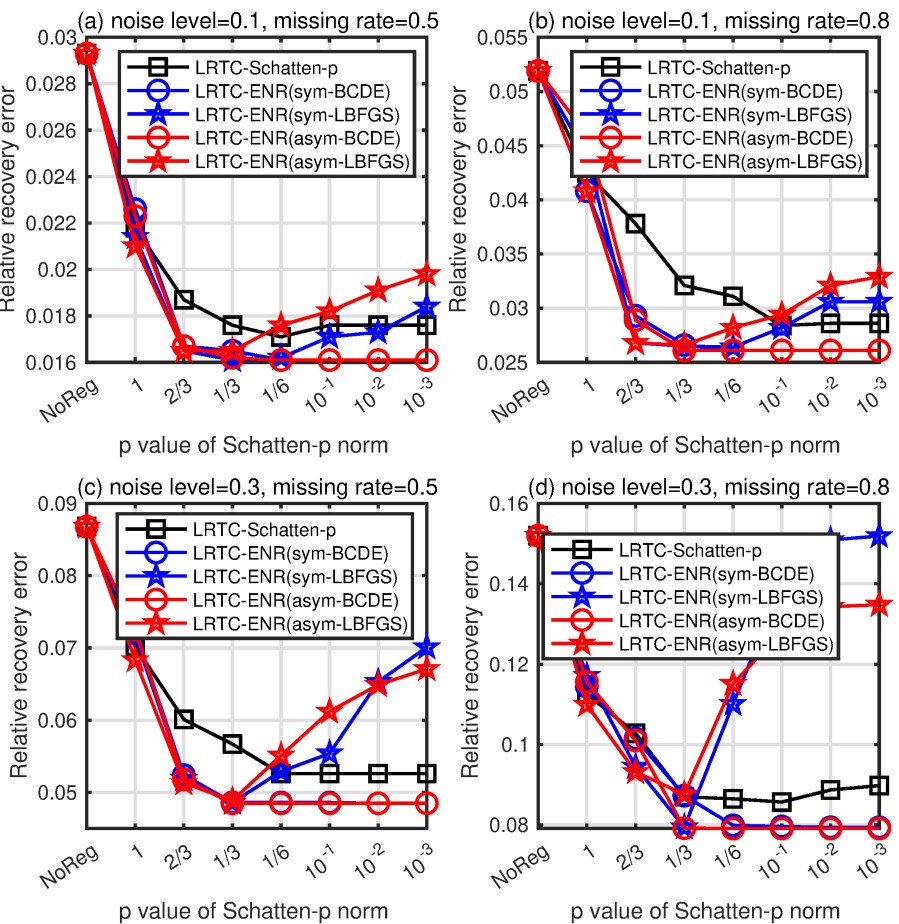

Figure 3: Performance (average of 20 repeated trials) of Schatten-$p$ (quasi) norm with different $p$ in the case of different noise level $v$ for the noise distribution $\mathcal{N}(0, (v\sigma_{\mathcal{T}})^2)$; "NoReg" means LRTC without regularization (Acar et al., 2011); the time costs (500 iterations) of the five methods are 8.6s, 3.0s, 2.1s, 2.8s, and 2.3s respectively). The results of LRTC-ENR with symmetric regularizers of $p = 1$, $2/3$, $1/3$ are attributed to (Cheng et al., 2016), (Bazerque et al., 2013), (Yang et al., 2016) respectively.

    e. *Symmetric regularizer v.s. asymmetric regularizer*    It can be found that the performance of asym-BCDE and sym-BCDE are very similar and sym-LBFGS outperforms asym-LBFGS in many cases. These empirical results are not consistent with our intuition that the asymmetric regularizers are easier to optimize than the symmetric regularizers. A possible reason is that the soft-thresholding for $q = 1$ and iteratively reweighted method for $q < 1$ of the symmetric regularizers are as effective as the gradient descent for $q = 2$ of the asymmetric regularizers. When $p < 1/3$, the $q$ in asym-LBFGS is much less than the $q$ in sym-LBFGS and hence leads to higher instability in computing the gradient and Hessian for LBFGS.

Based on the above results, we suggest using BCDE to solve LRTC-ENR on small datasets and using LBFGS to solve LRTC-ENR on large datasets. In Schatten-$p$ quasi norm, we suggest using $p = 1/3$ (Yang et al., 2016) or $p = 1/6$.

**Comparison with other tensor completion methods**    We compare LRTC-ENR with HaLRTC (Liu et al., 2012), TenALS (Jain & Oh, 2014), TMac (Xu et al., 2013), BCPF (Zhao et al., 2015), Rprecon (Kasai & Mishra, 2016), KBR-TC (Xie et al., 2018), TRLRF (Yuan et al., 2019). In TenALS, TMac, BCPF, Rprecon, TRLRF, and LRTC-ENR, we need to determine the factorization size (or the initial rank in other words) beforehand. These methods, compared to HaLRTC and KBR-TC, may apply to large tensors efficiently

provided that the ranks are sufficiently small. Particularly, in TMac, BCPF, and LRTC-ENR, the rank is adjusted adaptively.

Figure 4 shows the recovery of LRTC-ENR ($p = 1/3$(Yang et al., 2016), solved by LBFGS) and seven baselines on the synthetic data with $r = 10$. In all methods except HaLRTC and KBR-TC, one has to determine the initial rank beforehand. For these methods except Rprecon, we have set the initial rank to $2r$ because in practice it is difficult to know the true rank. The multi-rank of Rprecon is set to $(r, r, r)$ because it performs the best. The results in the figure show that Rprecon, BCPF, and LRTC-ENR ($p = 1/3$ (Yang et al., 2016)) outperformed other methods.

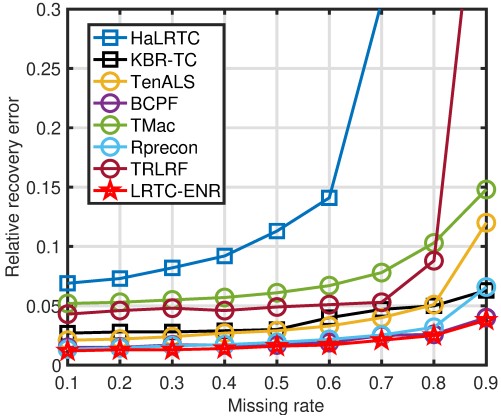

Figure 4: Low-rank tensor completion on synthetic data ($r = 10$): LRTC-ENR ($p = 1/3$ (Yang et al., 2016)) v.s. seven baseline methods.

Figure 5: Low-rank tensor completion on synthetic data: LRTC-ENR ($p = 1/3$ (Yang et al., 2016)) v.s. baseline methods, $r = 50$; (a) recovery error with different missing rates; (b) average time cost in all cases of missing rates.

When the missing rate is 0.9 in Figure 4, we observed $12,500$ entries, much more than the minimum number (1,500) of freedom parameters required to determine the tensor uniquely. It means the tensor completion problem is too easy. Hence we increase $r$ to 50 and report the recovery error and computational time of KBR-TC, TenALS, BCPF, TMac, and LRTC-ENR ($p = 1/3$ (Yang et al., 2016)) in Figure 5, where we did not report the results of other methods because their recovery errors are much higher than those reported methods. We see the recovery error of BCPF and LRTC-ENR ($p = 1/3$ (Yang et al., 2016)) are lower than those of other methods. When the missing rate is sufficiently high, LRTC-ENR ($p = 1/3$ (Yang et al., 2016)) has less recovery error than BCPF. Moreover, LRTC-ENR (200 iterations) is at least 15 times faster than BCPF and 5 times faster than KBR-TC.

### 5.1.2 Multi-spectral image inpainting

We use the Columbia multi-spectral image database (Yasuma et al., 2008) to show the performance of tensor completion methods in image inpainting problems. The dataset consists of the multi-spectral images of 32 real-world scenes of a variety of real-world materials and objects. The spatial resolution of the images is $512 \times 512$ and the number of spectral bands is 31. Thus the data of each image is a tensor of size $512 \times 512 \times 31$.

In our experiments, we pre-scale the image of every channel to $[0, 1]$. Since TRLRF (Yuan et al., 2019) and Rprecon (Kasai & Mishra, 2016) have extremely high computational costs on these tensors, we omit their implementations here. In (Xie et al., 2018), it has been shown that BCPF (Zhao et al., 2015) significantly outperformed HaLRTC (Liu et al., 2012) as well as a few other baselines on this dataset, so we will not compare HaLRTC. The initial rank of TMac (Xu et al., 2013) and LRTC-ENR are set to 100. When the initial rank is 100, it takes TenALS (Jain & Oh, 2014) and BCPF (Zhao et al., 2015) more than 1500 seconds on each image, thus we set the initial rank to 50. Since the tensor rank is very small compared to the size and the images are noiseless, we here consider highly incomplete images of which the missing rates are no less than 0.95.

As an example, Figure 6 shows the recovery errors of LRTC-ENR (at different $p$) and the competing methods on the first image of the dataset when the missing rate is 0.95, 0.97, or 0.99. When the missing rate is 0.95, KBR-TC outperformed LRTC-ENR. When the missing rate is 0.99, LRTC-ENR outperformed all other methods. In Figure 6 (a), for LRTC-ENR, $p = 1/6$ is the best choice, while in Figure 6 (b) and (c), $p = 1/3$ (Yang et al., 2016) has the least recovery error. These results are consistent with the theorems: a smaller $p$ but not too small leads to a lower recovery error; $p = 1/d$ is better than any $p > 1/d$.

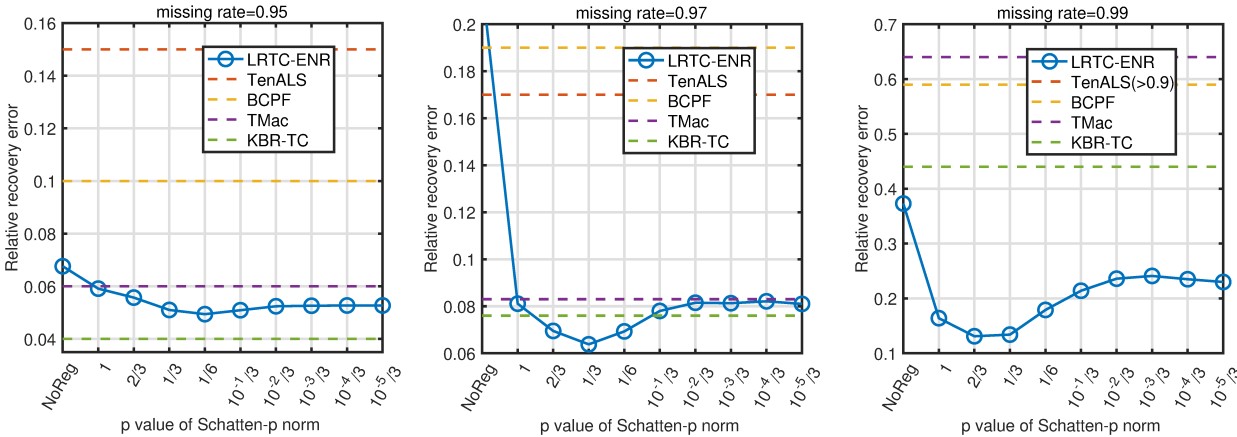

Figure 6: Recovery error on the first image of the MSI dataset. The results of LRTC-ENR with $p = 1$, $2/3$, $1/3$ are attributed to (Cheng et al., 2016), (Bazerque et al., 2013), (Yang et al., 2016) respectively.

Figures 7 and 8 visualize the recovery performance on the first image when the missing rates are 0.97 and 0.99 respectively. Visually, the recovery performance of LRTC-ENR ($p = 1/3$ (Yang et al., 2016)) is much better than those of other methods. The average results (PSNR) on the 32 images are reported in Table 2. KBR-TC and LRTC-ENR ($p = 1/3$ (Yang et al., 2016)) outperformed other methods significantly. When the missing rate is 0.97 or 0.99, LRTC-ENR ($p = 1/3$ (Yang et al., 2016)) outperformed KBR-TC, and the improvement is significant according to the paired t-test. In addition, the time cost of LRTC-ENR ((Yang et al., 2016)) is only 15% of that of KBR-TC.

Table 2: Average recovery performance (PSNR) on the Columbia MSI dataset of 32 images (the $p$-value is from the paired t-test between KBR-TC and LRTC-ENR)

| Missing rate | 0.95 | 0.97 | 0.99 | Time |
|---|---|---|---|---|
| TenALS (Jain & Oh, 2014) | 30.1±4.4 | 25.9±3.22 | 13.3±4.9 | 500s |
| BCPF (Zhao et al., 2015) | 30.4±5.8 | 26.8±5.6 | 20.2±3.3 | 450s |
| TMac (Xu et al., 2013) | 33.3±5.4 | 30.6±4.9 | 17.9±3.1 | 190s |
| KBR-TC (Xie et al., 2018) | **37.1**±4.9 | 31.4±4.9 | 20.5±2.8 | 730s |
| **LRTC-ENR** ($p = 1/3$ (Yang et al., 2016)) | 35.4±4.4 | **32.4**±5.1 | **26.1**±5.4 | **110**s |
| $p$-value (t-test) | $5\times10^{-11}$ | $5\times10^{-4}$ | $2\times10^{-8}$ | 0 |

## 5.2 Experiments of Tensor Robust PCA

### 5.2.1 Synthetic data

We generate synthetic tensors by $\mathcal{D} = \mathcal{T} + \mathcal{N} + \mathcal{E}$. The low-rank tensor $\mathcal{T}$ is given by $\mathcal{T} = \sum_{i=1}^{r} w_i \boldsymbol{x}_i^{(1)} \circ \boldsymbol{x}_i^{(2)} \circ \boldsymbol{x}_i^{(3)}$, where $w_i = i/r$ and the entries of $\boldsymbol{x}_i^{(j)} \in \mathbb{R}^n$ ($i \in [r]$, $j \in [3]$) are drawn from $\mathcal{N}(0, 1)$. $\mathcal{N}$ is a dense noise tensor drawn from $\mathcal{N}(0, (0.1\sigma_{\mathcal{T}})^2)$ and $\mathcal{E}$ is a sparse noise tensor drawn from $\mathcal{N}(0, \sigma_{\mathcal{T}}^2)$. We set $n = 50$, $r = 25$ and evaluate the TRPCA performance in recovering $\mathcal{T}$ in the case of different sparsity

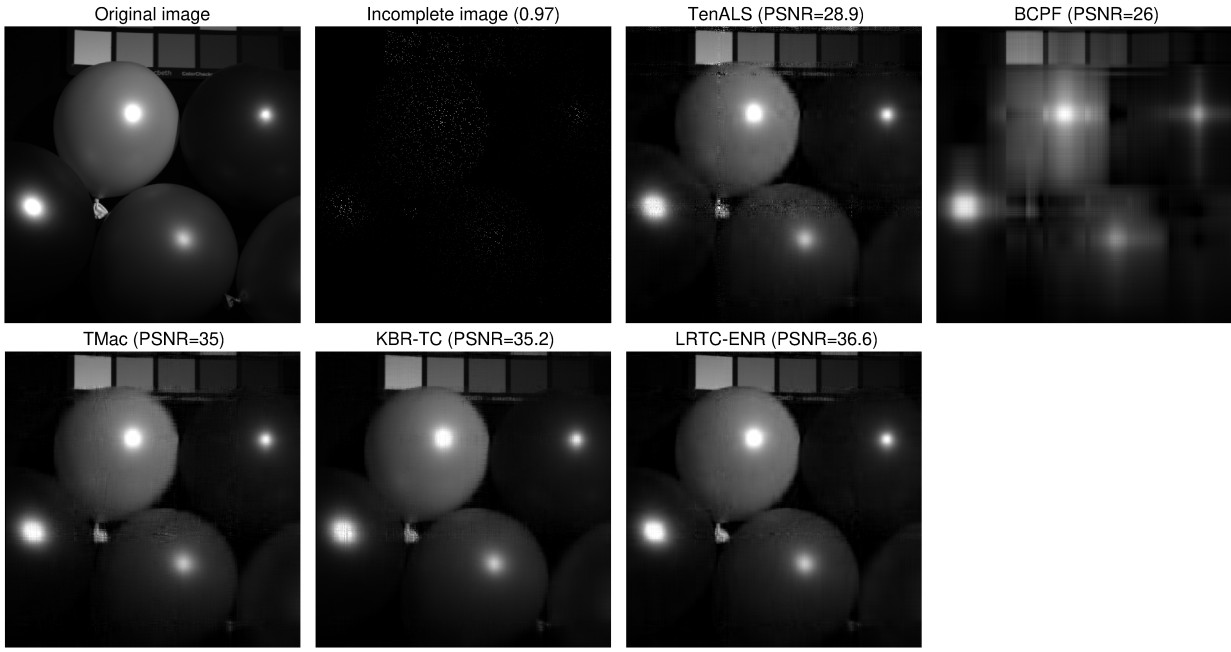

Figure 7: Inpainting performance on the first image (band 16) of the MSI dataset when the missing rate is 0.97. In LRTC-ENR, $p = 1/3$ (Yang et al., 2016).

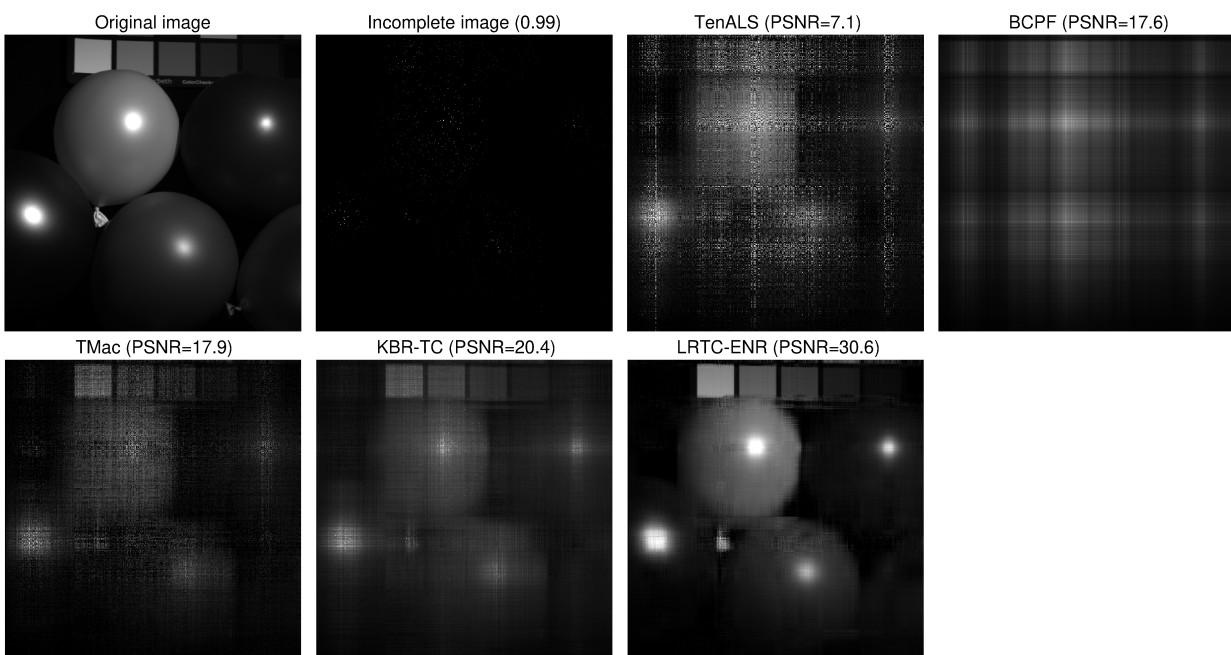

Figure 8: Inpainting performance on the first image (band 16) of the MSI dataset when the missing rate is 0.99. In LRTC-ENR, $p = 1/3$ (Yang et al., 2016).

(termed as noise density) of $\mathcal{E}$. The evaluation metric is the relative recovery error defined by

$$\text{Relative recovery error} = \|\boldsymbol{\mathcal{T}} - \hat{\boldsymbol{\mathcal{T}}}\|_F / \|\boldsymbol{\mathcal{T}}\|_F.$$

We compare TRPCA-ENR with KBR-PCA (Xie et al., 2018). TRPCA (denoted by T-TRPCA) proposed by (Lu et al., 2019) and the robust tensor decomposition method OITNN-L proposed by (Wang et al., 2020) are under the assumption of $t$-product (not CP decomposition) and hence are not compatible in our setting of synthetic data. We only compare them on real data later. The hyper-parameters of KBR-RPCA and TRPCA-ENR[3] are sufficiently tuned to provide their best performance. Figure 9(a) shows the recovery errors of KBR-RPCA and TRPCA-ENR (with $p = 2/3$ (Bazerque et al., 2013), 1/3 (Yang et al., 2016), and 1/6) when the noise density increases from 0.1 to 0.7. We see that TRPCA-ENR consistently outperformed KBR-RPCA. In TRPCA-ENR, $p = 1/6$ is better than $p = 2/3$ and 1/3 but the difference is not obvious. In Figure 9(b), we show the influence of different $p$ on the recovery error. It can be found that a smaller $p$ (but not too small, e.g. larger than $10^{-2}$) yields less recovery error.

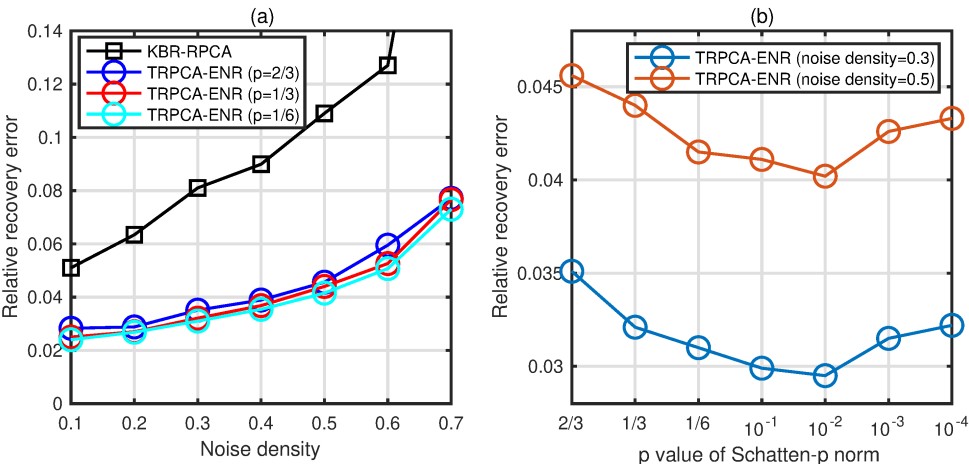

Figure 9: Performance of TRPCA on synthetic data (average of 20 repeated trials): (a) KBR-RPCA v.s. TRPCA-ENR (initial rank=2r); (b) relative recovery error of TRPCA-ENR with different $p$. The results of TRPCA-ENR with $p = 2/3$, 1/3 are attributed to (Bazerque et al., 2013), (Yang et al., 2016) respectively.

In Figure 10, we compare the performance of symmetric regularizers and asymmetric regularizers in TRPCA-ENR. We see that in all cases, the recovery error and time cost given by the asymmetric regularizers are lower than those given by the symmetric regularizers. The reason is that there are more subproblems in the optimization associated with the symmetric regularizers that have no closed-form solutions. The optimization with asymmetric regularizers reaches the stop criterion in 200 or 300 iterations while the optimization with symmetric regularizers requires much more iterations or even does not reach the stop criterion in 500 iterations.

### 5.2.2 Color image denoising

We compare TRPCA-ENR with KBR-PCA (Xie et al., 2018), T-TRPCA (Lu et al., 2019), and OITNN-L (Wang et al., 2020) in the task of color image denoising on nine color images (displayed by Figure 11) of size $256 \times 256 \times 3$ used in (Wang et al., 2020). For each image, we randomly set 10% of the tensor elements to random values in $[0, 1]$. We tuned all hyper-parameters carefully within wide ranges to provide the best denoising performance of the methods. As a result, in KBR-RPCA, we set $\lambda = 2$ or 3. In OITNN-L, we set $\lambda_L = 4$ or 5 and $\lambda_S = 0.16$. In T-TRPCA, we set $\lambda = 1/\sqrt{768}$ or $2/\sqrt{768}$. In TRPCA-ENR, we set the initial rank to a relatively large value 200 because the approximate rank of the image tensors is not too small and the method can adjust the rank adaptively; we set $\lambda_e = 0.02$ and consider the cases of $p = 2/3$, 1/3, 1/6, and 1/10, for which $\lambda_x = 0.1$, 0.1, 0.02, and 0.015.

The average PSNR with standard deviation across 20 repeated trials are reported in Table 3. We see that the PSNRs of TRPCA-ENR methods are much lower than those of other methods in all cases. In addition, TRPCA-ENR with smaller $p$ has higher PSNR and TRPCA-ENR with $p = 1/10$ performs the best. Figure

---

[3]Throughout this paper, we set $\mu = 10$ and $t_{\max} = 500$ for Algorithms 4 and 5.

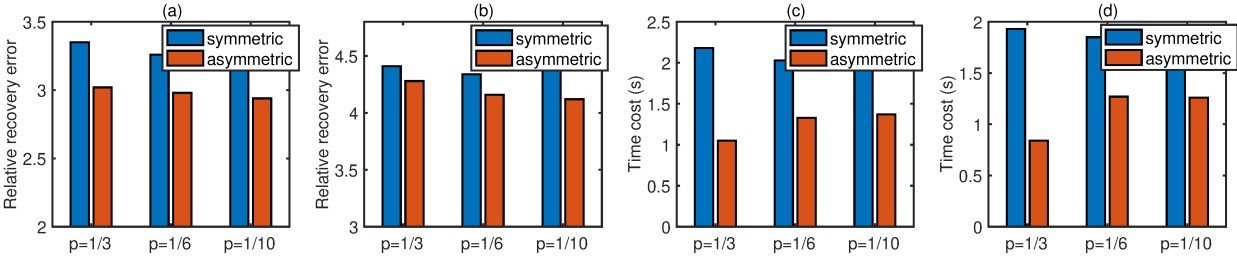

Figure 10: Comparison of symmetric and asymmetric regularizers in TRPCA-ENR on the synthetic data (average of 20 repeated trials): (a) relative recovery error when the noise density is 0.3; (b) relative recovery error when the noise density is 0.5; (c) time cost when the noise density is 0.3; (d) time cost when the noise density is 0.5 (the maximum iteration is 500 and the stop criterion is that the relative change of the decision variables is less than $10^{-4}$).

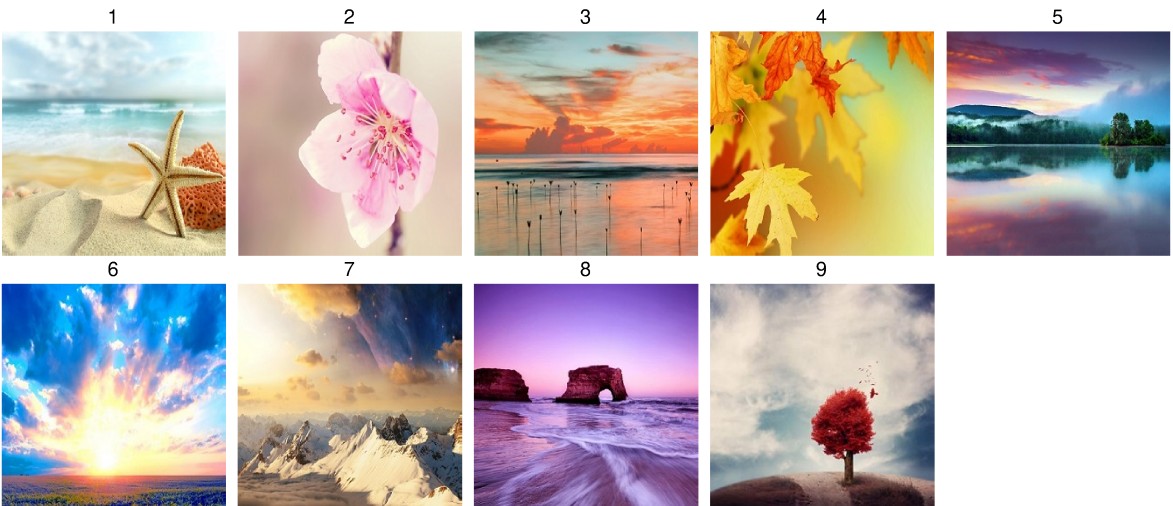

Figure 11: Nine color images for denoising

12 visualizes the denoising performance on Image 1, showing that TRPCA-ENR methods are better than other methods. These results provided evidence that tensor Schattern-$p$ quasi-norm minimization is able to provide higher recovery accuracy in TRPCA when a relatively smaller $p$ is used.

# 6 Conclusion

We presented a framework of Euclidean-Norm-Induced Regularization (ENR) for LRTC and TRPCA, though some special cases were already studied in previous works (Bazerque et al., 2013; Cheng et al., 2016; Shi et al., 2017; Lacroix et al., 2018). We in this work focused more on the theory of Schattern-$p$ quasi-norm and ENR in LRTC and TRPCA than on the numerical comparison of different LRTC or TRPCA methods. We proved that the Schattern-$p$ quasi-norm with a decently small $p$ or ENR with a decently small $q$ provide tighter generalization error bounds for LRTC. More importantly, our theory indicates that for LRTC on $d$-order tensors, $p = 1/d$ is always better than any $p > 1/d$ in terms of the generalization error bound. We also proved that in TRPCA for orthogonal tensors, a smaller $p$ in Schattern-$p$ quasi-norm can lead to a lower recovery error bound under some mild conditions. In addition to the theoretical results, our numerical results verified the effectiveness of ENR in comparison to a few baselines of LRTC and TRPCA.

Table 3: PSNR of color image denoising. The results of TRPCA-ENR with symmetric regularizers of $p = 2/3,\ 1/3$ are attributed to (Bazerque et al., 2013), (Yang et al., 2016) respectively.

| | Image 1 | Image 2 | Image 3 | Image 4 | Image5 |
|---|---|---|---|---|---|
| KBR-RPCA (Xie et al., 2018) | 30.25±0.06 | 33.36±0.22 | 31.93±0.12 | 31.87±0.08 | 35.08±0.12 |
| T-TRPCA (Lu et al., 2019) | 31.16±0.05 | 32.75±0.06 | 32.87±0.11 | 32.52±0.06 | 33.91±0.10 |
| OITNN-O (Wang et al., 2020) | 28.46±0.04 | 29.99±0.03 | 29.16±0.05 | 28.13±0.04 | 29.86±0.03 |
| TRPCA-ENR ($p=\frac{2}{3}$, sym) | 34.23±0.12 | 35.44±0.09 | 34.82±0.18 | 34.23±0.14 | 37.14±0.06 |
| TRPCA-ENR ($p=\frac{1}{3}$, sym) | 34.68±0.18 | 36.88±0.16 | 37.52±0.12 | 34.53±0.31 | 37.78±0.04 |
| TRPCA-ENR ($p=\frac{1}{3}$, asym) | 34.81±0.16 | 37.04±0.06 | 37.61±0.19 | 34.92±0.28 | 38.16±0.10 |
| TRPCA-ENR ($p=\frac{1}{6}$, asym) | 34.96±0.40 | 36.94±0.12 | 37.57±0.33 | 35.12±0.17 | 38.14±0.11 |
| TRPCA-ENR ($p=\frac{1}{10}$, asym) | **35.45**±0.12 | **37.14**±0.10 | **37.76**±0.14 | **35.38**±0.46 | **38.28**±0.13 |
| | Image 6 | Image 7 | Image 8 | Image 9 | |
| KBR-RPCA (Xie et al., 2018) | 31.36±0.09 | 30.08±0.10 | 32.41±0.05 | 34.28±0.12 | |
| T-TRPCA (Lu et al., 2019) | 28.96±0.06 | 29.36±0.09 | 31.49±0.11 | 33.67±0.14 | |
| OITNN-O (Wang et al., 2020) | 27.39±0.05 | 27.46±0.05 | 28.61±0.07 | 30.06±0.06 | |
| TRPCA-ENR ($p=\frac{2}{3}$, sym) | 35.09±0.12 | 33.67±0.10 | 35.66±0.16 | 36.12±0.05 | |
| TRPCA-ENR ($p=\frac{1}{3}$, sym) | 33.21±0.54 | 34.10±0.23 | 36.09±0.14 | 37.95±0.16 | |
| TRPCA-ENR ($p=\frac{1}{3}$, asym) | 33.54±0.71 | 34.29±0.18 | 36.38±0.09 | **38.13**±0.12 | |
| TRPCA-ENR ($p=\frac{1}{6}$, asym) | 33.62±0.76 | 34.44±0.12 | 36.49±0.08 | 37.96±0.10 | |
| TRPCA-ENR ($p=\frac{1}{10}$, asym) | **35.18**±0.35 | **34.57**±0.14 | **36.71**±0.06 | 38.01±0.16 | |

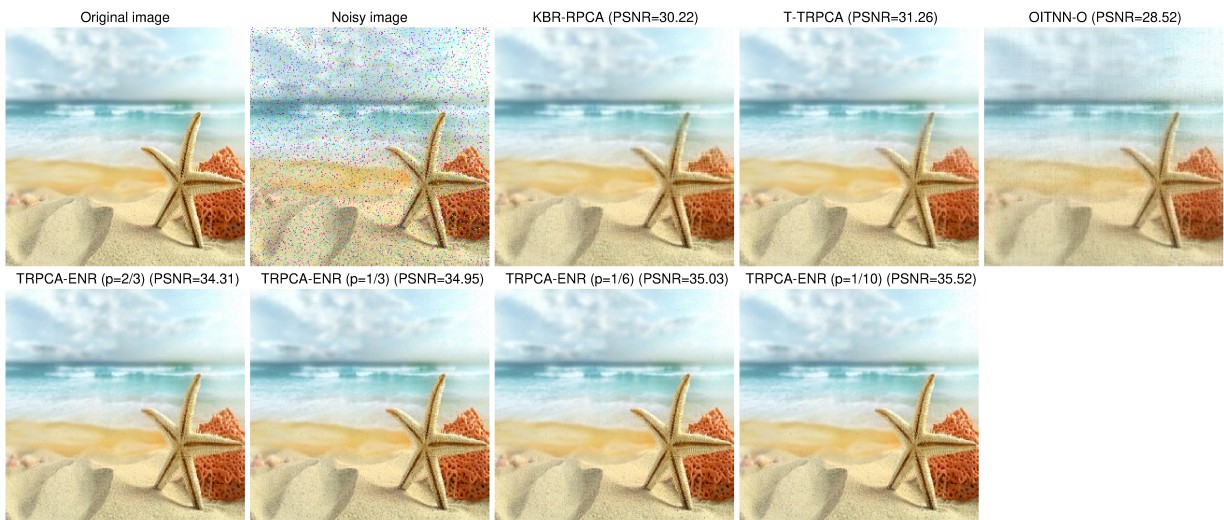

Figure 12: Denoising performance on Image 1.

## Acknowledgements

Jicong Fan gratefully acknowledges support from the Youth program 62106211 of the National Natural Science Foundation of China and the research funding T00120210002 of Shenzhen Research Institute of Big Data. Zhao Zhang gratefully acknowledges support from the National Natural Science Foundation of China under Grant no. 62072151 and Anhui Provincial Natural Science Fund for the Distinguished Young Scholars (2008085J30). Madeleine Udell gratefully acknowledges support from NSF Award IIS-1943131, the ONR Young Investigator Program, and the Alfred P. Sloan Foundation. Jicong Fan and Madeleine Udell are the corresponding authors of this paper.

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

# A  Optimization for LRTC-ENR

## A.1  Block Coordinate Descent with Extrapolation

There are $d$ blocks of decision variables in problem (2), i.e. $\{\boldsymbol{X}^{(j)}\}^{j \in [d]}$, where $\boldsymbol{X}^{(j)} = [\boldsymbol{x}_1^{(j)}, \boldsymbol{x}_2^{(j)}, \ldots, \boldsymbol{x}_k^{(j)}]$. We propose to find a critical point of (2) by Block Coordinate Descent (BCD) with Extrapolation (BCDE for short) (Xu & Yin, 2013), which is more efficient than BCD. For simplicity, here we only present the optimization of (2) with the regularizer shown in (3). The optimization can be easily extended to (2) with other regularizers we proposed in Section 2.

Let

$$
\begin{aligned}
\mathcal{L}\left(\{\boldsymbol{X}^{(j)}\}^{j \in [d]}\right) :=& \frac{1}{2}\left\|\boldsymbol{\mathcal{M}} * \left(\boldsymbol{\mathcal{D}} - \sum_{i=1}^{k} \boldsymbol{x}_i^{(1)} \circ \boldsymbol{x}_i^{(2)} \ldots \circ \boldsymbol{x}_i^{(d)}\right)\right\|_F^2 \\
=& \frac{1}{2}\left\|\boldsymbol{M}_{(j)} * \left(\boldsymbol{D}_{(j)} - \boldsymbol{X}^{(j)}[(\boldsymbol{X}^{(i)})^{\odot i \neq j}]^\top\right)\right\|_F^2
\end{aligned}
$$

and

$$
\mathcal{R}_{pd}(\boldsymbol{X}^{(j)}) := \sum_{j=1}^{d} \|\boldsymbol{x}_i^{(j)}\|^{pd},
$$

where $[(\boldsymbol{X}^{(i)})^{\odot i \neq j}] = \boldsymbol{X}^{(d)} \odot \ldots \odot \boldsymbol{X}^{(j+1)} \odot \boldsymbol{X}^{(j-1)} \odot \ldots \odot \boldsymbol{X}^{(1)}$. We initialize $\{\boldsymbol{X}^{(j)}\}^{j \in [d]}$ randomly and rescale $\boldsymbol{x}$ to have unit Euclidean norm:

$$
\boldsymbol{x}_i^{(j)} \sim \mathcal{N}(0, 1), \quad \boldsymbol{x}_i^{(j)} \leftarrow \boldsymbol{x}_i^{(j)}/\|\boldsymbol{x}_i^{(j)}\|, \quad i \in [k], \ j \in [d]. \tag{11}
$$

Then at iteration $t$, for $j = 1, 2, \ldots, d$, we first perform an extrapolation

$$
\hat{\boldsymbol{X}}_{t-1}^{(j)} = \boldsymbol{X}_{t-1}^{(j)} + \omega_{j,t-1}(\boldsymbol{X}_{t-1}^{(j)} - \boldsymbol{X}_{t-2}^{(j)}), \tag{12}
$$

where $\omega_{j,t-1} \geq 0$ controls the size of the extrapolation at iteration $t$. Then we perform a proximal step as

$$
\begin{aligned}
\boldsymbol{X}_t^{(j)} =& \underset{\boldsymbol{X}^{(j)}}{\operatorname{argmin}} \ \left\langle \boldsymbol{\nabla}_{\hat{\boldsymbol{X}}_{t-1}^{(j)}} \mathcal{L}, \ \boldsymbol{X}^{(j)} - \hat{\boldsymbol{X}}_{t-1}^{(j)} \right\rangle + \frac{\hat{L}_{j,t}}{2}\|\boldsymbol{X}^{(j)} - \hat{\boldsymbol{X}}_{t-1}^{(j)}\|_F^2 + \lambda \mathcal{R}_{pd}(\boldsymbol{X}^{(j)}) \\
\triangleq& \operatorname{prox}_{pd}^\lambda\left(\hat{\boldsymbol{X}}_{t-1}^{(j)} - \hat{L}_{j,t}^{-1}\boldsymbol{\nabla}_{\hat{\boldsymbol{X}}_{t-1}^{(j)}} \mathcal{L}\right).
\end{aligned} \tag{13}
$$

In (13), $\boldsymbol{\nabla}_{\hat{\boldsymbol{X}}_{t-1}^{(j)}} \mathcal{L} = \left(\boldsymbol{M}_{(j)} * \left(\boldsymbol{D}_{(j)} - \hat{\boldsymbol{X}}_{t-1}^{(j)}[(\boldsymbol{X}^{(i)})_t^{\odot i \neq j}]^\top\right)\right)\left(-[(\boldsymbol{X}^{(i)})_t^{\odot i \neq j}]\right)$, $[(\boldsymbol{X}^{(i)})_t^{\odot i \neq j}] = \boldsymbol{X}_{t-1}^{(d)} \odot \cdots \odot \boldsymbol{X}_{t-1}^{(j+1)} \odot \boldsymbol{X}_t^{(j-1)} \odot \cdots \odot \boldsymbol{X}_t^{(1)}$, and $\hat{L}_{j,t}$ is larger than the Lipschitz constant of $\boldsymbol{\nabla}_{\hat{\boldsymbol{X}}_{t-1}^{(j)}} \mathcal{L}$. We approximate it by

$$
\hat{L}_{j,t} = \varrho\sqrt{\frac{|\Omega|}{\prod_{i=1}^{d} n_i}}\left\|[(\boldsymbol{X}^{(i)})_t^{\odot i \neq j}]\right\|_2^2, \tag{14}
$$

where $\varrho = 0.5$, $1$, or $2$. The derivation of (14) is as follows. For convenience, we denote a perturbed copy of $\hat{\boldsymbol{X}}_{t-1}^{(j)}$ by $\hat{\boldsymbol{Z}}_{t-1}^{(j)}$. We have

$$
\begin{aligned}
&\|\boldsymbol{\nabla}_{\hat{\boldsymbol{X}}_{t-1}^{(j)}} \mathcal{L} - \boldsymbol{\nabla}_{\hat{\boldsymbol{Z}}_{t-1}^{(j)}} \mathcal{L}\|_F \\
=&\| \left( \boldsymbol{M}_{(j)} * \left( \boldsymbol{D}_{(j)} - \hat{\boldsymbol{X}}_{t-1}^{(j)}[(\boldsymbol{X}^{(i)})_t^{\odot i \neq j}]^\top \right) \right) \left( - [(\boldsymbol{X}^{(i)})_t^{\odot i \neq j}] \right) \\
&- \left( \boldsymbol{M}_{(j)} * \left( \boldsymbol{D}_{(j)} - \hat{\boldsymbol{Z}}_{t-1}^{(j)}[(\boldsymbol{X}^{(i)})_t^{\odot i \neq j}]^\top \right) \right) \left( - [(\boldsymbol{X}^{(i)})_t^{\odot i \neq j}] \right)\|_F \\
\leq&\|[(\boldsymbol{X}^{(i)})_t^{\odot i \neq j}]\|_2 \|\boldsymbol{M}_{(j)} * \left( \hat{\boldsymbol{X}}_{t-1}^{(j)}[(\boldsymbol{X}^{(i)})_t^{\odot i \neq j}]^\top - \hat{\boldsymbol{Z}}_{t-1}^{(j)}[(\boldsymbol{X}^{(i)})_t^{\odot i \neq j}]^\top \right)\|_F \\
\leq&\|[(\boldsymbol{X}^{(i)})_t^{\odot i \neq j}]\|_2 \varrho \sqrt{\frac{|\Omega|}{\prod_{i=1}^d n_i}} \|\left( \hat{\boldsymbol{X}}_{t-1}^{(j)} - \hat{\boldsymbol{Z}}_{t-1}^{(j)} \right)[(\boldsymbol{X}^{(i)})_t^{\odot i \neq j}]^\top\|_F \\
\leq&\varrho \sqrt{\frac{|\Omega|}{\prod_{i=1}^d n_i}} \|[(\boldsymbol{X}^{(i)})_t^{\odot i \neq j}]\|_2^2 \|\hat{\boldsymbol{X}}_{t-1}^{(j)} - \hat{\boldsymbol{Z}}_{t-1}^{(j)}\|_F.
\end{aligned}
$$

Here $0 < \varrho \leq \sqrt{\frac{\prod_{i=1}^d n_i}{|\Omega|}}$ is some suitable constant.

In (13), when $p = 1/d$, $\mathrm{prox}_1^\lambda(\boldsymbol{Y}) = \Phi_\lambda(\boldsymbol{Y})$, where $\Phi$ is the column-wise soft-thresholding operator (Parikh et al., 2014) defined by

$$
\Phi_\lambda(\boldsymbol{y}) = \begin{cases} \frac{(\|\boldsymbol{y}\| - \lambda)\boldsymbol{y}}{\|\boldsymbol{y}\|}, & \text{if } \|\boldsymbol{y}\| > \lambda; \\ \boldsymbol{0}, & \text{otherwise.} \end{cases}
$$

When $p = 2/d$, $\mathrm{prox}_2^\lambda(\boldsymbol{Y}) = \frac{\hat{L}_{j,t}}{\hat{L}_{j,t} + 2\lambda}\boldsymbol{Y}$. When $p > 2/d$, we can estimate $\boldsymbol{X}_t^{(j)}$ by gradient descent. When $p < 1/d$, (13) is nonconvex and nonsmooth. Then we update $\boldsymbol{X}^{(j)}$ with iteratively reweighted method (Lu, 2014), which is given by Algorithm 1.

---

**Algorithm 1** $\min_{\boldsymbol{Y}} \frac{1}{2}\|\boldsymbol{Y} - \boldsymbol{G}\|_F^2 + \tilde{\lambda} \sum_{i=1}^k \|\boldsymbol{y}_i\|^q$

**Require:** $\boldsymbol{G}$, $q$, $\tilde{\lambda}$, $t_q$, $\epsilon$.
1: $\boldsymbol{Y} \leftarrow \boldsymbol{G}$.
2: **for** $t = 1, 2, \ldots, t_q$ **do**
3:     $\boldsymbol{W} = \mathrm{diag}\left((\|\boldsymbol{y}_1\| + \epsilon)^{\frac{q-2}{2}}, \ldots, (\|\boldsymbol{y}_k\| + \epsilon)^{\frac{q-2}{2}}\right)$.
4:     $\boldsymbol{Y} = \boldsymbol{G}(\boldsymbol{I} + 2\tilde{\lambda}\boldsymbol{W}\boldsymbol{W}^T)^{-1}$.
5: **end for**
**Ensure:** $\boldsymbol{Y}$.

---

In (12), the parameter $\omega_{jt}$ is determined as

$$
\omega_{j,t-1} = \delta\sqrt{\hat{L}_{j,t-2}/\hat{L}_{j,t-1}}, \tag{15}
$$

where $\delta < 1$. We set $\delta = 0.95$ for simplicity. The whole procedure is summarized in Algorithm 2. The convergence analysis when $p \geq 1/d$ can be found in (Xu & Yin, 2013). When $p < 1/d$, suppose Algorithm 1 returns the optimal solution, we can get similar convergence result as the case $p = 1/d$. Empirically, we found that there is no need to find the exact solution for the subproblem and instead we just perform Algorithm 1 for a few iterations, which can still provide satisfactory result. Currently, proving the convergence is out of the scope of our paper.

## A.2 L-BFGS

Though faster than BCD, the computational cost per iteration of BCDE is high when $d$ is not small because we need to compute $\boldsymbol{X}^{(j)}[(\boldsymbol{X}^{(i)})^{\odot i \neq j}]^\top$ for $d$ times in every iteration. One may consider the Jacobi-type

---

**Algorithm 2** solve LRTC-ENR by BCDE

---

**Input:** $\mathcal{D}$, $\mathcal{M}$, $k$, $\lambda$, $t_{\max}$.

1: Initialize $\{\boldsymbol{x}_i^{(j)}\}_{i\in[k]}^{j\in[d]}$ with (11), let $t=0$.
2: **repeat**
3:      $t \leftarrow t+1$.
4:      **for** $j=1,2,\ldots,d$ **do**
5:          **if** $t <= 2$ **then**
6:              $\omega_{j,t-1}=0$.
7:          **else**
8:              Compute $\omega_{j,t-1}$ by (15).
9:          **end if**
10:          Compute $\hat{\boldsymbol{X}}_{t-1}^{(j)}$ by (12).
11:          Compute $L_{j,t-1}$ by (14).
12:          Compute $\boldsymbol{X}_t^{(j)}$ by (13) or Algorithm 1.
13:      **end for**
14:      Remove the zero columns of $\boldsymbol{X}_t^{(j)}$, $j\in[d]$.
15: **until** converged or $t=t_{\max}$

**Output:** $\hat{\boldsymbol{\mathcal{T}}} = \boldsymbol{\mathcal{I}} \times_1 \boldsymbol{X}_t^{(1)} \times_2 \boldsymbol{X}_t^{(2)} \ldots \times_d \boldsymbol{X}_t^{(d)}$.

---

iteration (cheaper computation but slower convergence) rather than the Gauss-Seidel iteration in BCD and BCDE. In practice, we can use quasi-Newton methods such as L-BFGS (Liu & Nocedal, 1989) to solve problem (2) even though the objective function is nonsmooth. Particularly, we can drop the columns of $\boldsymbol{X}_t^{(j)}$ ($j \in [d]$) with nearly-zero Euclidean norms for acceleration. The corresponding algorithm[4] is shown in Algorithm 3.

---

**Algorithm 3** solve LRTC-ENR by LBFGS

---

**Input:** $\mathcal{D}$, $\mathcal{M}$, $k$, $\lambda$, $t_{\max}$.

1: Initialize $\{\boldsymbol{x}_i^{(j)}\}_{i\in[k]}^{j\in[d]}$ with (11), let $t=0$.
2: **repeat**
3:      $t \leftarrow t+1$.
4:      Compute the search directions by LBFGS.
5:      Use line search to determine the step size.
6:      Update $\boldsymbol{X}_t^{(j)}$, $j \in [d]$.
7:      For $j \in [d]$, remove the columns of $\boldsymbol{X}_t^{(j)}$ with Euclidean norms less than a small threshold, e.g. $10^{-5}$.
8: **until** converged or $t=t_{\max}$

**Output:** $\hat{\boldsymbol{\mathcal{T}}} = \boldsymbol{\mathcal{I}} \times_1 \boldsymbol{X}_t^{(1)} \times_2 \boldsymbol{X}_t^{(2)} \ldots \times_d \boldsymbol{X}_t^{(d)}$.

---

## B  Optimization for TRPCA-ENR

When $p = 2/d$, we use alternating minimization to solve the optimization of TRPCA-ENR because every subproblem has a closed-form solution. When $p = 1/d$, we may, like Section A.1, use block coordinate descent with extrapolation (Xu & Yin, 2013). However, the additional variable $\boldsymbol{\mathcal{E}}$ further slows down the convergence. We hence propose to solve (9) by the (nonconvex) alternating direction method of multipliers (ADMM) (Wang et al., 2015). Specifically, by adding auxiliary variables $\{\boldsymbol{Y}^{(j)}\}_{j=1}^d$, we reformulate (9) as

$$\underset{\{\boldsymbol{X}^{(j)},\boldsymbol{Y}^{(j)}\}_{j=1}^d,\boldsymbol{\mathcal{E}}}{\text{minimize}} \quad \frac{1}{2} \left\| \boldsymbol{D}_{(j)} - \boldsymbol{X}^{(j)}[(\boldsymbol{X}^{(i)})^{\odot i\neq j}]^\top - \boldsymbol{E}_{(j)} \right\|_F^2 + \lambda_x \mathcal{R}\left(\{\boldsymbol{y}_i^{(j)}\}_{i\in[k]}^{j\in[d]}\right) + \lambda_e \|\boldsymbol{\mathcal{E}}\|_1$$

$$\text{subject to} \quad \boldsymbol{Y}^{(j)} = \boldsymbol{X}^{(j)}, j=1,2,\ldots,d.$$

---

[4] The implementation in this paper is based on the *minFunc* MATLAB toolbox of M. Schmidt: `http://www.cs.ubc.ca/~schmidtm/Software/minFunc.html`.

Let $\{\boldsymbol{Z}^{(j)}\}_{j=1}^d$ be Lagrange multipliers and solve

$$
\begin{aligned}
\underset{\{\boldsymbol{X}^{(j)},\boldsymbol{Y}^{(j)}\}_{j=1}^d,\,\boldsymbol{\mathcal{E}}}{\text{minimize}} \quad & \frac{1}{2}\left\|\boldsymbol{D}_{(j)} - \boldsymbol{X}^{(j)}[(\boldsymbol{X}^{(i)})^{\odot i\neq j}]^\top - \boldsymbol{E}_{(j)}\right\|_F^2 \\
& + \lambda_x \sum_{i=1}^k \sum_{j=1}^d \|\boldsymbol{y}_i^{(j)}\|^{pd} + \lambda_e \|\boldsymbol{\mathcal{E}}\|_1 + \sum_{j=1}^d \left\langle \boldsymbol{Y}^{(j)} - \boldsymbol{X}^{(j)}, \boldsymbol{Z}^{(j)}\right\rangle + \frac{\mu}{2}\|\boldsymbol{Y}^{(j)} - \boldsymbol{X}^{(j)}\|_F^2,
\end{aligned}
\tag{16}
$$

where $\mu$ is the augmented Lagrange penalty parameter. Then update $\{\boldsymbol{X}^{(j)},\boldsymbol{Y}^{(j)}\}_{j=1}^d$ and $\boldsymbol{\mathcal{E}}$ sequentially to minimize (16) and update $\{\boldsymbol{Z}^{(j)}\}_{j=1}^d$ lastly. The procedures are summarized into Algorithm 4, where

$$
\boldsymbol{X}_t^{(j)} = \left((\boldsymbol{D}_{(j)} - \boldsymbol{E}_{(j)})[(\boldsymbol{X}^{(i)})^{\odot i\neq j}] + \mu\boldsymbol{Y}_{t-1}^{(j)} - \boldsymbol{Z}^{(j)}\right)\left([(\boldsymbol{X}^{(i)})^{\odot i\neq j}]^T[(\boldsymbol{X}^{(i)})^{\odot i\neq j}] + \mu\boldsymbol{I}\right)^{-1},
\tag{17}
$$

$$
\boldsymbol{Y}_t^{(j)} = \Phi_{\lambda_x/\mu}\left(\boldsymbol{X}_t^{(j)} - \boldsymbol{Z}^{(j)}/\mu\right),
\tag{18}
$$

$$
\boldsymbol{Z}^{(j)} \longleftarrow \boldsymbol{Z}^{(j)} + \mu(\boldsymbol{Y}_t^{(j)} - \boldsymbol{X}_t^{(j)}),
\tag{19}
$$

$$
\boldsymbol{\mathcal{E}}_t = \Psi_{\lambda_e}\left(\boldsymbol{\mathcal{D}} - \boldsymbol{\mathcal{I}} \times_1 \boldsymbol{X}_t^{(1)} \times_2 \boldsymbol{X}_t^{(2)} \ldots \times_d \boldsymbol{X}_t^{(d)}\right).
\tag{20}
$$

$\Psi$ is the element-wise soft-thresholding operator (Parikh et al., 2014) defined by

$$
\Psi_{\lambda_e}(v) = \text{sign}(v)\max(0, |v| - \lambda_e).
$$

---

**Algorithm 4** TRPCA-ENR $(p = 1/d)$ solved by ADMM

---

**Input:** $\boldsymbol{\mathcal{D}}$, $k$, $\lambda_x$, $\lambda_e$, $\mu$, $t_{\max}$.
1: Initialize $\{\boldsymbol{x}_i^{(j)}\}_{i\in[k]}^{j\in[d]}$ with (11); for $j = 1,\ldots,d$, let $\boldsymbol{Y}_0^{(j)} = \boldsymbol{X}_0^{(j)}$ and $\boldsymbol{Z}^{(j)} = \boldsymbol{0}$; let $t = 0$ and $\boldsymbol{\mathcal{E}} = \boldsymbol{0}$.
2: **repeat**
3:     $t \leftarrow t + 1$.
4:     **for** $j = 1, 2, \ldots, d$ **do**
5:         Compute $\boldsymbol{X}_t^{(j)}$ by (17).
6:         Compute $\boldsymbol{Y}_t^{(j)}$ by (18).
7:         Update $\boldsymbol{Z}^{(j)}$ by (19).
8:     **end for**
9:     Compute $\boldsymbol{\mathcal{E}}_t$ by (20).
10: **until** converged or $t = t_{\max}$
**Output:** $\hat{\boldsymbol{\mathcal{T}}} = \boldsymbol{\mathcal{I}} \times_1 \boldsymbol{X}_t^{(1)} \times_2 \boldsymbol{X}_t^{(2)} \ldots \times_d \boldsymbol{X}_t^{(d)}$.

---

In (9), when $p \notin \{1/d, 2/d\}$, the optimization becomes more difficult because all $d$ groups of the regularizers are nonconvex and nonsmooth. Thanks to Theorem 2, especially its (b), we can obtain arbitrarily sharp Schatten-$p$ quasi-norm regularization by using only one group of nonconvex and nonsmooth regularizers on the component vectors. The corresponding problem is

$$
\underset{\{\boldsymbol{x}_i^{(j)}\}_{i\in[k]}^{j\in[d]},\,\boldsymbol{\mathcal{E}}}{\text{minimize}} \quad \frac{1}{2}\left\|\boldsymbol{\mathcal{D}} - \sum_{i=1}^k \boldsymbol{x}_i^{(1)} \circ \boldsymbol{x}_i^{(2)} \ldots \circ \boldsymbol{x}_i^{(d)} - \boldsymbol{\mathcal{E}}\right\|_F^2 + \lambda_x \sum_{i=1}^k \left(\frac{1}{q}\|\boldsymbol{x}_i^{(1)}\|^q + \frac{1}{2}\sum_{j=2}^d \|\boldsymbol{x}_i^{(j)}\|^2\right) + \lambda_e\|\boldsymbol{\mathcal{E}}\|_1.
\tag{21}
$$

We propose to solve (21) by ADMM with the iteratively reweighted update (Algorithm 1) embedded, which is shown in Algorithm 5. In the algorithm, for $j = 2, \ldots, d$, the subproblem of $\boldsymbol{X}_t^{(j)}$ has a closed-form solution, which makes the algorithm more efficient than Algorithm 4, especially when $d$ is large. Note that the time complexity per iteration of Algorithm 4 and Algorithm 5 are $O(dk^2n^{d-1} + dk^3 + (d+1)kn^d)$ and $O(dk^2n^{d-1} + dk^3 + (d+1)kn^d + t_qk^2n)$ respectively, which are similar because $t_qk^2n$ is much less than $(d+1)kn^d$ for high-order low-rank tensors. If $p < 1/d$ and we do not use the asymmetric regularizer given by Theorem 2(b), the time complexity per iteration of the optimization is $O(dk^2n^{d-1} + dk^3 + (d+1)kn^d + dt_qk^2n)$, which is higher than that of Algorithm 5.

---

**Algorithm 5** TRPCA-ENR when $p \notin \{1/d, 2/d\}$

---

**Input:** $\mathcal{D}$, $k$, $q$, $\lambda_x$, $\lambda_e$, $\mu$, $t_{\max}$.

1: Initialize $\{\boldsymbol{x}_i^{(j)}\}_{i \in [k]}^{j \in [d]}$ with (11); let $t = 0$ and $\boldsymbol{\mathcal{E}} = \mathbf{0}$; let $\boldsymbol{Y}_0^{(1)} = \boldsymbol{X}_0^{(1)}$ and $\boldsymbol{Z}^{(1)} = \mathbf{0}$.

2: **repeat**

3:   $t \leftarrow t + 1$.

4:   Compute $\boldsymbol{X}_t^{(1)}$ by (17).

5:   Compute $\boldsymbol{Y}_t^{(1)}$ by Algorithm 1.

6:   Update $\boldsymbol{Z}^{(1)}$ by (19).

7:   **for** $j = 2, 3, \ldots, d$ **do**

8:     Compute $\boldsymbol{X}_t^{(j)}$ by $\boldsymbol{X}_t^{(j)} = (\boldsymbol{D}_{(j)} - \boldsymbol{E}_{(j)})[(\boldsymbol{X}^{(i)})^{\odot_{i \neq j}}]\Big([(\boldsymbol{X}^{(i)})^{\odot_{i \neq j}}]^T[(\boldsymbol{X}^{(i)})^{\odot_{i \neq j}}] + \lambda_x \boldsymbol{I}\Big)^{-1}$.

9:   **end for**

10:   Compute $\boldsymbol{\mathcal{E}}_t$ by (20).

11: **until** converged or $t = t_{\max}$

**Output:** $\hat{\boldsymbol{\mathcal{T}}} = \boldsymbol{\mathcal{I}} \times_1 \boldsymbol{X}_t^{(1)} \times_2 \boldsymbol{X}_t^{(2)} \ldots \times_d \boldsymbol{X}_t^{(d)}$.

---

# C  More experimental results

## C.1  Theoretical bound v.s. empirical error

We apply LRTC-ENR (with symmetric regularizers) to the synthetic tensors used in Figure 1 and Figure 2, where we let the noise-signal ratio be 0.2. We let $\tilde{\Delta} = \left(\frac{1}{|\bar{\Omega}|}\|\mathcal{P}_{\bar{\Omega}}(\boldsymbol{\mathcal{D}} - \boldsymbol{\mathcal{X}})\|_F^2 - \frac{1}{|\Omega|}\|\mathcal{P}_{\Omega}(\boldsymbol{\mathcal{D}} - \boldsymbol{\mathcal{X}})\|_F^2\right)/\varepsilon^2$, where $\varepsilon = \max\{\|\boldsymbol{\mathcal{D}}\|_\infty, \|\boldsymbol{\mathcal{X}}\|_\infty\}$. Now in Figure 13, we compare $\tilde{\Delta}$ with the theoretical error upper bound $\Delta$ defined by (6). $\tilde{\Delta}$ is in log scale for better visualization. We see the empirical reconstruction error is much less than the theoretical error (upper) bound. One reason is that the bound involves $\varepsilon^2$, which is much larger (often 100 times) than the average of square entries of $\boldsymbol{\mathcal{D}}$ and $\boldsymbol{\mathcal{X}}$.

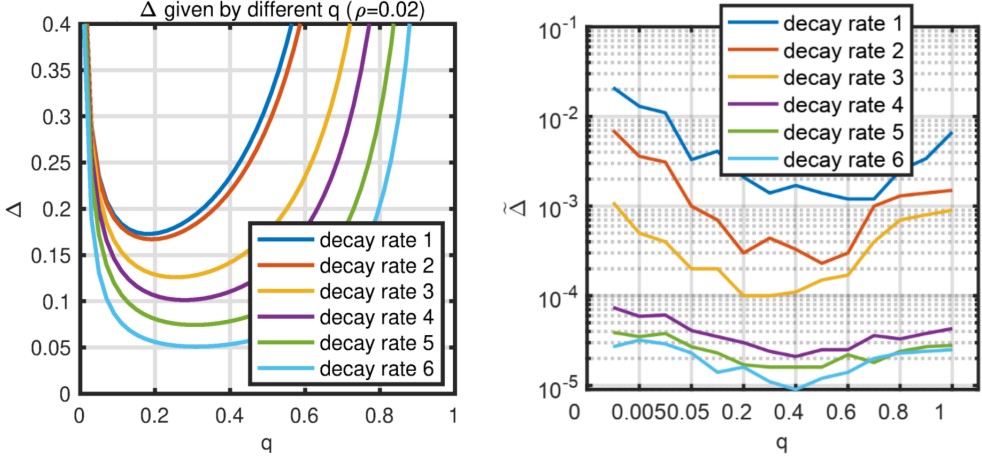

Figure 13: Left: theoretical upper bound $\Delta$ (Theorem 6). Right: empirical reconstruction error $\tilde{\Delta}$. The sampling rate $\rho$ is 0.02.

## C.2  Iterative performance of LRTC-ENR

Figure 14 shows the value of the objective function and relative recovery error of LRTC-ENR in each iteration on the synthetic data used in Section 5.1.1. We see that the optimization converged quickly and the relative recovery error decreased when the iteration number increased.

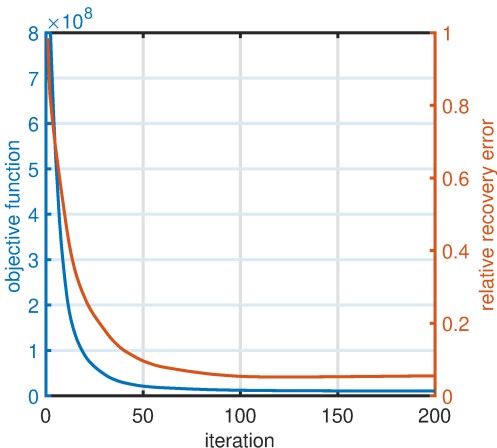

Figure 14: Iterative performance of LRTC-ENR ($p = 1/3$ (Yang et al., 2016))

### C.3 LRTC-ENR on higher-order tensors

We compare LRTC-ENR with HaLRTC, KBR-TC, and BCPF on a synthetic 4-order tensor of size 30×30×30×30 and rank 50. The relative recovery errors (average of ten trials) are reported in Table 4. LRTC-ENR ($p = 1/3$) outperformed the baselines.

Table 4: Tensor completion on 4-order tensor of size 30×30×30×30. The data generating model is similar to the one used in Section 5.1.1. In LRTC-ENR, we set $p = 1/3$ (Yang et al., 2016).

| methods | missing rate | |
|---|---|---|
| | 0.7 | 0.9 |
| HaLRTC | 0.98 | 0.99 |
| KBR-TC | 0.34 | 0.67 |
| BCPF | 0.11 | 0.21 |
| LRTC-ENR | **0.03** | **0.05** |

## D Proof of the theorems

### D.1 Proof of Theorem 1

*Proof.* Let $\boldsymbol{x}_i^{(j)} = \alpha_{ij}\bar{\boldsymbol{x}}_i^{(j)}$, where $\|\bar{\boldsymbol{x}}_i^{(j)}\| = 1$. Then $\prod_{j=1}^d \|\boldsymbol{x}_i^{(j)}\| = \prod_{j=1}^d \alpha_{ij} = |\lambda_i|$. We have

$$\frac{1}{d}\sum_{i=1}^k \sum_{j=1}^d \|\boldsymbol{x}_i^{(j)}\|^q \geq \sum_{i=1}^k \left(\prod_{j=1}^d \|\boldsymbol{x}_i^{(j)}\|^q\right)^{1/d}$$
$$= \sum_{i=1}^k \prod_{j=1}^d \|\boldsymbol{x}_i^{(j)}\|^{q/d} = \sum_{i=1}^k |\lambda_i|^{q/d},$$

in which the inequality holds according to the AM-GM inequality. When $\alpha_{i1} = \alpha_{i2} = \ldots \alpha_{id} = |\lambda_i|^{1/d}$, the equality of AM-GM inequality is true. Replacing $q$ with $pd$, we have

$$\frac{1}{d}\sum_{i=1}^k \sum_{j=1}^d \|\boldsymbol{x}_i^{(j)}\|^{pd} \geq \sum_{i=1}^k |\lambda_i|^p \geq \|\boldsymbol{\mathcal{X}}\|_{S_p}^p.$$

$\square$

### D.2 Proof of Theorem 2

*Proof.* (1) We have

$$\sum_{i=1}^{k}\Big(\frac{1}{q}\|\boldsymbol{x}_i^{(1)}\|^q + \sum_{j=2}^{d}\|\boldsymbol{x}_i^{(j)}\|\Big) = \sum_{i=1}^{k}\Big(\sum_{l=1}^{1/q}\|\boldsymbol{x}_i^{(1)}\|^q + \sum_{j=2}^{d}\|\boldsymbol{x}_i^{(j)}\|\Big)$$

$$\geq \sum_{i=1}^{k}(\frac{1}{q}+d-1)\left(\prod_{l=1}^{1/q}\|\boldsymbol{x}_i^{(1)}\|^q \prod_{j=2}^{d}\|\boldsymbol{x}_i^{(j)}\|\right)^{\frac{q}{1+qd-q}}$$

$$= \sum_{i=1}^{k}(\frac{1}{q}+d-1)\left(\prod_{j=1}^{d}\|\boldsymbol{x}_i^{(j)}\|\right)^{\frac{q}{1+qd-q}}$$

$$= \frac{1+qd-q}{q}\sum_{i=1}^{k}|\lambda_i|^{\frac{q}{1+qd-q}} \geq \frac{1+qd-q}{q}\|\boldsymbol{\mathcal{X}}\|_{S_{q/(1+qd-q)}}^{q/(1+qd-q)}.$$

(2) We have

$$\sum_{i=1}^{k}\Big(\frac{2}{q}\|\boldsymbol{x}_i^{(1)}\|^q + \sum_{j=2}^{d}\|\boldsymbol{x}_i^{(j)}\|^2\Big) = \sum_{i=1}^{k}\Big(\sum_{l=1}^{2/q}\|\boldsymbol{x}_i^{(1)}\|^q + \sum_{j=2}^{d}\|\boldsymbol{x}_i^{(j)}\|^2\Big)$$

$$\geq \sum_{i=1}^{k}(\frac{2}{q}+d-1)\left(\prod_{l=1}^{2/q}\|\boldsymbol{x}_i^{(1)}\|^q \prod_{j=2}^{d}\|\boldsymbol{x}_i^{(j)}\|^2\right)^{\frac{q}{2+qd-q}}$$

$$= \sum_{i=1}^{k}(\frac{2}{q}+d-1)\left(\prod_{j=1}^{d}\|\boldsymbol{x}_i^{(j)}\|^2\right)^{\frac{q}{2+qd-q}}$$

$$= \frac{2+qd-q}{q}\sum_{i=1}^{k}|\lambda_i|^{\frac{2q}{2+qd-q}} \geq \frac{2+qd-q}{q}\|\boldsymbol{\mathcal{X}}\|_{S_{2q/(2+qd-q)}}^{2q/(2+qd-q)}.$$

The equality in the second row of formula (1) (also (2)) holds when all terms in the parentheses are equal, for each $i$. $\qquad\square$

### D.3 Proof of the last two in Table 1

*Proof.* Recall the definition of $\lambda_1, \ldots, \lambda_k$. We have

$$\frac{16^{1/5}}{5}\sum_{i=1}^{k}\Big(\|\boldsymbol{x}_i^{(1)}\|^2 + \|\boldsymbol{x}_i^{(2)}\| + \|\boldsymbol{x}_i^{(3)}\|\Big)$$

$$= \frac{16^{1/5}}{5}\sum_{i=1}^{k}\Big(\|\boldsymbol{x}_i^{(1)}\|^2 + \frac{1}{2}\|\boldsymbol{x}_i^{(2)}\| + \frac{1}{2}\|\boldsymbol{x}_i^{(2)}\| + \frac{1}{2}\|\boldsymbol{x}_i^{(3)}\| + \frac{1}{2}\|\boldsymbol{x}_i^{(3)}\|\Big)$$

$$\geq \frac{16^{1/5}}{5}\sum_{i=1}^{k}5\left(\frac{1}{16}\|\boldsymbol{x}_i^{(1)}\|^2\|\boldsymbol{x}_i^{(2)}\|\|\boldsymbol{x}_i^{(2)}\|\|\boldsymbol{x}_i^{(3)}\|\|\boldsymbol{x}_i^{(3)}\|\right)^{1/5}$$

$$= \sum_{i=1}^{k}\Big(\|\boldsymbol{x}_i^{(1)}\|^2\|\boldsymbol{x}_i^{(2)}\|^2\|\boldsymbol{x}_i^{(3)}\|^2\Big)^{1/5}$$

$$= \sum_{i=1}^{k}|\lambda_i|^{2/5} \geq \|\boldsymbol{\mathcal{X}}\|_{S_{2/5}}^{2/5}.$$

Similarly, we have

$$
\frac{81^{1/7}}{7} \sum_{i=1}^{k} \left( \|\boldsymbol{x}_i^{(1)}\|^3 + \|\boldsymbol{x}_i^{(2)}\| + \|\boldsymbol{x}_i^{(3)}\| \right)
$$

$$
= \frac{81^{1/7}}{7} \sum_{i=1}^{k} \left( \|\boldsymbol{x}_i^{(1)}\|^3 + \sum_{j=1}^{3} \left( \frac{1}{3}\|\boldsymbol{x}_i^{(2)}\| + \frac{1}{3}\|\boldsymbol{x}_i^{(3)}\| \right) \right)
$$

$$
\geq \frac{81^{1/7}}{7} \sum_{i=1}^{k} 7 \left( \frac{1}{3^6} \|\boldsymbol{x}_i^{(1)}\|^3 \|\boldsymbol{x}_i^{(2)}\|^3 \|\boldsymbol{x}_i^{(3)}\|^3 \right)^{1/7}
$$

$$
= \sum_{i=1}^{k} \left( \|\boldsymbol{x}_i^{(1)}\| \|\boldsymbol{x}_i^{(2)}\| \|\boldsymbol{x}_i^{(3)}\| \right)^{2/7}
$$

$$
= \sum_{i=1}^{k} |\lambda_i|^{3/7} \geq \|\boldsymbol{\mathcal{X}}\|_{S_{3/7}}^{3/7}.
$$

$\square$

## D.4   Proof of Theorem 3

First, we give the following lemma, which is proved in Appendix E.1.

**Lemma 1.** *Let* $\mathcal{S}_{d,n,2}^{k,\perp} = \{\boldsymbol{\mathcal{X}} \in \mathcal{S}_{d,n}^{\perp} : \ \mathrm{rank}(\boldsymbol{\mathcal{X}}) \leq k, \ \|\boldsymbol{\mathcal{X}}\|_{S_2^{\perp}} \leq 1\}$. *Then the covering numbers of* $\mathcal{S}_{d,n,2}^{k,\perp}$ *with respect to* $\|\cdot\|_{S_2^{\perp}}$ *satisfy*

$$
\log \mathcal{N}(\mathcal{S}_{d,n,2}^{k,\perp}, \|\cdot\|_{S_2^{\perp}}, \epsilon) \leq \left( \frac{c(d+1)}{\epsilon} \right)^{dk(n-(k+1)/2)+k},
$$

*where* $c > 0$ *is a universal constant.*

*Proof.* We follow the idea of Theorem 4.3 of (Hinrichs et al., 2017) and analyze the entropy number first. According to the monotonicity of dyadic entropy numbers, it is enough to let $n = 2^\alpha$ and $\eta = 2^\alpha \cdot 2^\beta$, where $1 \leq \alpha < \beta$ are natural numbers. We have $n/\eta = 2^{-\beta}$. Let $\boldsymbol{\mathcal{X}} \in \mathcal{S}_{d,n,p}^{\perp}$ with $\psi = 1$. The orthogonal CP decomposition of $\boldsymbol{\mathcal{X}}$ is denoted by $\boldsymbol{\mathcal{X}} = \sum_{i=1}^{n} s_i \boldsymbol{u}_i^{(1)} \circ \boldsymbol{u}_i^{(2)} \cdots \circ \boldsymbol{u}_i^{(d)}$ and $s_1 \geq s_2 \geq \cdots \geq s_n$. We define

$$
\boldsymbol{\mathcal{X}}_1 := \sum_{i=1}^{n} \bar{s}_{1i} \boldsymbol{u}_i^{(1)} \circ \boldsymbol{u}_i^{(2)} \cdots \circ \boldsymbol{u}_i^{(d)}, \quad \bar{\boldsymbol{s}}_1 = (s_1, 0, \dots, 0)^\top, \tag{22}
$$

and, for $j = 2, 3, \dots, \beta$,

$$
\boldsymbol{\mathcal{X}}_j = \sum_{i=1}^{n} \bar{s}_{ji} \boldsymbol{u}_i^{(1)} \circ \boldsymbol{u}_i^{(2)} \cdots \circ \boldsymbol{u}_i^{(d)}, \quad \bar{\boldsymbol{s}}_j = (0, \dots, 0, \overbrace{s_{2^{j-1}}, \dots, s_{2^j-1}}^{2^{j-1}}, 0, \dots, 0)^\top. \tag{23}
$$

It follows that $\mathrm{rank}(\boldsymbol{\mathcal{X}}_j) \leq 2^{j-1}$, $j \in [\beta]$. Then we have $\|\boldsymbol{\mathcal{X}}_1\|_{S_p^{\perp}} \leq 1$, and $j = 2, 3, \dots, \beta$,

$$
\|\boldsymbol{\mathcal{X}}_j\|_{S_q^{\perp}} = \left( \sum_{t=2^{j-1}}^{2^j-1} s_t^q \right)^{1/q} \leq \left( 2^{j-1} s_{2^{j-1}}^q \right)^{1/q} = 2^{(j-1)/q} \left( s_{2^{j-1}}^p \right)^{1/p}
$$

$$
\leq 2^{(j-1)/q} \left( \frac{1}{2^{j-1}} \sum_{t=1}^{2^j-1} s_t^p \right)^{1/p} \leq 2^{(j-1)(1/q-1/p)}, \tag{24}
$$

where the last inequality holds because of $\sum_{t=1}^{2^{j-1}} s_t^p \leq 1$. Now we can decompose $\boldsymbol{\mathcal{X}}$ as

$$\boldsymbol{\mathcal{X}} = \boldsymbol{\mathcal{X}}_1 + \boldsymbol{\mathcal{X}}_2 + \cdots + \boldsymbol{\mathcal{X}}_\beta + \bar{\boldsymbol{\mathcal{X}}}. \tag{25}$$

Suppose $q \geq p$, we have

$$
\begin{aligned}
\|\bar{\boldsymbol{\mathcal{X}}}\|_{S_q^\perp}^q &= \sum_{t=2^\beta}^{\tilde{n}} s_t^q = \sum_{t=2^\beta}^{\tilde{n}} s_t^p s_t^{q-p} \leq s_{2^\beta}^{q-p} \sum_{t=2^\beta}^{\tilde{n}} s_t^p \\
&\leq \left( \frac{1}{2^\beta} \sum_{t=1}^{2^\beta} s_{2^\beta}^p \right)^{\frac{q-p}{p}} \left( \sum_{t=2^\beta}^{\tilde{n}} s_t^p \right) \\
&\leq 2^{-\beta(q-p)/p} \|\boldsymbol{\mathcal{X}}\|_{S_p^\perp}^{q-p} \|\boldsymbol{\mathcal{X}}\|_{S_p^\perp}^p = 2^{-\beta(q-p)/p} \|\boldsymbol{\mathcal{X}}\|_{S_p^\perp}^q \leq 2^{-\beta(q-p)/p}.
\end{aligned}
\tag{26}
$$

It follows that

$$\left\| \boldsymbol{\mathcal{X}} - \sum_{j=1}^\beta \boldsymbol{\mathcal{X}}_j \right\|_{S_q^\perp} = \|\bar{\boldsymbol{\mathcal{X}}}\|_{S_q^\perp} \leq 2^{\beta(1/q-1/p)}.$$

We focus on the case $q = 2$. We see that $2^{(j-1)(1/p-1/2)} \boldsymbol{\mathcal{X}}_j \in \mathcal{S}_{d,n,2}^{2^{j-1},\perp}$. Let $\mathcal{N}_j \subseteq \mathcal{S}_{d,n,2}^{2^{j-1},\perp}$ be an $\epsilon_j$-net, $j \in [\beta]$, and define

$$\mathcal{N} := \left\{ \sum_{j=1}^\beta 2^{-(j-1)(1/p-1/2)} \boldsymbol{\mathcal{Z}}_j : \ \boldsymbol{\mathcal{Z}}_j \in \mathcal{N}_j, \ j \in [\beta] \right\}.$$

Then $\mathcal{N}$ is an $\epsilon$-net of $\mathcal{S}_{d,n,p}^\perp$ with $\psi = 1$ in $\|\cdot\|_{\mathcal{S}_2^\perp}$, where

$$\epsilon = \sum_{j=1}^\beta 2^{-(j-1)(1/p-1/2)} \epsilon_j + 2^{\beta(1/2-1/p)}$$

and $|\mathcal{N}| = \prod_{j=1}^\beta |\mathcal{N}_j|$.

Now let $\epsilon_j = c2^{(j-\beta)(1/p-1/2+1)}$, where $c$ is the constant in Lemma 1. We have

$$
\begin{aligned}
\epsilon &= \sum_{j=1}^\beta c2^{(1-\beta)(1/p-1/2)} 2^{(j-\beta)} + 2^{\beta(1/2-1/p)} \\
&= c2^{(1-\beta)(1/p-1/2)} \sum_{j=1}^\beta 2^{(j-\beta)} + 2^{\beta(1/2-1/p)} \\
&\leq c'2^{-\beta(1/p-1/2)} \\
&= c' \left( \frac{n}{\eta} \right)^{1/p-1/2},
\end{aligned}
$$

where $c'$ is a constant. Then we have

$$\eta \leq n \left( \frac{c'}{\epsilon} \right)^{2p/(2-p)}. \tag{27}$$

Using Lemma 1 and the definition of $\mathcal{N}_j$, we obtain

$$
\begin{aligned}
\log|\mathcal{N}| = \log\prod_{j=1}^{\beta}|\mathcal{N}_j| &\leq \log\prod_{j=1}^{\beta}\left(\frac{c(d+1)}{c2^{(j-\beta)(1/p-1/2+1)}}\right)^{d2^{j-1}(n-(2^{j-1}+1)/2)+2^{j-1}} \\
&= \sum_{j=1}^{\beta}\left(d2^{j-1}\left(n-\frac{2^{j-1}+1}{2}\right)+2^{j-1}\right)\log\left((d+1)2^{(\beta-j)(1/p-1/2+1)}\right) \\
&\leq \sum_{j=1}^{\beta}dn2^{j-1}\left((\beta-j)(1/p-1/2+1)+\log(d+1)\right) \\
&\leq (1/p-1/2+1)d\log(d+1)n\sum_{j=1}^{\beta}2^{j-1}(\beta-j) \\
&= (1/p-1/2+1)d\log(d+1)n2^{\beta}\sum_{j=1}^{\beta}2^{j-\beta-1}(\beta-j) \\
&\leq (1/p-1/2+1)d\log(d+1)\eta \\
&\leq (1/p-1/2+1)nd\log(d+1)\left(\frac{c'}{\epsilon}\right)^{2p/(2-p)}.
\end{aligned}
\tag{28}
$$

Now using a general $\psi_p$ instead of 1, we finish the proof. $\qquad\square$

### D.5    Proof of Theorem 4

Before proving the theorem, we give the following lemma.

**Lemma 2.** *Let $\mathcal{S}$ be a set of $d$-order hyper-cubic tensors of side length $n$. Suppose the $\epsilon$-covering numbers of $\mathcal{S}$ with respect to the Frobenius norm are upper-bounded by $B$. Suppose $\max\{\|\boldsymbol{\mathcal{D}}\|_{\infty},\|\boldsymbol{\mathcal{X}}\|_{\infty}\}\leq\varepsilon$. Then the following inequality holds with probability at least $1-2n^{-d}$,*

$$
\sup_{\boldsymbol{\mathcal{X}}\in\mathcal{S}}\left|\frac{1}{\sqrt{n^d}}\|\boldsymbol{\mathcal{D}}-\boldsymbol{\mathcal{X}}\|_F-\frac{1}{\sqrt{|\Omega|}}\|\mathcal{P}_{\Omega}(\boldsymbol{\mathcal{D}}-\boldsymbol{\mathcal{X}})\|_F\right|\leq\frac{2\epsilon}{\sqrt{|\Omega|}}+2\varepsilon\left(\frac{d\log n+\log|B|)}{2|\Omega|}\right)^{1/4}.
$$

The proof of the lemma can be found in Appendix E.2. Now we prove Theorem 4 as follows.

*Proof.* According to Theorem 3, the covering numbers of $\mathcal{S}_p^{n,\perp}$ with respect to the Frobenius norm satisfy

$$
\log|\mathcal{S}_p^{n,\perp}|\leq (1/2+1/p)nd\left(\log(d+1)\right)\left(\frac{c\|\boldsymbol{\mathcal{X}}\|_{S_p^{\perp}}}{\epsilon}\right)^{2p/(2-p)}\triangleq\log B,
\tag{29}
$$

where $c > 0$ is a universal constant. Now substituting $\log B$ into Lemma 2 and letting $\epsilon = c\tau\varepsilon\sqrt{dn}$, we get

$$
\begin{aligned}
\sup_{\boldsymbol{\mathcal{X}}\in\mathcal{S}} & \left| \frac{1}{\sqrt{n^d}}\|\boldsymbol{\mathcal{D}} - \boldsymbol{\mathcal{X}}\|_F - \frac{1}{\sqrt{|\Omega|}}\|\mathcal{P}_\Omega(\boldsymbol{\mathcal{D}} - \boldsymbol{\mathcal{X}})\|_F \right| \\
& \leq \frac{2\epsilon}{\sqrt{|\Omega|}} + 2\varepsilon \left( \frac{d\log n + (\frac{1}{2} + \frac{1}{p})nd\left(\log(d+1)\right)\left(\frac{c\|\boldsymbol{\mathcal{X}}\|_{S_p^\perp}}{\epsilon}\right)^{2p/(2-p)}}{2|\Omega|} \right)^{1/4} \\
& = \frac{2c\tau\varepsilon\sqrt{dn}}{\sqrt{|\Omega|}} + 2\varepsilon \left( \frac{d\log n + (\frac{1}{2} + \frac{1}{p})nd\left(\log(d+1)\right)\left(\frac{\|\boldsymbol{\mathcal{X}}\|_{S_p^\perp}}{\varepsilon\tau\sqrt{dn}}\right)^{2p/(2-p)}}{2|\Omega|} \right)^{1/4} \\
& \leq c'\varepsilon \left( \frac{(\frac{1}{2} + \frac{1}{p})nd\left(\log(d+1)\right)\left(\frac{\|\boldsymbol{\mathcal{X}}\|_{S_p^\perp}}{\varepsilon\sqrt{dn}}\right)^{2p/(2-p)}}{|\Omega|} \right)^{1/4},
\end{aligned}
\tag{30}
$$

where we have let $\tau = 1$ and $c'$ is a universal constant. This finished the proof. $\qquad\square$

## D.6 Proof of Theorem 5

The following lemma (proved in Appendix E.3) will be used in the proof of the theorem.

**Lemma 3.** *Define $B_{2,q}^{n,k} := \{\boldsymbol{X} \in \mathbb{R}^{n\times k} : \|\boldsymbol{X}\|_{2,q} \leq 1\}$. Then the covering numbers of $B_{2,q}^{n,k}$ with respect to the Frobenius norm satisfy:*
*(a) $\log \mathcal{N}(\mathcal{B}_{2,q}^{n,k}, \|\cdot\|_F, \epsilon) \leq c_q(n + \log(ek))\epsilon^{-2q/(2-q)}$, when $0 < q < 1$,*
*(b) $\log \mathcal{N}(\mathcal{B}_{2,q}^{n,k}, \|\cdot\|_F, \epsilon) \leq \lceil nk^{2(q-1)/q}\epsilon^{-2}\rceil \log(2nk)$, when $q \geq 1$,*
*where $c_q = O\left(\frac{1}{q}\right)$.*

*Proof.* Let $\boldsymbol{\mathcal{X}} = \boldsymbol{\mathcal{I}} \times_1 \boldsymbol{X}^{(1)} \times_2 \cdots \times_d \boldsymbol{X}^{(d)}$ be the CP (or Tucker equivalently) decomposition of $\boldsymbol{\mathcal{X}} \in \mathcal{S}_{k,p}^n$, where $\boldsymbol{\mathcal{I}}$ is super-diagonal. tensor of 1s.

Let $\bar{\boldsymbol{\mathcal{X}}} = \boldsymbol{\mathcal{I}} \times_1 \bar{\boldsymbol{X}}^{(1)} \times_2 \cdots \times_d \bar{\boldsymbol{X}}^{(d)}$ and denote $\varsigma = \|\boldsymbol{\mathcal{C}}\|_F$. Let $\|\boldsymbol{X}^{(j)} - \bar{\boldsymbol{X}}^{(j)}\|_F \leq \frac{\epsilon}{d} \prod_{i \neq j} \gamma_j^{-1}$, $j \in [d]$. We have

$$
\begin{aligned}
&\|\boldsymbol{\mathcal{X}} - \bar{\boldsymbol{\mathcal{X}}}\|_F \\
=&\|\boldsymbol{\mathcal{I}} \times_1 \boldsymbol{X}^{(1)} \times_2 \ldots \times_d \boldsymbol{X}^{(d)} - \boldsymbol{\mathcal{I}} \times_1 \bar{\boldsymbol{X}}^{(1)} \times_2 \ldots \times_d \bar{\boldsymbol{X}}^{(d)}\|_F \\
=&\|\boldsymbol{\mathcal{I}} \times_1 \boldsymbol{X}^{(1)} \times_2 \ldots \times_d \boldsymbol{X}^{(d)} \pm \boldsymbol{\mathcal{I}} \times_1 \boldsymbol{X}^{(1)} \times_2 \ldots \times_d \bar{\boldsymbol{X}}^{(d)} \\
&\pm \boldsymbol{\mathcal{I}} \times_1 \boldsymbol{X}^{(1)} \times_2 \ldots \times_{d-1} \bar{\boldsymbol{X}}^{(d-1)} \times_d \bar{\boldsymbol{X}}^{(d)} \\
&\pm \ldots \pm \boldsymbol{\mathcal{I}} \times_1 \bar{\boldsymbol{X}}^{(1)} \times_2 \ldots \times_{d-1} \bar{\boldsymbol{X}}^{(d-1)} \times_d \bar{\boldsymbol{X}}^{(d)} \\
&- \boldsymbol{\mathcal{I}} \times_1 \bar{\boldsymbol{X}}^{(1)} \times_2 \ldots \times_d \bar{\boldsymbol{X}}^{(d)}\|_F \\
\leq&\|\boldsymbol{\mathcal{I}} \times_1 \boldsymbol{X}^{(1)} \times_2 \ldots \times_d (\boldsymbol{X}^{(d)} - \bar{\boldsymbol{X}}^{(d)})\|_F \\
&+ \|\boldsymbol{\mathcal{I}} \times_1 \boldsymbol{X}^{(1)} \times_2 \ldots \times_{d-1} (\boldsymbol{X}^{(d-1)} - \bar{\boldsymbol{X}}^{(d-1)}) \times_d \bar{\boldsymbol{X}}^{(d)}\|_F \\
&+ \ldots + \|\boldsymbol{\mathcal{I}} \times_1 (\boldsymbol{X}^{(1)} - \bar{\boldsymbol{X}}^{(1)}) \times_2 \bar{\boldsymbol{X}}^{(2)} \ldots \times_d \bar{\boldsymbol{X}}^{(d)}\|_F \\
\leq&\|\boldsymbol{\mathcal{I}}\|_{op}\|\boldsymbol{X}^{(1)}\|_{op} \ldots \|\boldsymbol{X}^{(d-1)}\|_{op}\|\boldsymbol{X}^{(d)} - \bar{\boldsymbol{X}}^{(d)}\|_F \\
&+ \|\boldsymbol{\mathcal{I}}\|_{op}\|\boldsymbol{X}^{(1)}\|_{op} \ldots \|\boldsymbol{X}^{(d-1)} - \bar{\boldsymbol{X}}^{(d-1)}\|_F\|\bar{\boldsymbol{X}}^{(d)}\|_{op} \\
&+ \ldots + \|\boldsymbol{\mathcal{I}}\|_{op}\|\boldsymbol{X}^{(1)} - \bar{\boldsymbol{X}}^{(1)}\|_F\|\bar{\boldsymbol{X}}^{(2)}\|_{op} \ldots \|\bar{\boldsymbol{X}}^{(d-1)}\|_{op}\|\bar{\boldsymbol{X}}^{(d)}\|_{op} \\
\leq&\frac{\epsilon}{d} + \frac{\epsilon}{d} + \cdots + \frac{\epsilon}{d} \\
=&\epsilon.
\end{aligned}
$$

Denote $\phi = \prod_{j=1}^d \gamma_j$. When $0 < q < 1$, the the covering number of $\mathcal{S}_{d,n,q}^k$ can be bounded as

$$
\mathcal{N}(\mathcal{S}_{d,n,q}^k, \|\cdot\|_F, \epsilon) \leq \prod_{i=1}^d \exp\left(c_q(n + \log(ek))\left(\frac{d\phi\alpha_q^{(j)}\gamma_j^{-1}}{\epsilon}\right)^{2q/(2-q)}\right),
$$

where $c_q$ is a universal constant. It follows that

$$
\log\mathcal{N}(\mathcal{S}_{d,n,q}^k, \|\cdot\|_F, \epsilon) \leq \frac{c(n + \log(ek))}{q}\sum_{j=1}^d \left(\frac{d\phi\alpha_q^{(j)}\gamma_j^{-1}}{\epsilon}\right)^{2q/(2-q)},
$$

where $c$ is a constant.

When $q \geq 1$, the the covering number of $\mathcal{S}_{d,n,q}^k$ can be bounded as

$$
\mathcal{N}(\mathcal{S}_{d,n,q}^k, \|\cdot\|_F, \epsilon) \leq \prod_{i=1}^d \exp\left(c' n k^{2(q-1)/q} \log(2nk)\left(\frac{d\phi\alpha_q^{(j)}\gamma_j^{-1}}{\epsilon}\right)^2\right),
$$

where we have converted the *ceil* operation into multiplying a constant $c'$ for simplicity. It follows that

$$
\log\mathcal{N}(\mathcal{S}_{d,n,q}^k, \|\cdot\|_F, \epsilon) \leq c' n k^{2(q-1)/q} \log(2nk)\sum_{j=1}^d \left(\frac{d\phi\alpha_q^{(j)}\gamma_j^{-1}}{\epsilon}\right)^2.
$$

$\square$

## D.7 Proof for Theorem 6

The following lemma provides a sample complexity bound for transductive learning.

**Lemma 4** (Corollary 1 of (El-Yaniv & Pechyony, 2009), reformulated). *Let $\mathcal{F}$ be a fixed hypothesis set and suppose $\sup_{i,j|\boldsymbol{X}\in\mathcal{H}} |\ell(Y_{ij}, X_{ij})| \leq \tau_\ell$. Suppose a fixed set $S$ of distinct indices is uniformly and randomly*

*split to two subsets $S_{\text{train}}$ and $S_{\text{test}}$, where $|S_{\text{test}}| > |S_{\text{train}}|$. Then with probability at least $1 - \delta$ over the random split, we have*

$$\frac{1}{|S_{\text{test}}|} \sum_{(i,j)\in S_{\text{test}}} \ell\left(Y_{ij}, X_{ij}\right) \leq \frac{1}{|S_{\text{train}}|} \sum_{(i,j)\in S_{\text{train}}} \ell\left(Y_{ij}, X_{ij}\right) + \frac{(|S_{\text{train}}| + |S_{\text{test}}|)^2}{|S_{\text{train}}||S_{\text{test}}|} \mathcal{R}_S(\ell \circ \mathcal{F})$$

$$+ \frac{11\tau_\ell\left(|S_{\text{train}}| + |S_{\text{test}}|\right)}{\sqrt{|S_{\text{train}}|}|S_{\text{test}}|} + 3\tau_\ell \sqrt{\frac{(|S_{\text{train}}| + |S_{\text{test}}|)}{|S_{\text{train}}||S_{\text{test}}|} \log \frac{1}{\delta}}. \tag{31}$$

Before proof, we give the following lemma, which is a variant of the Dudley entropy integral bound on Rademacher complexity.

**Lemma 5** (Theorem 3 of (Schreuder, 2020)). *Let $\mathcal{F} \subset \{f : \mathcal{X} \mapsto \mathbb{R}\}$ be any class of measurable functions containing the uniformly zero function and let $S_n(\mathcal{F}) = \sup_{f \in \mathcal{F}} \|f\|_{L_2(P_n)}$. Then*

$$\mathcal{R}_n(\mathcal{F}) \leq \inf_{\tau > 0} \left(4\tau + \frac{12}{\sqrt{n}} \int_\tau^{S_n(\mathcal{F})} \sqrt{\log \mathcal{N}(\mathcal{F}, L_2(P_n), \zeta)} \mathrm{d}\zeta\right). \tag{32}$$

In the lemma, $\|f\|_{L_2(P_n)}$ is defined as $\|f\|_{L_2(P_n)}^2 = \frac{1}{n} \sum_{i=1}^n f(X_i)^2$, which means

$$\mathcal{N}(\mathcal{F}, L_2(P_n), \zeta) = \mathcal{N}(\mathcal{F}, \|\cdot\|_F, \sqrt{n}\zeta). \tag{33}$$

Then we have

$$\mathcal{R}_n(\mathcal{F}) \leq \inf_{\tau > 0} \left(4\tau + \frac{12}{\sqrt{n}} \int_\tau^{S_n(\mathcal{F})} \sqrt{\log \mathcal{N}(\mathcal{F}, \|\cdot\|_F, \sqrt{n}\zeta)} \mathrm{d}\zeta\right)$$

$$= \inf_{\tau > 0} \left(\frac{4\tau}{\sqrt{n}} + \frac{12}{n} \int_\tau^{S_n(\mathcal{F})\sqrt{n}} \sqrt{\log \mathcal{N}(\mathcal{F}, \|\cdot\|_F, \epsilon)} \mathrm{d}\epsilon\right). \tag{34}$$

Now we use Theorem 5 and (34) to obtain the Rademacher complexity of the tensor decomposition model in LRTC-ENR. When $0 < q < 1$, we have

$$\mathcal{R}_{|\Omega|}(\mathcal{F}) \leq \inf_{\tau > 0} \left(\frac{4\tau}{\sqrt{|\Omega|}} + \frac{12}{|\Omega|} \int_\tau^{\varepsilon\sqrt{|\Omega|}} \sqrt{\frac{c(n + \log(ek))}{q} \sum_{j=1}^d \left(\frac{d\phi\alpha_q^{(j)}}{\gamma_j \epsilon}\right)^{\frac{2q}{2-q}}} \mathrm{d}\epsilon\right)$$

$$= \inf_{\tau > 0} \left(\frac{4\tau}{\sqrt{|\Omega|}} + \frac{12}{|\Omega|} \sqrt{\frac{c(n + \log(ek))}{q} \sum_{j=1}^d \left(\frac{d\phi\alpha_q^{(j)}}{\gamma_j}\right)^{\frac{2q}{2-q}}} \int_\tau^{\varepsilon\sqrt{|\Omega|}} \epsilon^{-\frac{q}{2-q}} \mathrm{d}\epsilon\right)$$

$$= \inf_{\tau > 0} \left(\frac{4\tau}{\sqrt{|\Omega|}} + \frac{12}{|\Omega|} \sqrt{\frac{c(n + \log(ek))}{q} \sum_{j=1}^d \left(\frac{d\phi\alpha_q^{(j)}}{\gamma_j}\right)^{\frac{2q}{2-q}} \left(\frac{2-q}{2-2q}\right) \left((\varepsilon\sqrt{|\Omega|})^{\frac{2-2q}{2-q}} - \tau^{\frac{2-2q}{2-q}}\right)}\right) \tag{35}$$

$$\leq \frac{4\varepsilon}{|\Omega|} + \frac{12}{|\Omega|} \sqrt{\frac{c(n + \log(ek))}{q} \sum_{j=1}^d \left(\frac{d\phi\alpha_q^{(j)}}{\gamma_j}\right)^{\frac{2q}{2-q}} \left(\frac{2-q}{2-2q}\right) (\varepsilon\sqrt{|\Omega|})^{\frac{2-2q}{2-q}}}$$

$$\leq \frac{c'}{|\Omega|} \sqrt{\frac{(n + \log(ek))(2-q)^2(\varepsilon\sqrt{|\Omega|})^{\frac{2-2q}{2-q}}}{q(2-2q)^2} \sum_{j=1}^d \left(\frac{d\phi\alpha_q^{(j)}}{\gamma_j}\right)^{\frac{2q}{2-q}}}$$

$$= \frac{c'}{|\Omega|} \sqrt{\frac{(n + \log(ek))(2-q)^2\varepsilon^2|\Omega|^{\frac{1-q}{2-q}}}{q(2-2q)^2} \sum_{j=1}^d \left(\frac{d\phi\alpha_q^{(j)}}{\varepsilon\gamma_j}\right)^{\frac{2q}{2-q}}}$$

where $c'$ is a suitable numerical constant.

When $q \geq 1$, we have

$$
\begin{aligned}
\mathcal{R}_{|\Omega|}(\mathcal{F}) &\leq \inf_{\tau > 0} \left( \frac{4\tau}{\sqrt{|\Omega|}} + \frac{12}{|\Omega|} \int_\tau^{\varepsilon\sqrt{|\Omega|}} \sqrt{cnk^{2(q-1)/q} \log(2nk) \sum_{j=1}^d \left( \frac{d\phi\alpha_q^{(j)} q^{(j)}}{\gamma_j \epsilon} \right)^2} \, \mathrm{d}\epsilon \right) \\
&= \inf_{\tau > 0} \left( \frac{4\tau}{\sqrt{|\Omega|}} + \frac{12}{|\Omega|} \sqrt{cnk^{2(q-1)/q} \log(2nk) \sum_{j=1}^d \left( \frac{d\phi\alpha_q^{(j)}}{\gamma_j} \right)^2} \int_\tau^{\varepsilon\sqrt{|\Omega|}} \epsilon^{-1} \mathrm{d}\epsilon \right) \\
&\leq \inf_{\tau > 0} \left( \frac{4\tau}{|\Omega|} + \frac{12}{|\Omega|} \sqrt{cnk^{2(q-1)/q} \log(2nk) \sum_{j=1}^d \left( \frac{d\phi\alpha_q^{(j)}}{\gamma_j} \right)^2} \log \frac{\varepsilon\sqrt{|\Omega|}}{\tau} \right) \\
&\leq \frac{c'}{|\Omega|} \sqrt{nk^{2(q-1)/q} \log(2nk) \sum_{j=1}^d \left( \frac{d\phi\alpha_q^{(j)}}{\gamma_j} \right)^2} \log |\Omega|,
\end{aligned}
\tag{36}
$$

where $c'$ is a suitable numerical constant.

Note that

$$
\mathcal{R}_{|\Omega|}(\ell \circ \mathcal{F}) \leq \eta_\ell \mathcal{R}_{|\Omega|}(\mathcal{F}),
\tag{37}
$$

where $\eta_\ell$ is the Lipschitz constant of function $\ell$. In this work, $\ell$ is the square loss, which means $\eta_\ell = 4\varepsilon$. Finally, integrating (35), (36), and (37) with Lemma 4 and renaming the constants, we get the desired results.

### D.8 Proof of Corollary 1

*Proof.* In Theorem 6, letting $\theta = \sum_{j=1}^d \left( \frac{d\phi\alpha_q^{(j)}}{\varepsilon\gamma_j} \right)^t$, we have

$$
\theta \leq \left( \frac{d\bar{\gamma}^{d-1}}{\varepsilon} \right)^t \sum_{j=1}^d (\alpha_q^{(j)})^t,
\tag{38}
$$

where $t = 2$ or $t = 2q/(2-q)$. Recall that in the proof for Theorem 1, the equality holds only when $\alpha_{i1} = \alpha_{i2} = \ldots \alpha_{id} = |\lambda_i|^{1/d}$, for all $i \in [k]$. Here we have $\alpha_q^{(j)} = (\sum_{i \in [k]} \alpha_{ij}^p)^{1/p}$. That means we can get $\|\boldsymbol{\mathcal{X}}\|_{S_{p/d}}$ only when $\alpha_q^{(1)} = \alpha_q^{(2)} = \cdots = \alpha_q^{(d)} = \|\boldsymbol{\mathcal{X}}\|_{S_{p/d}}^{1/d}$. Then we have

$$
\theta \leq d \left( \frac{d\bar{\gamma}^{d-1}}{\varepsilon} \right)^t \|\boldsymbol{\mathcal{X}}\|_{S_{q/d}}^{t/d}.
\tag{39}
$$

Now in Theorem 6, we have

$$
B_{\mathcal{R}} \leq \begin{cases} \frac{c_1\kappa}{|\Omega|} \sqrt{nk^{2(q-1)/q} \log(2nk) d \left( \frac{d\bar{\gamma}^{d-1}}{\varepsilon} \right)^2 \|\boldsymbol{\mathcal{X}}\|_{S_{q/d}}^{2/d} \log |\Omega|} & \text{if } q \geq 1 \\ \frac{c_2\kappa}{|\Omega|} \sqrt{\frac{(n+\log(ek))(2-q)^2 |\Omega|^{\frac{1-q}{2-q}}}{q(2-2q)^2} d \left( \frac{d\bar{\gamma}^{d-1}}{\varepsilon} \right)^{2q/(2-q)} \|\boldsymbol{\mathcal{X}}\|_{S_{q/d}}^{2q/(2-q)/d}} & \text{if } 0 < q < 1 \end{cases}
$$

Letting $q = pd$, we arrive at

$$
B_{\mathcal{R}} \leq \begin{cases} \frac{c_1\kappa}{|\Omega|} \sqrt{nk^{2(pd-1)/pd} \log(2nk) d \left( \frac{d\bar{\gamma}^{d-1}}{\varepsilon} \right)^2 \|\boldsymbol{\mathcal{X}}\|_{S_p}^{2/d} \log |\Omega|} & \text{if } p \geq 1/d \\ \frac{c_2\kappa}{|\Omega|} \sqrt{\frac{(n+\log(ek))(2-pd)^2}{pd(2-2pd)^2} d \left( \frac{d\bar{\gamma}^{d-1}}{\varepsilon} \right)^{2pd/(2-pd)} \|\boldsymbol{\mathcal{X}}\|_{S_p}^{2p/(2-pd)} |\Omega|^{\frac{1-pd}{4-2pd}}} & \text{if } 0 < p < 1/d \end{cases}
$$

Now rename $N_{\mathcal{R}}$ as the upper bound, we finish the proof. □

### D.9 Proof of Theorem 7

*Proof.* The following three lemmas will be used. Their proof are in Section E.

**Lemma 6.** *For any $\boldsymbol{z} \in \mathbb{R}_+^n$ and $\tau > 0$ for, the following inequality holds*

$$\|\boldsymbol{z}\|_1 \le \tau^{-p/2}\|\boldsymbol{z}\|_2\sqrt{\|\boldsymbol{z}\|_P^p} + \tau^{1-p}\|\boldsymbol{z}\|_p^p. \tag{40}$$

**Lemma 7.** *Suppose the entries of $\boldsymbol{\mathcal{N}} \in \mathbb{R}^{n^{\otimes d}}$ are drawn from $\mathcal{N}(0, \sigma^2)$ independently. Then the following inequality holds with probability at least $1 - 2n^{-d}$*

$$\|\boldsymbol{\mathcal{N}}\|_2 \le 2\sqrt{2}\sigma\sqrt{dn\log(5d) + d\log(n)}.$$

**Lemma 8.** *Suppose the entries of $\boldsymbol{\mathcal{N}} \in \mathbb{R}^{n^{\otimes d}}$ are drawn from $\mathcal{N}(0, \sigma^2)$ independently. Then the following inequality holds with probability at least $1 - 2n^{-d}$*

$$\|\boldsymbol{\mathcal{N}}\|_\infty \le 2\sigma\sqrt{d\log(n)}.$$

Let $\hat{\boldsymbol{\mathcal{X}}}, \hat{\boldsymbol{\mathcal{E}}}$ be the optimal solution of (10). We have

$$\|\boldsymbol{\mathcal{D}} - \hat{\boldsymbol{\mathcal{X}}} - \hat{\boldsymbol{\mathcal{E}}}\|_F^2 \le \|\boldsymbol{\mathcal{D}} - \boldsymbol{\mathcal{X}}^* - \boldsymbol{\mathcal{E}}^*\|_F^2. \tag{41}$$

Since $\boldsymbol{\mathcal{D}} = \boldsymbol{\mathcal{X}}^* + \boldsymbol{\mathcal{N}}^* + \boldsymbol{\mathcal{E}}^*$, we obtain

$$\|\boldsymbol{\mathcal{X}}^* + \boldsymbol{\mathcal{N}}^* + \boldsymbol{\mathcal{E}}^* - \hat{\boldsymbol{\mathcal{X}}} - \hat{\boldsymbol{\mathcal{E}}}\|_F^2 \le \|\boldsymbol{\mathcal{N}}^*\|_F^2. \tag{42}$$

It follows that

$$
\begin{aligned}
&\|\boldsymbol{\mathcal{X}}^* - \hat{\boldsymbol{\mathcal{X}}}\|_F^2 + \|\boldsymbol{\mathcal{E}}^* - \hat{\boldsymbol{\mathcal{E}}}\|_F^2 \\
&\le -2\langle\boldsymbol{\mathcal{X}}^* - \hat{\boldsymbol{\mathcal{X}}}, \boldsymbol{\mathcal{N}}^*\rangle - 2\langle\boldsymbol{\mathcal{X}}^* - \hat{\boldsymbol{\mathcal{X}}}, \boldsymbol{\mathcal{E}}^* - \hat{\boldsymbol{\mathcal{E}}}\rangle - 2\langle\boldsymbol{\mathcal{E}}^* - \hat{\boldsymbol{\mathcal{E}}}, \boldsymbol{\mathcal{N}}^*\rangle \\
&\le 2\|\boldsymbol{\mathcal{X}}^* - \hat{\boldsymbol{\mathcal{X}}}\|_{S_1^\perp}\|\boldsymbol{\mathcal{N}}^*\|_2 + 2\|\boldsymbol{\mathcal{E}}^* - \hat{\boldsymbol{\mathcal{E}}}\|_1 \left(\|\boldsymbol{\mathcal{X}}^* - \hat{\boldsymbol{\mathcal{X}}}\|_\infty + \|\boldsymbol{\mathcal{N}}^*\|_\infty\right).
\end{aligned}
\tag{43}
$$

Using Lemma 6, we obtain

$$
\begin{aligned}
\|\boldsymbol{\mathcal{X}}^* - \hat{\boldsymbol{\mathcal{X}}}\|_{S_1^\perp} &\le \tau_1^{-p/2}\|\boldsymbol{\mathcal{X}}^* - \hat{\boldsymbol{\mathcal{X}}}\|_F\sqrt{\|\boldsymbol{\mathcal{X}}^* - \hat{\boldsymbol{\mathcal{X}}}\|_{S_p^\perp}^p} + \tau_1^{1-p}\|\boldsymbol{\mathcal{X}}^* - \hat{\boldsymbol{\mathcal{X}}}\|_{S_p^\perp}^p \\
&\le \tau_1^{-p/2}\|\boldsymbol{\mathcal{X}}^* - \hat{\boldsymbol{\mathcal{X}}}\|_F\sqrt{2R_x^p} + 2\tau_1^{1-p}R_x^p
\end{aligned}
\tag{44}
$$

and

$$
\begin{aligned}
\|\boldsymbol{\mathcal{E}}^* - \hat{\boldsymbol{\mathcal{E}}}\|_1 &\le \tau_2^{-p'/2}\|\boldsymbol{\mathcal{E}}^* - \hat{\boldsymbol{\mathcal{E}}}\|_F\sqrt{\|\boldsymbol{\mathcal{E}}^* - \hat{\boldsymbol{\mathcal{E}}}\|_{p'}^{p'}} + \tau_2^{1-p'}\|\boldsymbol{\mathcal{E}}^* - \hat{\boldsymbol{\mathcal{E}}}\|_{p'}^{p'} \\
&\le \tau_2^{-p'/2}\|\boldsymbol{\mathcal{E}}^* - \hat{\boldsymbol{\mathcal{E}}}\|_F\sqrt{2R_e^{p'}} + 2\tau_2^{1-p'}R_e^{p'}
\end{aligned}
\tag{45}
$$

which hold for any $\tau_1 > 0$ and $\tau_2 > 0$. Substituting (44) and (45) into (43), we have

$$
\begin{aligned}
&\|\boldsymbol{\mathcal{X}}^* - \hat{\boldsymbol{\mathcal{X}}}\|_F^2 + \|\boldsymbol{\mathcal{E}}^* - \hat{\boldsymbol{\mathcal{E}}}\|_F^2 \\
&\le 2\tau_1^{-p/2}\sqrt{2R_x^p}\|\boldsymbol{\mathcal{N}}^*\|_2\|\boldsymbol{\mathcal{X}}^* - \hat{\boldsymbol{\mathcal{X}}}\|_F \\
&\quad + 2\tau_2^{-p'/2}\sqrt{2R_e^{p'}}\left(\|\boldsymbol{\mathcal{X}}^* - \hat{\boldsymbol{\mathcal{X}}}\|_\infty + \|\boldsymbol{\mathcal{N}}^*\|_\infty\right)\|\boldsymbol{\mathcal{E}}^* - \hat{\boldsymbol{\mathcal{E}}}\|_F \\
&\quad + 4\tau_1^{1-p}R_x^p\|\boldsymbol{\mathcal{N}}^*\|_2 + 4\tau_2^{1-p'}R_e^{p'}\left(\|\boldsymbol{\mathcal{X}}^* - \hat{\boldsymbol{\mathcal{X}}}\|_\infty + \|\boldsymbol{\mathcal{N}}^*\|_\infty\right).
\end{aligned}
\tag{46}
$$

For convenience, let

$$u = \|\boldsymbol{\mathcal{X}}^* - \hat{\boldsymbol{\mathcal{X}}}\|_F$$
$$v = \|\boldsymbol{\mathcal{E}}^* - \hat{\boldsymbol{\mathcal{E}}}\|_F$$
$$c_1 = 2\tau_1^{-p/2}\sqrt{2R_x^p}\|\boldsymbol{\mathcal{N}}^*\|_2$$
$$c_2 = 2\tau_2^{-p'/2}\sqrt{2R_e^{p'}}\left(\|\boldsymbol{\mathcal{X}}^* - \hat{\boldsymbol{\mathcal{X}}}\|_\infty + \|\boldsymbol{\mathcal{N}}^*\|_\infty\right)$$
$$c_3 = 4\tau_1^{1-p}R_x^p\|\boldsymbol{\mathcal{N}}^*\|_2 + 4\tau_2^{1-p'}R_e^{p'}\left(\|\boldsymbol{\mathcal{X}}^* - \hat{\boldsymbol{\mathcal{X}}}\|_\infty + \|\boldsymbol{\mathcal{N}}^*\|_\infty\right)$$
$$2\alpha \geq \|\boldsymbol{\mathcal{X}}^* - \hat{\boldsymbol{\mathcal{X}}}\|_\infty$$
$$\beta \geq \|\boldsymbol{\mathcal{N}}^*\|_\infty$$
$$\gamma \geq \|\boldsymbol{\mathcal{N}}^*\|_2$$

We rewrite (46) as

$$
\begin{aligned}
u^2 + v^2 &\leq c_1 u + c_2 v + c_3 \\
&\leq (c_1 + c_2)(u + v) + c_3 \\
&\leq (c_1 + c_2)\sqrt{2(u^2 + v^2)} + c_3.
\end{aligned}
\tag{47}
$$

Because $c_3 > 0$, the quadratic inequality of $\sqrt{u^2 + v^2}$ has non-empty solution. We have

$$
\max\left(0, \frac{\sqrt{2}(c_1 + c_2) - \sqrt{2(c_1 + c_2)^2 + 4c_3}}{2}\right) \leq \sqrt{u^2 + v^2} \leq \frac{\sqrt{2}(c_1 + c_2) + \sqrt{2(c_1 + c_2)^2 + 4c_3}}{2}.
\tag{48}
$$

Let $\tau_1 = \gamma$ and $\tau_2 = 2\alpha + \beta$. We have

$$
\begin{aligned}
\sqrt{u^2 + v^2} &\leq 2\gamma^{1-\frac{p}{2}}\sqrt{R_x^p} + 2(2\alpha + \beta)^{1-\frac{p'}{2}}\sqrt{R_e^{p'}} \\
&\quad + 2\sqrt{\left(\gamma^{1-\frac{p}{2}}\sqrt{R_x^p} + (2\alpha + \beta)^{1-\frac{p'}{2}}\sqrt{R_e^{p'}}\right)^2 + \gamma^{2-p}R_x^p + (2\alpha + \beta)^{2-p'}R_e^{p'}} \\
&\leq 2\gamma^{1-\frac{p}{2}}\sqrt{R_x^p} + 2(2\alpha + \beta)^{1-\frac{p'}{2}}\sqrt{R_e^{p'}} + 2\sqrt{2}\left(\gamma^{1-\frac{p}{2}}\sqrt{R_x^p} + (2\alpha + \beta)^{1-\frac{p'}{2}}\sqrt{R_e^{p'}}\right) \\
&= (2 + 2\sqrt{2})\left(\gamma^{1-\frac{p}{2}}\sqrt{R_x^p} + (2\alpha + \beta)^{1-\frac{p'}{2}}\sqrt{R_e^{p'}}\right).
\end{aligned}
\tag{49}
$$

It follows that

$$
\begin{aligned}
u^2 + v^2 &\leq (12 + 8\sqrt{2})\left(\gamma^{1-\frac{p}{2}}\sqrt{R_x^p} + (2\alpha + \beta)^{1-\frac{p'}{2}}\sqrt{R_e^{p'}}\right)^2 \\
&\leq (24 + 16\sqrt{2})\left(\gamma^{2-p}R_x^p + (2\alpha + \beta)^{2-p'}R_e^{p'}\right).
\end{aligned}
\tag{50}
$$

$$\square$$

Using Lemma 7 and Lemma 8 for $\gamma$ and $\beta$ respectively, we have

$$
\begin{aligned}
&\|\boldsymbol{\mathcal{X}}^* - \hat{\boldsymbol{\mathcal{X}}}\|_F^2 + \|\boldsymbol{\mathcal{E}}^* - \hat{\boldsymbol{\mathcal{E}}}\|_F^2 \\
&\leq (24 + 16\sqrt{2})\left(\left(2\sqrt{2}\sigma\sqrt{dn\log(5d) + d\log(n)}\right)^{2-p}R_x^p + \left(2\alpha + 2\sigma\sqrt{d\log(n)}\right)^{2-p'}R_e^{p'}\right)
\end{aligned}
\tag{51}
$$

with probability at least $1 - 4n^{-d}$. This finished the proof.

# E   Proof of the lemmas

## E.1   Proof of Lemma 1

Before proof, we restate Lemma 4.1 of (Hinrichs et al., 2017) here.

**Lemma 9** (Lemma 4.1 of (Hinrichs et al., 2017)). *Define $V_k^n = \{U \in \mathbb{R}^{n \times k} : U^\top U = I_k, \; k, n \in \mathbb{N}, \; k \leq n\}$. Let $0 < \epsilon < 1$. Then*

$$\mathcal{N}(V_k^n, \|\cdot\|_{op}, \epsilon) \leq \left(\frac{c}{\epsilon}\right)^{k(n-(k+1)/2)},$$

*where $c > 0$ is a universal constant.*

*Proof.* Let $\boldsymbol{\mathcal{X}} = \boldsymbol{\mathcal{C}} \times_1 \boldsymbol{X}^{(1)} \times_2 \cdots \times_d \boldsymbol{X}^{(d)}$ be the orthogonal CP (or Tucker equivalently) decomposition of $\boldsymbol{\mathcal{X}} \in \mathcal{S}_{d,n,2}^{k,\perp}$, where $\boldsymbol{\mathcal{C}}$ is super-diagonal. We have $\|\boldsymbol{\mathcal{C}}\|_{S_2^\perp} = \|\mathrm{diag}(\boldsymbol{\mathcal{C}})\|_2 \leq 1$. Then the covering number of $\boldsymbol{\mathcal{C}}$ with respective to $\|\cdot\|_{S_2^\perp}$ is equal to the covering number of $\mathrm{diag}(\boldsymbol{\mathcal{C}})$ with respect to $\|\cdot\|_2$, i.e., $|\mathcal{N}_{\boldsymbol{\mathcal{C}}}| \leq \left(3\epsilon^{-1}\right)^k$. According to Lemma 9, the covering number of $\boldsymbol{X}^{(j)}$ with respect to $\|\cdot\|_{op}$ satisfies $|\mathcal{N}_{\boldsymbol{X}^{(j)}}| \leq \left(c\epsilon^{-1}\right)^{k(n-(k+1)/2)}, \; j \in [d]$. We have

$$
\begin{aligned}
\|\boldsymbol{\mathcal{X}} - \bar{\boldsymbol{\mathcal{X}}}\|_{S_2^\perp} =& \|\boldsymbol{\mathcal{C}} \times_1 \boldsymbol{X}^{(1)} \times_2 \cdots \times_d \boldsymbol{X}^{(d)} - \bar{\boldsymbol{\mathcal{C}}} \times_1 \bar{\boldsymbol{X}}^{(1)} \times_2 \cdots \times_d \bar{\boldsymbol{X}}^{(d)}\|_{S_2^\perp} \\
=& \|\boldsymbol{\mathcal{C}} \times_1 \boldsymbol{X}^{(1)} \times_2 \cdots \times_d \boldsymbol{X}^{(d)} \pm \boldsymbol{\mathcal{C}} \times_1 \boldsymbol{X}^{(1)} \times_2 \cdots \times_d \bar{\boldsymbol{X}}^{(d)} \\
& \pm \boldsymbol{\mathcal{C}} \times_1 \boldsymbol{X}^{(1)} \times_2 \cdots \times_{d-1} \bar{\boldsymbol{X}}^{(d-1)} \times_d \bar{\boldsymbol{X}}^{(d)} \\
& \pm \cdots \pm \boldsymbol{\mathcal{C}} \times_1 \bar{\boldsymbol{X}}^{(1)} \times_2 \cdots \times_{d-1} \bar{\boldsymbol{X}}^{(d-1)} \times_d \bar{\boldsymbol{X}}^{(d)} - \bar{\boldsymbol{\mathcal{C}}} \times_1 \bar{\boldsymbol{X}}^{(1)} \times_2 \cdots \times_d \bar{\boldsymbol{X}}^{(d)}\|_{S_2^\perp} \\
\overset{(a)}{\leq}& \|\boldsymbol{\mathcal{C}} \times_1 \boldsymbol{X}^{(1)} \times_2 \cdots \times_d (\boldsymbol{X}^{(d)} - \bar{\boldsymbol{X}}^{(d)})\|_{S_2^\perp} \\
& + \|\boldsymbol{\mathcal{C}} \times_1 \boldsymbol{X}^{(1)} \times_2 \cdots \times_{d-1} (\boldsymbol{X}^{(d-1)} - \bar{\boldsymbol{X}}^{(d-1)}) \times_d \bar{\boldsymbol{X}}^{(d)}\|_{S_2^\perp} \\
& + \cdots + \|(\boldsymbol{\mathcal{C}} - \bar{\boldsymbol{\mathcal{C}}}) \times_1 \bar{\boldsymbol{X}}^{(1)} \times_2 \bar{\boldsymbol{X}}^{(2)} \cdots \times_d \bar{\boldsymbol{X}}^{(d)}\|_{S_2^\perp} \\
\overset{(b)}{\leq}& \|\boldsymbol{\mathcal{C}}\|_{S_2^\perp} \|\boldsymbol{X}^{(1)}\|_{op} \cdots \|\boldsymbol{X}^{(d-1)}\|_{op} \|\boldsymbol{X}^{(d)} - \bar{\boldsymbol{X}}^{(d)}\|_{op} \\
& + \|\boldsymbol{\mathcal{C}}\|_{S_2^\perp} \|\boldsymbol{X}^{(1)}\|_{op} \cdots \|\boldsymbol{X}^{(d-1)} - \bar{\boldsymbol{X}}^{(d-1)}\|_{op} \|\bar{\boldsymbol{X}}^{(d)}\|_{op} \\
& + \cdots + \|\boldsymbol{\mathcal{C}} - \bar{\boldsymbol{\mathcal{C}}}\|_{S_2^\perp} \|\bar{\boldsymbol{X}}^{(1)}\|_{op} \|\bar{\boldsymbol{X}}^{(2)}\|_{op} \cdots \|\bar{\boldsymbol{X}}^{(d-1)}\|_{op} \|\bar{\boldsymbol{X}}^{(d)}\|_{op} \\
=& \|\boldsymbol{\mathcal{C}}\|_{S_2^\perp} \|\boldsymbol{X}^{(d)} - \bar{\boldsymbol{X}}^{(d)}\|_{op} + \|\boldsymbol{\mathcal{C}}\|_{S_2^\perp} \|\boldsymbol{X}^{(d-1)} - \bar{\boldsymbol{X}}^{(d-1)}\|_{op} + \cdots \\
& + \|\boldsymbol{\mathcal{C}}\|_{S_2^\perp} \|\boldsymbol{X}^{(1)} - \bar{\boldsymbol{X}}^{(1)}\|_{op} + \|\boldsymbol{\mathcal{C}} - \bar{\boldsymbol{\mathcal{C}}}\|_{S_2^\perp}.
\end{aligned}
$$

Note that $(a)$ holds owing to the triangle inequality of norms and $(b)$ holds because the inequality $\|\boldsymbol{A}\boldsymbol{B}\|_{S_q} \leq \|\boldsymbol{A}\|_{op} \|\boldsymbol{B}\|_{S_q}$ (Bhatia, 2013) can be easily extended to orthogonally decomposable tensors. Now let $\|\boldsymbol{\mathcal{C}} - \bar{\boldsymbol{\mathcal{C}}}\|_{S_2^\perp} \leq \epsilon/(d+1)$, and $\|\boldsymbol{X}^{(j)} - \bar{\boldsymbol{X}}^{(j)}\|_{op} \leq \epsilon/(d+1), \; j \in [d]$. We arrive at

$$\|\boldsymbol{\mathcal{X}} - \bar{\boldsymbol{\mathcal{X}}}\|_{S_2^\perp} \leq \frac{\epsilon}{d+1} + \frac{\epsilon}{d+1} + \cdots + \frac{\epsilon}{d+1} \leq \epsilon.$$

Then the covering number of $\mathcal{S}_{d,n,2}^{k,\perp}$ can be bounded as

$$
\begin{aligned}
\mathcal{N}(\mathcal{S}_{d,n,2}^{k,\perp}, \|\cdot\|_{S_2^\perp}, \epsilon) &\leq \left(\frac{3(d+1)}{\epsilon}\right)^k \prod_{i=1}^d \left(\frac{c(d+1)}{\epsilon}\right)^{k(n-(k+1)/2)} \\
&\leq \left(\frac{c'(d+1)}{\epsilon}\right)^{dk(n-(k+1)/2)+k},
\end{aligned}
$$

where $c'$ is a universal constant. This finished the proof. □

### E.2 Proof of Lemma 2

*Proof.* For convenience, we define

$$\hat{h}(\boldsymbol{\mathcal{X}}) = \frac{1}{|\Omega|}\|\mathcal{P}_\Omega(\boldsymbol{\mathcal{D}} - \boldsymbol{\mathcal{X}})\|_F^2, \quad h(\boldsymbol{\mathcal{X}}) = \frac{1}{n^d}\|\boldsymbol{\mathcal{D}} - \boldsymbol{\mathcal{X}}\|_F^2.$$

According to the following lemma

**Lemma 10** ((Hoeffding inequality for sampling without replacement)**.** *Let $X_1, X_2, \ldots, Xs$ be a set of samples taken without replacement from a distribution $\{x_1, x_2, \ldots, x_N\}$ of mean $u$ and variance $\sigma^2$. Denote $a = \min_i x_i$ and $b = \max_i x_i$. Then*

$$\mathbb{P}\left[\left|\frac{1}{s}\sum_{i=1}^{s} X_i - u\right| \geq t\right] \leq 2\exp\left(-\frac{2st^2}{(1-(s-1)/N)(b-a)^2}\right).$$

we have

$$\mathbb{P}\left[|\hat{h} - h| \geq t\right] \leq 2\exp\left(-\frac{2|\Omega|t^2}{(1-(|\Omega|-1)/n^d)\varsigma^2}\right),$$

where $\varsigma = 4\varepsilon^2$. Using union bound for all $\bar{\boldsymbol{\mathcal{X}}} \in \mathcal{S}$ yields

$$\mathbb{P}\left[\sup_{\bar{\boldsymbol{\mathcal{X}}}\in\mathcal{S}} |\hat{h}(\bar{\boldsymbol{\mathcal{X}}}) - h(\bar{\boldsymbol{\mathcal{X}}})| \geq t\right] \leq 2|\mathcal{S}|\exp\left(-\frac{2|\Omega|t^2}{(1-(|\Omega|-1)/n^d)\varsigma^2}\right).$$

Or equivalently, with probability at least $1 - 2n^{-d}$,

$$\sup_{\bar{\boldsymbol{\mathcal{X}}}\in\mathcal{S}} |\hat{h}(\bar{\boldsymbol{\mathcal{X}}}) - h(\bar{\boldsymbol{\mathcal{X}}})| \leq \sqrt{\frac{\varsigma^2\log\left(|\mathcal{S}|n^d\right)}{2}\left(\frac{1}{|\Omega|} - \frac{1}{n^d} + \frac{1}{n^d|\Omega|}\right)}.$$

Then we have

$$g(\Omega) \triangleq \sup_{\bar{\boldsymbol{\mathcal{X}}}\in\mathcal{S}} |\hat{h}(\bar{\boldsymbol{\mathcal{X}}}) - h(\bar{\boldsymbol{\mathcal{X}}})|$$

$$\leq \sqrt{\frac{\varsigma^2}{2}\left(d\log n + \log|\mathcal{S}|\right)\left(\frac{1}{|\Omega|} - \frac{1}{n^d} + \frac{1}{n^d|\Omega|}\right)}.$$

Since $|\sqrt{u} - \sqrt{v}| \leq \sqrt{|u-v|}$ holds for any non-negative $u$ and $v$, we have

$$\sup_{\bar{\boldsymbol{\mathcal{X}}}\in\mathcal{S}} \left|\sqrt{\hat{h}(\bar{\boldsymbol{\mathcal{X}}})} - \sqrt{h(\bar{\boldsymbol{\mathcal{X}}})}\right| \leq \sqrt{g(\Omega)}.$$

Recall that $\epsilon \geq \|\boldsymbol{\mathcal{X}} - \bar{\boldsymbol{\mathcal{X}}}\|_F \geq \|\mathcal{P}(\boldsymbol{\mathcal{X}} - \bar{\boldsymbol{\mathcal{X}}})\|_F$, we have

$$\left|\sqrt{h(\boldsymbol{\mathcal{X}})} - \sqrt{h(\bar{\boldsymbol{\mathcal{X}}})}\right| = \frac{1}{\sqrt{n^d}}\left|\|\boldsymbol{\mathcal{D}} - \boldsymbol{\mathcal{X}}\|_F - \|\boldsymbol{\mathcal{D}} - \bar{\boldsymbol{\mathcal{X}}}\|_F\right| \leq \frac{\epsilon}{\sqrt{n^d}}$$

and

$$\left|\sqrt{\hat{h}(\boldsymbol{\mathcal{X}})} - \sqrt{\hat{h}(\bar{\boldsymbol{\mathcal{X}}})}\right| = \frac{1}{\sqrt{|\Omega|}}\left|\|\mathcal{P}_\Omega(\boldsymbol{\mathcal{D}} - \boldsymbol{\mathcal{X}})\|_F - \|\mathcal{P}_\Omega(\boldsymbol{\mathcal{D}} - \bar{\boldsymbol{\mathcal{X}}})\|_F\right| \leq \frac{\epsilon}{\sqrt{|\Omega|}}.$$

It follows that

$$\sup_{\boldsymbol{\mathcal{X}}\in\mathcal{S}} \left|\sqrt{\hat{h}(\boldsymbol{\mathcal{X}})} - \sqrt{h(\boldsymbol{\mathcal{X}})}\right|$$

$$\leq \sup_{\boldsymbol{\mathcal{X}}\in\mathcal{S}} \left|\sqrt{\hat{h}(\boldsymbol{\mathcal{X}})} - \sqrt{\hat{h}(\bar{\boldsymbol{\mathcal{X}}})}\right| + \left|\sqrt{\hat{h}(\bar{\boldsymbol{\mathcal{X}}})} - \sqrt{h(\bar{\boldsymbol{\mathcal{X}}})}\right| + \left|\sqrt{h(\bar{\boldsymbol{\mathcal{X}}})} - \sqrt{h(\boldsymbol{\mathcal{X}})}\right|$$

$$\leq \frac{\epsilon}{\sqrt{|\Omega|}} + \sqrt{g(\Omega)} + \frac{\epsilon}{\sqrt{n^d}}.$$

Using the definition of $g(\Omega)$ and letting $\epsilon = 3d$, we have

$$\sup_{\boldsymbol{\mathcal{X}} \in \mathcal{S}} \left| \sqrt{\hat{h}(\boldsymbol{\mathcal{X}})} - \sqrt{h(\boldsymbol{\mathcal{X}})} \right|$$

$$\leq \frac{2\epsilon}{\sqrt{|\Omega|}} + \left( \frac{\varsigma^2}{2} \left( d \log n + \log |\mathcal{S}| \right) \left( \frac{1}{|\Omega|} - \frac{1}{n^d} + \frac{1}{n^d |\Omega|} \right) \right)^{1/4}$$

$$\leq \frac{2\epsilon}{\sqrt{|\Omega|}} + 2\varepsilon \left( \frac{d \log n + \log |B|}{2|\Omega|} \right)^{1/4}.$$

$\square$

### E.3 Proof of Lemma 3

*Proof.* Case (a) can be easily obtained by transforming the entropy number result of the special case $p = u = r = 2$ (we have exchanged $p$ and $q$) of Theorem 13 in (Mayer & Ullrich, 2021) to covering number. Specifically, let

$$e_\eta \leq c_q \left( \frac{\log(ek/\eta) + n}{\eta} \right)^{1/q - 1/2} \leq c_q \left( \frac{\log(ek) + n}{\eta} \right)^{1/q - 1/2},$$

where $c_q$ is a constant depending only on $q$ and $c_q = O(1/q)$. It follows that

$$\eta \leq (n + \log(ek)) \left( \frac{c_q}{e_\eta} \right)^{2q/(2-q)}.$$

Then the covering number is bounded as

$$\log \mathcal{N}(\mathcal{B}_{2,q}^{n,k}, \|\cdot\|_F, \epsilon) \leq (n + \log(ek)) \left( \frac{c_q}{\epsilon} \right)^{2q/(2-q)} \log 2$$
$$= c_q'(n + \log(ek)) \epsilon^{-2q/(2-q)},$$

where $c_q' = O(1/q)$.

Case (b) is a special case of Lemma 3.2 of (Bartlett et al., 2017). Namely, in the lemma, letting $\boldsymbol{X}$ be an identity matrix and $p = q = 2$ and $\frac{1}{r} + \frac{1}{s} = 1$, renaming the variables, we have $\log \mathcal{N}(\mathcal{B}_{2,q}^{n,k}, \|\cdot\|_F, \epsilon) \leq \lceil nk^{2(q-1)/q} \epsilon^{-2} \rceil \log(2nk)$.

$\square$

### E.4 Proof of Lemma 6

*Proof.* Let $S = \{i : z_i > \tau\}$. We have

$$\|\boldsymbol{z}\|_1 = \|\boldsymbol{z}_S\|_1 + \sum_{i \notin S} z_i \leq \sqrt{|S|} \|\boldsymbol{z}\|_2 + \tau \sum_{i \notin S} \frac{z_i}{\tau}. \tag{52}$$

Since $\frac{z_i}{\tau} \leq 1$ and $0 < p \leq 1$, we have

$$\|\boldsymbol{z}\|_1 \leq \sqrt{|S|} \|\boldsymbol{z}\|_2 + \tau \sum_{i \notin S} \left( \frac{z_i}{\tau} \right)^p \leq \sqrt{|S|} \|\boldsymbol{z}\|_2 + \tau^{1-p} \|\boldsymbol{z}\|_p^p. \tag{53}$$

On the other hand, we have

$$|S| \tau^p \leq \sum_{i \in S} z_i^p \leq \|\boldsymbol{z}\|_p^p. \tag{54}$$

Combining (53) and (54), we arrive at

$$\|\boldsymbol{z}\|_1 \leq \tau^{-p/2} \|\boldsymbol{z}\|_2 \sqrt{\|\boldsymbol{z}\|_p^p} + \tau^{1-p} \|\boldsymbol{z}\|_p^p. \tag{55}$$

$\square$

### E.5 Proof of Lemma 7

This is a special case of Corollary 2 in Tomioka & Suzuki (2014).

### E.6 Proof of Lemma 8

*Proof.* The Chernof bound of Gaussian distribution indicates

$$\mathbb{P}[|\mathcal{N}_{j_1 j_2 \cdots j_d}| \geq t] \leq 2e^{-\frac{t^2}{2\sigma^2}}, \quad (j_1, j_2, \ldots, j_d) \in [n] \times [n] \cdots \times [n]. \tag{56}$$

Using Boole's inequality (union bound), we obtain

$$\mathbb{P}\left[ \bigcup_{(j_1, j_2, \ldots, j_d)} |\mathcal{N}_{j_1 j_2 \cdots j_d}| \geq t \right] \leq \sum_{(j_1, j_2, \ldots, j_d)} \mathbb{P}\left[|\mathcal{N}_{j_1 j_2 \cdots j_d}| \geq t\right]. \tag{57}$$

It follows that

$$\mathbb{P}\left[\|\boldsymbol{\mathcal{N}}\|_\infty \leq t\right] \geq 1 - 2n^d 2e^{-\frac{t^2}{2\sigma^2}}. \tag{58}$$

Let $t = \sqrt{2\sigma^2 \log(n^{2d})}$. Then

$$\mathbb{P}\left[\|\boldsymbol{\mathcal{N}}\|_\infty \leq 2\sigma\sqrt{d\log(n)}\right] \geq 1 - 2n^{-d}. \tag{59}$$

$\square$

