# OpenReview forum: "Euclidean-Norm-Induced Schatten-p Quasi-Norm Regularization for Low-Rank Tensor Completion and Tensor Robust Principal Component Analysis"
_TMLR — Accepted by TMLR_

### Review · Reviewer_Co8K · 2022-07-18

**Summary Of Contributions:**

This paper extends the Shatten-$p$ (quasi-)norm from matrix to tensor under CP decomposition. The proposed norm is applied for tensor completion and tensor RPCA. The authors also establishes upper bounds on the estimation error for tensor completion. Experiments on both synthetic and real datasets show the effectiveness of the propose algorithms.

**Broader Impact Concerns:**

It seems that this work has no  ethical implications.

**Requested Changes:**

This proposal is new and interesting. I suggest several minor changes as follows:

Change 1:  Before giving new notions with equations, it is suggested to add more explanations to make the new notions easier to follow.

C1.1) Give interpretations of "Definition 4. the inverse CP operator".

C1.2) In Definition 5, the set $\mathcal{S}^{\bot}_{d,n}$ is directly defined in equations. It is suggested to provide some literal interpretations for it.


Change 2: The proposed algorithms seem a bit involved, and the implementation may require much effort in parameter tuning. Can the implementation be open-sourced with sufficient explains on the parameter tuning?

Change 3: The relationship between the proposed norm and the matrix Shatten- (quasi-)norm seems not well explained. It is suggested to give such discussions.

Change 4: Go through a thorough proofreading of the paper.

**Strengths And Weaknesses:**

[Strengths]

Strength 1: To the best of my knowledge, this is the first paper that extends the Shatten-$p$ (quasi-) norm from matrix to tensor under CP decomposition. The new (quasi-)norm is new and interesting.

Strength 2: The rigorous theoretical analysis of the proposed tensor completion model is new and of theoretical significance. The covering number-based analysis sheds new lights on error analysis of tensor recovery.

Strength 3: The empirical performance of the proposed algorithms is promising.

[Weaknesses]

Weakness 1: The writing can be improved by adding more explanations of the new notions.

Weakness 2: The proposed algorithms seem a bit involved, and the implementation may require much effort in parameter tuning.

Weakness 3: The relationship between the proposed norm and the matrix Shatten-$p$ (quasi-)norm seems not well explained.

---

### Review · Reviewer_2bj2 · 2022-07-18

**Summary Of Contributions:**

The paper proposes a new class of tensor regularizers based on the mixed Euclidean norms of tensor CP factors, and then adopt it to the tensor completion framework and the robust tensor principal component analysis, respectively. The generalization error bound for low-rank tensor completion is discussed and provided. Numerical experiments on synthetic and image data sets have empirically shown the proposed performance. Overall, the work is interesting with certain novelty in the proposed tensor regularization framework. However, there are several issues that need to be addressed, e.g., clarity and organization.

**Requested Changes:**

1. There are some minor typos and wording issues that should be carefully checked, e.g., "syhthetic" -> "synthetic" in the abstract, "noise density" -> "noise level".
2. In Definition 4, what is the value or the proper range of $k$? Is it a pre-assigned integer or simply the rank of $\mathcal{X}$?
3. In Theorem 2, if $q=1/3$, then $p_1=1/5$ which is not included in Table 1. Besides, it would be clearer to specify the values of $p,q,p_1,p_2$ in Table 1 to get straightforward connections with Theorems 1&2.
4. In the proposed model (2), should $k$ be replaced by $r$ since $\mathcal{T}$ is a rank-$r$ tensor that is specified at the beginning of Section 3.1? In addition, if $k$ is a pre-assigned number, does that pose a hidden low-rank assumption on the unknown $\mathcal{X}$? If this is the case, both proposed models in Section 3.1 may not be equivalent to the original unconstrained low-rank tensor completion models.
5. In Figure 1, it is not clear how the different low rankness is assigned to get different decay rates. What are "decay rates"? Detailed description of generation of those curves in the figure should be given. Why is "reducing $p$ is more useful when the decay is low"? What is the usage of $p$ being small here?
6. In Section 3.2, it mentions the "generalization error" which is not a common term and should be defined rigorously somewhere.
7. In the comment after Theorem 6, "a smaller $p$ but not too small" is confusing.
8. The organization of Section 5 could be improved. The computer configuration in p.10 should be moved to before Sec.5.1 if it applies to both experiment settings. Common comparison metrics should be defined or explained at the very beginning as well. On the right subfigure of Figures 3&6, the horizontal scale is strange since those values of $p$ are not evenly distributed.

**Strengths And Weaknesses:**

Strengths:
The proposed family of regularizers is novel which includes the symmetric and asymmetric versions. Numerical results have shown they are useful in improving the tensor recovery accuracy.

Weaknesses:
1. Some definitions are confusing without a full explanation of each single notation.
2. Complete discussion of exact recovery guarantees is missing.
3. Although a standard ADMM is applied, algorithm description/convergence analysis should be provided in detail.

---

### Review · Reviewer_ssb8 · 2022-08-09

**Summary Of Contributions:**

The paper proposed a general class of tensor rank regularizers based on CP decomposition. This class of regularizers are monotonic transformation of tensor Schatten-p quasi-norm based regularizers (p<1). The paper propose two variations of such regularizers - symmetric and assymetric. Such regularizers enable minimize Schatten-p quasi-norms of low-rank (CP) tensors efficiently, without performing SVD. The paper employ these regularizers in two applications - low-rank tensor completion (LRTC) and tensor robust principal component analysis (TRPCA). The paper also provide generalization error bounds for LRTC problem with Schatten-p quasi-norm regularization and with the proposed regularizers.


**Broader Impact Concerns:**

No comments.

**Requested Changes:**

Please refer to Strengths And Weaknesses section. It will be great if the concerns can be addressed. I will then revisit my recommendation.

**Strengths And Weaknesses:**

Strength

1. The paper provides a good literature survey of existing works related to the propsed problem.
2. Theorem 1 and Theorem 2 provide interesting variational form of the tensor Schatten-p quasi norm for CP-decomposed tensors.

Weakness and Questions

1. The paper writeup and presentation should be improved as its focus. Few comments related to this issue:

	a. The paper introduces the symmetric/assymmetric regularizers in Section 2. It also tries to justify the utility of assymmetric regularizers in Section 3. However, in Section 3, when the paper briefly discusses the optimization algorithm, no specific discussion is done for them. Does the paper suggest different optimization strategies for symmetric and assymmetric regularizers (since one class has less non-smooth terms)? If yes, then it should be discussed. Similar comment for Section 4.1.

	b. The discussion of generalization results in Section 3.2 should be improved. As an instance, the text below Theorem 4 has the phrase ".. a smaller p but not too small will lead to a tighter error bound ..". Can such statements be concretized? For example, what does too small mean here.

	c. Regarding Figure 1 and similar toy experiments - While the intuitive example of error bound is useful in toy datasets, can some practical algorithm/heuristic be devised (for tuning p) from the given theoretical results?

	d. The significance of the different theoretical results in this work should be made clearer. The paper should also discuss these results in the light of other generalization results on low-rank CP decomposition of tensors.

	e. It is not clear how 'q' appears in LRTC-ENR (2) - for example see the sentence below fig 1 caption. Figure 2 also shows results w.r.t. q rather than p.

	f. If a section has only one subsection (Section 4 has only Section 4.1), then that subsection need not be made explicitly.

	g. The paper should ideally use the same size font everywhere in main text (except what is permitted by the style file). For example the discussion of fig 3 results is in smaller font size.

	h. Figure 6 x-axis is not legible.


2. In fig 3, it seems that LRTC-ENR symmetric performs similar to LRTC-ENR asymmetric. The paper states that "the asymmetric regularization slightly outperforms the symmetric regularization in a few cases". This seems incorrect as in plots (a)-(c), performance of asym-BCDE and sym-BCDE is similar while sym-LBFGS seems to do better than asym-LBFGS. Similar comment for (d), expect for p <0.1, when asym-LBFGS is better than sym-LBFGS. How does the paper reconcile the intuitive advantage of assymetric regularizers (in Section 2) with the empirical results?

3. Rprecon is Tucker decomposition based low-rank tensor completion method. Its rank hyper-parameters should be different than CP-decomposition based methods. For example, Rprecon will take 3 difference ranks (r1,r2,r3) for d=3 tensor. Since the paper treats its rank same as CP-rank, it found Rprecon to have high computational costs. Please set (r1,r2,r3) for Rprecon according to the best practices of Tucker decomposition.

4. In the experiment section, should not the results with Schatten-p quasi-norm p=1/3,2/3, and 1 be attributed to the formulations proposed in Bazerque et al.(2013), Yang et al.(2016), and  Shi et al. (2017), respectively? This is because these quasi-norms were first studied in these works. This may have a broader impact on the work's empirical contribution as the paper might need to empirically show the utility of non-standard p-norm values (those other than 1/3, 2/3 and 1) in the experiments.

5. In Section 2, it is suggested that asymmetric regularizer (Theorem 2b) would be easier to optimize as it involves fewer non-smooth terms than those in Theorem 1. However, since Theorem 2b involves one non-smooth term, will not it make the optimization equally hard? It will be great if this intuition can be justified more rigorously, e.g., theoretically or empirically. It should be noted that empirically this does not seem to hold (see comment 2).

---

### Decision · Action_Editors · 2022-09-23

**Recommendation:** Accept with minor revision

**Comment:**

As the reviews suggest, there are certainly new contributions to the norms' generalization and analysis of the problems. To this end, there seems to be no confusion in the reviewers’ minds on the merits. However, after having a deep look at the paper, especially at the experiments section, I feel certain things can be done properly (look at the comments of the reviewers) to show the benefit of proposed general norms or at least put them in the right perspective and not overselling them.

Statements like “The numerical results of LRTC and TRPCA on synthetic data, image inpainting, and image denoising corroborate the effectiveness and superiority of our methods over state-of-the-art baseline methods.” are not being justified from the experiments, and therefore, problematic. Please tone down such statements.

The font size of Figure 15 looks very odd.

I believe the paper can be accepted after some fair amount of revision that I would like to see and follow up:
1. both polishing the paper storyline and organization, and
2. relooking at experiments to put the contributions in the right perspective,
and hence, the decision. Please address all the comments of all the reviewers.